



# Trends in soil solution dissolved organic carbon (DOC) concentrations across European forests

M. Camino-Serrano[1], E. Graf Pannatier[2], S. Vicca[1], S. Luyssaert[3], M. Jonard[4], P. Ciais[3], B. Guenet[3], B. Gielen[1], J. Peñuelas[5,6], J. Sardans[5,6], P. Waldner[2], S. Etzold[2], G. Cecchini[7], N. Clarke[8], Z. Galić[9], L. Gandois[10,11], K. Hansen[12], J. Johnson[13], U. Klinck[14], Z. Lachmanová[15], A.J. Lindroos[16], H. Meesenburg[14], T.M. Nieminen[16], T.G.M. Sanders[17], K. Sawicka[18], W. Seidling[17], A. Thimonier[2], E. Vanguelova[19], A. Verstraeten[20], L. Vesterdal[21], I.A. Janssens[1]

[1]{Department of Biology, PLECO, University of Antwerp, Belgium}

[2] {WSL, Swiss Federal Institute for Forest, Snow and Landscape Research, Zürcherstrasse 111, 8903, Birmensdorf, Switzerland}

[3] {Laboratoire des Sciences du Climat et de l'Environnement, LSCE/IPSL, CEA-CNRS-UVSQ, Université Paris-Saclay, F-91191 Gif-sur-Yvette, France}

[4] {UCL-ELI, Université catholique de Louvain, Earth and Life Institute, Croix du Sud 2, BE-1348 Louvain-la-Neuve , Belgium}

[5] {CREAF, Cerdanyola del Vallès,  Catalonia, Spain}

[6] {CSIC, Global Ecology Unit CREAF-CSIC-UAB,  Cerdanyola del Vallès,  Catalonia, Spain}

[7] {Earth Sciences Department, University of Florence, Italy}

[8] {Norwegian Institute of Bioeconomy Research, , N-1431, Ås, Norway}

[9] {University of Novi Sad-Institute of Lowland Forestry and Enviroment, Serbia}

[10] {Université de Toulouse: UPS, INP, EcoLab (Laboratoire Ecologie fonctionnelle et Environnement), ENSAT, Castanet-Tolosan, France}

[11] {CNRS, EcoLab, Castanet-Tolosan, France}

[12] {IVL Swedish Environmental Research Institute, Natural Resources & Environmental Effects, SE-100 31, Stockholm, Sweden}



[13] {UCD School of Agriculture and Food Science, University College Dublin, Belfield,
Ireland}
[14] {Northwest German Forest Research Station, Grätzelstr. 2, D-37079, Göttingen,
Germany}
[15] {FGMRI, Forestry and Game Management Research Institute, Strnady 136, 252 02
Jíloviště,Czech Republic}
[16] {Natural Resources Institute Finland (Luke), P.O. Box 18, 01301 Vantaa, Finland }
[17] {Thünen Institute of Forest Ecosystems, Alfred-Möller-Strasse 1, D-16225, Eberswalde,
Germany}
[18] {University of Reading, Environmental Science, UK}
[19] {Centre for Ecosystem, Society and Biosecurity, Forest Research, Alice Holt Lodge,
Wrecclesham, Farnham, Surrey GU10 4LH, UK}
[20] {Research Institute for Nature and Forest (INBO), Kliniekstraat 25, BE-1070 Brussels,
Belgium}
[21] {University of Copenhagen, Department of Geosciences and Natural Resource
Management, Rolighedsvej 23, DK-1958 Frederiksberg C, Denmark}
Correspondence to: M. Camino-Serrano (marta.caminoserrano@uantwerpen.be)

## 46    **Abstract**

Dissolved organic carbon (DOC) in soil solution is connected to DOC in surface waters
through hydrological flows. Therefore, it is expected that long-term dynamics of DOC in
surface waters reflect DOC trends in soil solution. However, a multitude of site-studies has
failed so far to establish consistent trends in soil solution DOC, whereas increasing
concentrations in European surface waters over the past decades appear to be the norm,
possibly as a result from acidification recovery. The objectives of this study were therefore to
understand the long-term trends of soil solution DOC from a large number of European
forests (ICP Forests Level II plots) and determine their main physico-chemical and biological
controls. We applied trend analysis at two levels: 1) to the entire European dataset and 2) to
the individual time series and related trends with plot characteristics, i.e., soil and vegetation
properties, soil solution chemistry and atmospheric deposition loads. Analyses of the entire





dataset showed an overall increasing trend in DOC concentrations in the organic layers, but,
at individual plots and depths, there was no clear overall trend in soil solution DOC across
Europe with temporal slopes of soil solution DOC ranging between -16.8% $yr^{-1}$ and +23% $yr^{-1}$
(median= +0.4% $yr^{-1}$). The non-significant trends (40%) outnumbered the increasing (35%)
and decreasing trends (25%) across the 97 ICP Forests Level II sites. By means of
multivariate statistics, we found increasing DOC concentrations with increasing mean nitrate
($NO_3^-$) deposition and decreasing DOC concentrations with decreasing mean sulphate ($SO_4^{2-}$)
deposition, with the magnitude of these relationships depending on plot deposition history.
While the attribution of increasing trends in DOC to the reduction of $SO_4^{2-}$ deposition could
be confirmed in N-poorer forests, in agreement with observations in surface waters, this was
not the case in N-richer forests. In conclusion, long-term trends of soil solution DOC
reflected the interactions between controls acting at local (soil and vegetation properties) and
regional (atmospheric deposition of $SO_4^{2-}$ and inorganic N) scales.
**1  Introduction**
Dissolved organic carbon (DOC) in soil solution is the source of much of the terrestrially
derived DOC in surface waters (Battin et al., 2009; Bianchi, 2011; Regnier et al., 2013). Soil
solution DOC in forests is connected to streams through different hydrological pathways:
DOC mobilized in the forest floor may be transported laterally at the interface of forest floor
and mineral soil to surface waters or percolates into the mineral soil, where additional DOC
can be mobilized and/or DOC is partly adsorbed on particle surfaces and mineralized
thereafter. From the mineral soil DOC may be either leached laterally or vertically via
groundwater into surface waters (Mcdowell and Likens, 1988). Therefore, it could be
expected that long-term dynamics of DOC in ecosystem soil solutions mirror those observed
in surface waters.
Drivers related to climate change (temperature increase, precipitation change, atmospheric
$CO_2$ increase), the decrease in acidifying deposition or land use change and management may
individually or jointly explain trends in surface water DOC concentrations (Evans et al.,
2012; Freeman et al., 2004; Oulehle et al., 2011; Sarkkola et al., 2009; Worrall and Burt,
2004). Increasing air temperatures warm the soil, thus stimulating soil organic matter (SOM)
decomposition through greater microbial activity (Davidson and Janssens, 2006; Hartley and
Ineson, 2008; Kalbitz et al., 2000). Other drivers, such as increased atmospheric $CO_2$ and the
accumulation of atmospherically deposited inorganic nitrogen are thought to increase the
sources of DOC by enhancing primary plant productivity (i.e., through stimulating root



exudates, litterfall) (Sucker and Krause, 2010). Changes in precipitation, land use and
management (e.g. drainage of peatlands, changes in forest management or grazing systems)
may alter the flux of DOC leaving the ecosystem but no consistent trends in the hydrologic
regime or due to land use changes were detected in areas where increasing DOC trends have
been observed (Monteith et al., 2007).
Recent focus was mainly on decreasing acidifying deposition as an explanatory factor for
DOC increases in surface waters in Europe and North America by means of decreasing ionic
strength (Hruška et al., 2009) and increasing the pH of soil solution, consequently increasing
DOC solubility (Evans et al., 2005; Haaland et al., 2010; Monteith et al., 2007). Although the
hypothesis of an increase in surface water DOC concentration due to a recovery from past
acidification was confirmed in studies of soil solution DOC in the UK and Northern Belgium
(Vanguelova et al., 2010; Verstraeten et al., 2014), it is not consistent with observed trends in
soil solution DOC concentrations measured in Finnish, Norwegian, and Swedish forests
(Löfgren and Zetterberg, 2011; Ukonmaanaho et al., 2014; Wu et al., 2010). This
inconsistency between soil solution DOC and stream DOC trends could suggest that DOC in
surface water and soil solution responds differently to (changes in) environmental conditions
in different regions (Akselsson et al., 2013; Clark et al., 2010; Löfgren et al., 2010).
Alternatively, other factors such as tree species and soil type, may be co-governing organic
matter dynamics and input, generation and retention of DOC in soils.
Trends of soil solution DOC not only vary among forests but often also within the same site
(Löfgren et al., 2010). Forest characteristics such as tree species composition, soil fertility,
texture or sorption capacity may affect the response of soil solution DOC to environmental
controls, for instance, by controlling the rate of soil acidification through soil buffering and
nutrient plant uptake processes (Vanguelova et al., 2010). Within a site, DOC variability with
soil depth is typically caused by different intensity of DOC production, transformation and
sorption along the soil profile. Positive temporal trends in soil solution DOC (increasing
concentrations over time) are frequently reported for the organic layers and shallow soils
where production and decomposition processes control the DOC concentration (Löfgren and
Zetterberg, 2011). However, no dominant trends are found for the mineral soil horizons,
where physico-chemical processes, such as sorption, become more influential (Borken et al.,
2011; Buckingham et al., 2008). Furthermore, previous studies have used different temporal
and spatial scales which may have further added to the inconsistency in the DOC trends
reported in the literature (Clark et al., 2010).



In this context, the International Co-operative Programme on Assessment and Monitoring of
Air Pollution Effects on Forests (ICP Forests, 2010) compiled a unique dataset containing
data from more than 100 intensively monitored forest plots (Level II) which allow to unravel
regional trends in soil solution DOC of forests at European scale, and perform statistical
analysis of the main controls behind these regional trends. Long-term measurements of soil
solution DOC are available for these plots, along with information on aboveground biomass,
soil properties, and atmospheric deposition of inorganic N and $SO_4^{2-}$, collected using a
harmonized sampling protocol across Europe (Ferretti and Fischer, 2013). This dataset has
previously been used to investigate the spatial variability of DOC in forests at European scale
(Camino-Serrano et al., 2014), but an assessment of the temporal trends in soil solution DOC
using this large dataset has not been attempted so far. The main objective of this study was to
understand the long-term temporal trends of DOC concentrations in soil solution measured at
the ICP Forests Level II plots across Europe. Based on the increasing DOC trends in surface
water, we hypothesized that temporal trends in soil solution DOC would also be positive, but
with trends varying locally depending on plot characteristics. We further investigated whether
plot characteristics, specifically climate, inorganic N and $SO_4^{2-}$ deposition loads, forest type,
soil properties, and changes in soil solution chemistry can explain differences across sites in
DOC trends.
**2    Materials and Methods**
**2.1    Data description**
Soil solution chemistry has been monitored within the ICP Forests Programme since the
1990s on most Level II plots. The ICP Forests data were extracted from the pan-European
Forest Monitoring Database (Granke, 2013). A list of the Level II plots used for this study
can be found in Supplementary material S1, Table S1. The methods for collection and
analysis of soil solution used in the various countries (Switzerland: Graf Pannatier et al.
(2011); Flanders: Verstraeten et al. (2012); Finland: Lindroos et al. (2000); UK: Vanguelova
et al. (2010), Denmark: Hansen et al. (2007)) follow the ICP Forests manual (Nieminen,
2011). Generally, lysimeters were installed at several fixed depth intervals starting at 0 cm,
defined as the interface between the surface organic layer and underlying mineral soil. These
depths are typically aligned with soil "organic layer", "mineral topsoil", "mineral subsoil"
and "deeper mineral soil" but sampling depths vary among countries and even among plots
within a country. Normally, zero-tension lysimeters were installed under the surface organic



layer and tension lysimeters within the mineral soil. However, in some countries zero-tension
lysimeters were also used within the mineral layers and in some tension lysimeters below the
organic layer. Multiple collectors (replicates) were installed per plot and per depth to assess
plots spatial variability. However, in some countries, samples from these replicates were
pooled before analyses or averaged prior to data transmission. The quality assurance and
control procedures included the use of control charts for internal reference material to check
long-term comparability within national laboratories as well as participation in periodic
laboratory ring tests (e.g., Marchetto et al., 2011) to check the international comparability.
Data were reported annually to the pan-European data center, checked for consistency and
stored in the pan-European Forest Monitoring Database (Granke, 2013).
Soil water was usually collected fortnightly or monthly, although for some plots sampling
periods with sufficient soil water for collection were scarce, especially in prolonged dry
periods or in winter due to snow and ice. After collection, the samples were filtered through a
0.45 µm membrane filter, stored below 4 °C and then analyzed for DOC, together with other
soil solution chemical properties ($NO_3^-$, $Ca^{2+}$, $Mg^{2+}$, $NH_4^+$, $SO_4^{-2}$, total dissolved Al, total
dissolved Fe, pH, electrical conductivity). The precision of DOC analysis differed among the
laboratories. The coefficient of variation of repeatedly measured reference material was 3.7%
on average. The time span of soil solution time series used for this study ranged from 1991 to
2011, although coverage of this period varied from plot to plot (Supplementary material S1,
Table S1).
Soil properties, bulk and throughfall atmospheric deposition of $NO_3^-$, $NH_4^+$ and $SO_4^{2-}$,
meteorological variables and stem volume increment were also measured at the plots. Stem
volume growth was calculated by the ICP Forests network from diameter at breast height
(DBH), live tree status, and tree height which were assessed for every tree (DBH > 5 cm)
within a monitoring plot approximately every five years since the early 1990. Tree stem
volumes were derived from allometric relationships based on diameter and height
measurements according to De Vries et al. (2003), accounting for species and regional
differences. Stem volume growth (in $m^3$) between two consecutive inventories was calculated
as the difference between stem volumes at the beginning and the end of one inventory period
for living trees. Stem volume data were corrected for all trees that were lost during one
inventory period, including thinning. Stem volume at the time of disappearance (assumed at
half of the time of the inventory period) was estimated from functions relating stem volume
of standing living trees at the end of the period *vs* volume at the beginning of the period. The





methods used for collection of these data can be found in the Manuals of the ICP Forests
Monitoring Programme (ICP Forests, 2010). The soil properties at the plots used for this
study were derived from the ICP Forests aggregated soil database (AFSCDB.LII.2.1) (Cools
and De Vos, 2014).
Since continuous precipitation measurements are not commonly available for the Level II
plots, precipitation measurements for the location of the plots were extracted from the
observational station data of the European Climate Assessment & Dataset (ECA&D) and the
ENSEMBLES Observations (E-OBS) gridded dataset (Haylock et al., 2008). We used
precipitation measurements extracted from the E-OBS gridded dataset to improve the
temporal and spatial coverage and to reduce methodological differences of precipitation
measurements across the plots. The E-OBS dataset contains daily values of precipitation and
temperature from stations data gridded at 0.25 degrees resolution. When E-OBS data was not
available, it was gap-filled with ICP Forests precipitation values gained by deposition
measurements where available (open field bulk deposition or throughfall deposition).

## 203  2.2  Data preparation

We extracted data from plots with time series covering more than 10 years and including
more than 60 observations of soil solution DOC concentrations of individual or groups of
collectors. Outliers, defined as $\pm$ 3 interquartile range of the 25 and 75 quantiles of the time
series, were removed from each time series to avoid influence of few extreme values in the
long-term trend (Schwertman et al., 2004). Values under 1 mg $L^{-1}$, which is the detection
limit for DOC in the ICP Level II plots, were replaced by 1 mg $L^{-1}$. After this filtering, 529
time series from 118 plots, spanning from Italy to Norway, were available for analysis. Soil
solution, precipitation, and temperature were aggregated to monthly data by the median of the
observations in each month and by the sum of daily values in the case of precipitation. Data
of inorganic N ($NH_4^+$ and $NO_3^-$) and $SO_4^{2-}$ canopy throughfall and open field bulk deposition
measured at the plots were interpolated to monthly data (Waldner et al., 2014).
The plots were classified according to their forest type (broadleaved/coniferous dominated),
soil type (World Reference Base, Reference Soil Group (WRB 2006)), their stem growth
(slow, < 6 $m^3$ $ha^{-1}$ $yr^{-1}$, intermediate, 6–12 $m^3$ $ha^{-1}$ $yr^{-1}$; and fast, > 12 $m^3$ $ha^{-1}$ $yr^{-1}$), and soil pH
(low, <4.2, intermediate, 4.2–5, high, >5). Plots were also classified based on throughfall
inorganic N ($NO_3^-$ +$NH_4^+$) deposition level, defined as: high deposition (HD, >15 kg N $ha^{-1}$
$yr^{-1}$), medium deposition (MD, 5–15 kg N $ha^{-1}$ $yr^{-1}$), and low deposition (LD, <5 kg N $ha^{-1}$ $yr^-$



$^{1}$) and throughfall $SO_4^{2-}$ level, defined as: high deposition (HD, >6 kg S ha$^{-1}$ yr$^{-1}$), and low
deposition (LD, < 6 kg S ha$^{-1}$ yr$^{-1}$).

### 2.3   Statistical methods

The sequence of methods applied is summarized in Fig. 1. The analysis of temporal trends in
soil solution DOC concentrations was carried out at two levels: 1) the European level and 2)
the plot level of each individual time series. While the first analysis allows an evaluation of
the overall trend in soil solution DOC at a continental scale, the second analysis indicates
whether the observed large scale trends are occurring at local scales as well, and tests whether
local trends in DOC can be attributed to certain driver variables.
Linear mixed-effects models (LMM) were used to detect the temporal trends in soil solution
DOC concentration at the European scale (Fig. 1). For these models, the selected 529 time
series were used. For the trend analysis of individual time series, however, we focused on the
investigation of the potential long-term trends in soil solution DOC at European forests that
show monotonicity. Therefore, DOC time series were first analyzed using the Breaks For
Additive Seasonal and Trend (BFAST) algorithm to detect the presence of breakpoints
(Verbesselt et al., 2010) with the time series showing breakpoints, i.e., not monotonic, being
discarded (Supplementary material S2.2.) (Fig. 1). Then, monotonic trend analyses were
carried out using the Seasonal Mann Kendall (SMK) test for monthly DOC concentrations
(Hirsch et al., 1982; Marchetto et al., 2013). Partial Mann Kendall (PMK) tests were also
used to test the influence of precipitation as a co-variable to detect if the trend might be due
to a DOC dilution/concentration effect (Libiseller and Grimvall, 2002). Sen (1968) slope
values were calculated for SMK and PMK. Moreover, LMMs were performed again with the
filtered dataset to compare results with and without time series showing breakpoints (Fig. 1).
For this study, five depth intervals were considered: the organic layer (0 cm), topsoil (0-20
cm), intermediate (20-40 cm), subsoil (40-80 cm) and deep subsoil (> 80 cm). The slopes of
each time series were standardized by dividing them by the median DOC concentration over
the sampling period, aggregated to a unique plot-soil depth slope and classified by the
direction of the trend as significantly positive (P, $p < 0.05$), significantly negative (N, $p <$
0.05), and non-significant (NS, $p \geq 0.05$). When there was more than one collector per depth,
the median of the slopes was used when the direction of the trend (P, N, or NS) was similar.
When the different trends at the same plot-soil depth combination were either P and NS, or N
and NS, it was marked as "Weighted positive" and "Weighted negative" to indicate that there





was potential predominant direction of the trend but with less significance. Trends for other
soil solution parameters ($NO_3^-$, $Ca^{2+}$, $Mg^{2+}$, $NH_4^+$, $SO_4^{2-}$, total dissolved Al, total dissolved
Fe, pH, electrical conductivity), precipitation and temperature were calculated using the same
methodology as for DOC. Finally, two multivariate statistical analyses were performed,
General Discriminant Analysis (GDA) and Structural Equation Models (SEM), to investigate
the main factors explaining differences in DOC trends among the selected plots (Fig. 1). All
the statistical analyses were performed in R software version 3.1.2 (R Core Team, 2014)
using the "rkt" (Marchetto et al., 2013), "bfast01" (de Jong et al., 2013) and "sem" (Fox et
al., 2013) packages, except for the GDA that was performed using Statistica 8.0 (StatSoft,
Inc. Tule, Oklahoma, USA) and the LMMs that were performed using SAS 9.3 (SAS
institute, Inc., Cary, NC, USA). More detailed information on the statistical methods used can
be found in Supplementary material S2.
**3  Results**
**3.1  Soil solution DOC trends at European scale**
First, temporal trends in DOC were analyzed for all the European DOC data pooled together
by means of LMM models to test for the presence of overall trends. A significantly increasing
DOC trend (p<0.05) in soil solution collected with zero-tension lysimeters in the organic
layer was observed mainly under coniferous forest plots (Table 1). Similarly, a significantly
increasing DOC trend (p<0.05) in DOC for soil solution collected with tension lysimeters
was found in deep mineral horizon (>80cm) for all sites, but mainly for coniferous forest sites
(Table 1). By contrast, non-significant trends were found in other mineral horizons (0-20 cm,
20-40 cm, 40-80 cm) by means of the LMM models. When the same analysis was applied to
the filtered European dataset, i.e., without the time series including breakpoints (see Sect.
3.2), fewer significant trends were observed: only an overall positive trend was found for
DOC in the organic layer using zero-tension lysimeters, again mainly under coniferous forest
sites but no statistically significant trends were found in the mineral soil (Table 1).
**3.2  Soil solution DOC trend analysis of individual time series**
3.2.1  Comparison of methods of individual trend analysis
We applied the BFAST analysis to select the monotonic time series in order to assure that the
overall detected trends were not influenced by breakpoints in the time series. Time series





with breakpoints represented more than 50% of the total time series aggregated by soil depth
interval (245 out of 436). In total, 191 plot-soil depth combinations from 97 plots were
analyzed after filtering out the time series showing breakpoints and 94% of the analyzed plot-
depth combinations showed consistent trends among replicates collected at the same depth. In
contrast, when also considering the time series with breakpoints, the trends calculated for
plot-depth combinations agreed only in 75% of the cases implying that the proportion of
contradictory trends within plot-depth combinations increased from 6% in the dataset without
breakpoints to 25% in the entire dataset (Fig. 2). For both datasets, the majority of the trends
were not statistically significant (44% and 41%, for the dataset with and without breakpoints,
respectively). In other words, filtering the time series for breakpoints reduced the within-plot
variability, while most of the plots showed similar aggregated trends per plot-depth
combinations. For this reason, the results discussed from here on correspond only to the
trends of monotonic (breakpoint filtered) time series of soil solution DOC concentrations.
There was a good agreement between results using the three methods: BFAST, SMK, and
PMK (Table 2). The direction and significance of the trend agreed for 84.5% of the time
series analyzed. For the majority of the remaining time series for which the trends did not
agree, BFAST did not detect a trend when SMK and PMK did, thus, the latter two methods
seemed more sensitive for trend detection than BFAST. Trends computed with SMK and
PMK agreed well.
For virtually all plots, including precipitation as a co-variable in the PMK test gave the same
result as the SMK test, which indicates that precipitation (through dilution or concentration
effects) did not affect the DOC concentration trends. Dilution/concentration effect was only
detected in four plots (Supplementary material S1, Table S1).

### 3.2.2    Soil solution DOC concentration trends using the SMK test

The individual trend analysis using the SMK test showed temporal slopes of soil solution
DOC concentration ranging from -16.8% yr$^{-1}$ to +23% yr$^{-1}$ (median= +0.4% yr$^{-1}$, interquartile
range = +4.3% yr$^{-1}$). Among all the time series analyzed, the majority were not statistically
significant trends (40%, 104 time series), followed by significantly positive trends (35%, 91
time series) and significantly negative trends (24%, 63 time series) (Table 2). There was,
thus, no uniform trend in soil solution DOC in forests across a large part of Europe. Although
a slight tendency of increasing trends in central and decreasing trends in north and south
Europe was observed (Fig. 3), the uneven number of analyzed time series for each country





315 (few in Austria, Italy or Finland and many in Germany) made it difficult to draw firm

316 conclusions about the spatial pattern of the trends in soil solution DOC concentrations in

317 Europe. Furthermore, the regional trend differences were inconsistent when looking at

318 different soil depth intervals separately (Fig. 4 and 5).

319 The variability in trends was high, not only at continental scale, but also at plot level (Fig. 6).

320 We found consistent within-plot trends only for 50 out of the 97 sites. Moreover, some plots

321 even showed different trends (P, N or NS) in DOC within the same depth interval, which was

322 the case for 17 plot-depth combinations (16 in Germany and one in Norway), evidencing a

323 high small-scale plot heterogeneity.

324 Trend directions often differed among depths. For instance, in the organic layer, we found

325 mainly non-significant trends and, when a trend was detected, it was more often positive than

326 negative, while positive trends were the most frequent in the subsoil (below 40 cm) (Table 2).

327 Nevertheless, it is important to note that a statistical test of whether there was a real

328 difference in DOC trends between depths was not possible as the set of plots differed

329 between the different soil depth intervals. However, a visual comparison of trends for the few

330 plots in which trends were evaluated for more than three soil depths showed that, at first

331 sight, there was no difference in DOC trends between soil depths (Supplementary material

332 S3, Fig. S1 and S2).

### 3.3 Factors explaining the direction and slopes of the soil solution DOC trends

335 A stratification of the forests into broadleaved and coniferous forest revealed no direct effect

336 of forest type on the direction of the statistically significant trends in soil solution DOC (Fig.

337 7C). Both positive and negative trends were equally found under broadleaved and coniferous

338 forests ($\chi^2$(1, n = 97) = 0.073, $p$ = 0.8). Increasing DOC trends, however, occurred more often

339 under forests with a mean stem growth less than 6 m$^3$ ha$^{-1}$ yr$^{-1}$ over the study period, whereas

340 decreasing DOC trends were more often associated with forests with a mean stem growth

341 between 6 and 12 m$^3$ ha$^{-1}$ yr$^{-1}$ ($\chi^2$(2, n = 53) = 5.8, $p$ = 0.05) (Fig. 7D).

342 Mean annual throughfall SO$_4^{2-}$ and inorganic N deposition both had a significant effect on the

343 direction of the trends in soil solution DOC (Fig. 7A, 7B). Increasing trends were more

344 frequent in forests with high or medium inorganic N deposition than in forests with low

345 inorganic N deposition where only decreasing trends were found ($\chi^2$(2, N = 57) = 9.58, $p$ =





0.008). Correspondingly, the probability of positive trends in soil solution DOC was higher at
high inorganic N deposition loads (Fig. 8A). Also throughfall $SO_4^{2-}$ deposition significantly
influenced the direction of the trend in soil solution DOC, with more positive trends found for
sites with high mean throughfall $SO_4^{2-}$ deposition (> 6 kg S ha$^{-1}$ yr$^{-1}$) than for sites with low
$SO_4^{2-}$ deposition ($\chi^2$(1, N = 57) = 8.75, $p$ = 0.003). However, while there were also relatively
more positive trends at high and medium $SO_4^{2-}$ than at low $SO_4^{2-}$ , this pattern is less clear
than for inorganic N deposition (Fig. 8B).
Regarding the soil properties, more than half of the plots showing a consistent increasing
DOC trend at all evaluated soil depth intervals were located in Cambisols, (6 out of 11 plots),
which are rather fertile soils, whereas plots showing consistent negative trends covered six
different soil types. Other soil properties, like clay content, cation exchange capacity or pH,
did not clearly differ between sites with positive and negative DOC trends (Table 3). It is
remarkable that trends in soil solution pH, $Mg^{2+}$ and $Ca^{2+}$ concentrations were similar across
plots with both positive and negative DOC trends. Soil solution pH increased distinctly in
almost all the sites, while $Ca^{2+}$ and $Mg^{2+}$ decreased markedly (Table 3). However, we found
evidence that soil acidity controlled the $SO_4^{2-}$ deposition effect on the trends of DOC in soil
solution (Fig. 9). In very acid soils, a higher mean $SO_4^{2-}$ deposition enhanced the temporal
increase in soil solution DOC, while in less acid soils, there was no significant effect of mean
$SO_4^{2-}$ on DOC trends. Finally, no significant correlations were found between trends in
temperature or precipitation and trends in soil solution DOC, with the exception of an
increasing trend in temperature in the soil depth interval 20-40 cm (r = 0.47, $p$ = 0.03).
Results from the GDA analysis showed a marginally significant separation of plot-soil depth
combinations with negative and positive DOC trends ($p$ = 0.06) (Fig. 10). Median soil
solution conductivity, median soil solution $NO_3^-$, and median soil solution $SO_4^{2-}$ were
significant in the model and thus played an important role in the distinction between positive
and negative DOC trends (Table 4). The fitted GDA model was able to predict 63.1% of the
variance in DOC trends within the first axis (Fig. 10).
To test whether the influence of stem growth and soil solution chemistry was related to the
effect of $SO_4^{2-}$ and/or $NO_3^-$ deposition on soil solution DOC, we applied SEM to determine
the capacity of these variables in explaining variability in the slope of DOC trends. We
evaluated the influence of both the annual mean (kg ha$^{-1}$ yr$^{-1}$) and the trends (% yr$^{-1}$) in
deposition and soil solution parameters.



### 3.3.1 Effects of mean deposition and soil solution parameters

Analyzing different models that could explain the DOC trends using the overall dataset
indicated both direct and indirect effects of the annual mean $SO_4^{2-}$ and $NO_3^-$ throughfall
atmospheric deposition on the slopes of DOC trends. The SEM accounted for 32.7% of the
variance in DOC trend slopes (Fig. 11A). This model identified a significantly negative direct
effect of $SO_4^{2-}$ deposition on trends in soil solution DOC. On the other hand, throughfall $NO_3^-$
deposition had a significantly positive direct effect on DOC trends (Fig. 11A).
The variables in the model that best explained temporal changes in DOC were the same for
the forests with low and medium N deposition; for both groups, $NO_3^-$ deposition and $SO_4^{2-}$
deposition (directly, or indirectly through its influence on plant growth) influenced the trend
in DOC (Fig. 11B). Mean $SO_4^{2-}$ deposition again had a significant negative effect on DOC
slopes, while $NO_3^-$ deposition had a significantly positive effect. The percentage of variance
in DOC trend slopes explained by the model was 33%. For the plots with high N deposition,
however, we found no model for explaining the trends in DOC using the mean annual $SO_4^{2-}$
and $NO_3^-$ throughfall deposition.

### 3.3.2 Effects of trends in deposition and soil solution parameters

When the SEM is applied using the trends in $SO_4^{2-}$ and $NO_3^-$ deposition instead of the mean
values, a positive significant effect of trend in $NO_3^-$ and a negative of $SO_4^{2-}$ deposition were
also apparent, but the latter was non-significant (Supplementary material S4, Fig. S3A).
However, the percentage of variance in DOC trend slopes explained by the model was now
much lower (16%). The SEM applied with the trends in $SO_4^{2-}$ and $NO_3^-$ throughfall
deposition for forests with low and medium N deposition explained 24.4% of the variance of
DOC trends, and showed a significantly negative effect of trends in $SO_4^{2-}$ deposition on
trends in DOC (Supplementary material S4, Fig. S3B).
For the forests with high N deposition, the best model used the relative trends in $SO_4^{2-}$, $NO_3^-$
deposition and in median soil solution conductivity (% $yr^{-1}$) as explaining variables (Fig.
11C). The relative trend slopes of $NO_3^-$ were positively related to the DOC trend slopes. Also
both the trend slopes of $SO_4^{2-}$ and $NO_3^-$ deposition affected the trend slopes of DOC
indirectly through an effect on the trends of soil solution conductivity, although acting in
opposite directions: while trends in $NO_3^-$ deposition negatively affected the trends on soil
solution conductivity, trends in $SO_4^{2-}$ deposition had a marginally significant positive effect
($p=0.06$) on the trends on soil solution conductivity. The trends in conductivity, in turn,



positively affected the trend slopes of DOC. The percentage of the variance in DOC trend
slopes explained by the model was 25% (Fig. 11C).
In summary, long-term trends in soil solution DOC were negatively related to mean $SO_4^{2-}$
deposition (except for sites with high N deposition, where the effects of mean and trends in
$SO_4^{2-}$ deposition were not significant, Fig. 11A and 11B versus 11C) and positively related to
N deposition (Fig. 11). Also, trends in soil solution DOC were negatively correlated with
trends in $SO_4^{2-}$ deposition when the N deposition was low or intermediate (Supplementary
material S4, Fig. S3).

## 418    4    Discussion

### 419    4.1    Trend analysis of soil solution DOC in Europe

#### 420    4.1.1    Are the many non-significant trends real?

Non-significant trends dominated the site-level DOC concentrations across the ICP Forests
network. Measurement precision, strength of the trend, and the choice of the method may all
affect trend detection (Sulkava et al., 2005; Waldner et al., 2014). Evidently, strong trends are
easier to detect than weak trends. To detect a weak trend, either very long time series or very
accurate and precise data are needed. The quality of the data is assured within the ICP Forests
by means of repeated ring tests that are required for all participating laboratories and the
accuracy of the data has been improved considerably over an eight years period (Ferretti and
König, 2013; König et al., 2013). However, the precision and accuracy of the dataset still
varies across countries and plots. By filtering out the time series with breakpoints and
removing outliers, we improved the overall quality of the data, and thus guaranteed that the
detected positive and negative trends were factual at a 0.05 significance level. Nevertheless,
we found a majority of non-significant trends. For these cases, we cannot state with certainty
that DOC did not change over time: it might be that the trend was not strong enough to be
detected, or that the data quality was insufficient for the period length available for the trend
analysis (more than 9 years in all the cases). For example, the mixed-effects models detected
a positive trend in the organic layer, and while many of the individual time series measured in
the organic layer also showed a positive trend, most were classified as non-significant trends
(Fig. 4). This probably led to an underestimation of trends that separately might not be strong
enough to be detected by the individual trend analysis but combined with the other European





data these sites may contribute to an overall trend of increasing DOC concentrations in soils
of European forests.
The uncertainty in the interpretation of the non-significant trends was compensated by using
the SMK and PMK tests applied to monthly data for the trend analysis, which can detect
weaker trends (Marchetto et al., 2013; Waldner et al., 2014). In summary, while there is
probability (at p<0.05) that the detected statistically significant trends are genuine and not
influenced by artifacts in the time series, the group of non-significant trends in DOC might
well contain plots with significant trends that could not (yet) be detected statistically.
Nevertheless, the selected trend analysis technique is the most suitable to detect weak trends,
thus reducing the chances of hidden trends within the non-significant trends category.

### 4.1.2  Analysis of breakpoints in the time series

Soil solution DOC time series measured with lysimeters are subject to possible interruptions
of monotonicity, which is manifested by breakpoints. For instance, installation effect,
collector replacement, local forest management, disturbance by small animals, or by single or
repeated canopy insect infestations may disrupt DOC concentrations through abrupt soil
disturbances and/or enhanced throughfall chemical input to soil (Akselsson et al., 2013;
Kvaalen et al., 2002; Lange et al., 2006; Moffat et al., 2002; Pitman et al., 2010). In general,
detailed information on the management history and other local disturbances was not
available for the majority of Level II plots, which hinders selection of individual monotonic
time series based on specific site conditions. The BFAST analysis allowed us to filter out
time series affected by local disturbances (natural or artefacts) from the dataset and retain
time series that represented monotonic trends. Thereby, we removed some of the within-plot
variability (Fig. 2) that might be caused by local factors not directly explaining the long-term
monotonic trends in DOC and thus complicating or confounding the trend analysis (Clark et
al., 2010).
In view of these results, we recommend that testing for monotonicity of the individual time
series is a necessary first step in this type of analyses and that the breakpoint analysis is an
appropriate tool to filter large datasets prior to analyzing the long-term temporal trends in
DOC concentrations. It is worth mentioning that, since our main goal was to study general
monotonic trends, we did not focus on finding the direct causes of breakpoints in time series.
Further work is needed to interpret the causes of these abrupt changes and verify if these are





artefacts or mechanisms, since they may also contain useful information on local factors
affecting DOC trends, such as forest management or extreme events (Tetzlaff et al., 2007).

### 4.1.3  Variability in soil solution DOC trends within plots

Even after removing sites with breakpoints in the time series, within-plot variability remained
high (median within-plot range: 3.3% $yr^{-1}$), with different trends observed for different
collectors from the same plot (Fig. 6). This high small-scale variability in soil solution DOC
makes it difficult to draw conclusions about long-term DOC trends from individual site
measurements, particularly in plots with heterogeneous soil and site conditions (Löfgren et
al., 2010).
The trends in soil solution DOC also varied across soil depth intervals. The mixed-effect
models suggested an increasing trend in soil solution DOC concentration in the organic layer,
and an increasing trend in soil solution DOC concentration under 80 cm depth when the
entire dataset (with breakpoints) was analyzed. The individual trend analyses seemed to
confirm the increasing trend under the organic layer (Table 1), while more heterogeneous
trends in the mineral soil were found, which is in line with previous findings (Borken et al.,
2011; Evans et al., 2012; Hruška et al., 2009; Löfgren and Zetterberg, 2011; Vanguelova et
al., 2010). This difference has been attributed to different processes affecting DOC in the
organic and shallow soils and in the subsoil. External factors such as acid deposition may
have a more direct effect in the organic layer where interaction between DOC and mineral
phases is less important compared to deeper layers of the mineral soil (Fröberg et al., 2006).
However, DOC measurements are not available for all depths at each site, complicating the
comparison of trends across soil depth intervals. Hence, the depth-effect on trends in soil
solution DOC cannot be consistently addressed within this study (see Supplementary material
S3).
Finally, the direction of the trends in soil solution DOC concentrations did not follow a clear
regional pattern across Europe (Fig. 4 and 5) and even contrasted with other soil solution
parameters that showed widespread trends over Europe, such as decreasing $SO_4^{2-}$ and
increasing pH. This finding indicates that effects of environmental controls on soil solution
DOC concentrations may differ depending on local factors like soil type (e.g., soil acidity) as
well as site and stand characteristics (e.g., tree growth or acidification history). Thus, the
trends in DOC in soil solution appear to be an outcome of interactions between controls
acting at local and regional scales.



### 4.2 Controls on soil solution DOC temporal trends

#### 4.2.1 Vegetation

Biological controls on DOC production and consumption, like stem growth, operating at site or catchment level, are particularly important when studying soil solution as plant-derived carbon is the main source of DOC (Harrison et al., 2008). Stem growth was available only for 53 sites as the increment between inventories carried out every five years, and as such no annual growth estimates were available. Nevertheless, our results suggest that vegetation growth is an important driver of DOC temporal dynamics in forests, as reported for peatlands (Billett et al., 2010; Dinsmore et al., 2013). Differences in DOC temporal trends across all soil depths were not related to forest type but to stem growth: more fertile plots, as indicated by higher stem volume increment, exhibited more often decreasing trends in DOC (Fig. 7 and 11), possibly in response to reduced C allocation to belowground nutrient acquisition system (Vicca et al., 2012).

It is well-established that N-enrichment favors the above-ground tissue production (as indicated by a higher stem volume increment) in forests (Janssens et al., 2010; Vicca et al., 2012) at the expense of C allocation to the root system, hence, reducing an important source of belowground DOC. On the other hand, forests with higher production would also have higher aboveground litterfall (Hansen et al., 2009), providing a higher input of labile carbon as a source for DOC leaching. Nevertheless, fertile forests may exhibit a higher microbial use efficiency, which may lead to proportionally more DOC being consumed, i.e., less DOC remaining in soil solution (Manzoni et al., 2012). Also, compared to vigorously growing forests with dense canopies, slower forest growth with less dense canopies have less interception and higher soil water input, which could stimulate litter decomposition and thus DOC production. Finally, forest growth might indirectly affect DOC trends through changes in soil solution chemistry (via cation uptake) (Vanguelova et al., 2007), but our data did not allow to test these pathways and thus the DOC response to vegetation uptake remains hypothetical.

#### 4.2.2 Acidifying deposition

Decreased atmospheric $SO_4^{2-}$ deposition and accumulation of atmospherically deposited N were hypothesized to increase DOC in European surface waters over the last 20 years (Evans et al., 2005; Hruška et al., 2009; Monteith et al., 2007). Sulphate and inorganic N deposition





decreased in Europe over the past decades (Waldner et al., 2014) but trends in soil solution
DOC concentrations varied largely, with increases, decreases, as well as steady states being
observed across respectively 56, 41 and 77 time series in European forests (Fig. 4 and 5).
Although we could not demonstrate a direct effect of trends in $SO_4^{2-}$ and inorganic N
deposition on the trends of soil solution DOC concentration, we observed a switch in the
direction of the DOC trends according to the mean $SO_4^{2-}$ and inorganic N deposition levels
(Fig. 7 and 8), with increasing soil solution DOC trends occurring more often in plots with
high N and, to a lesser extent, $SO_4^{2-}$ deposition. This suggests an interaction between the
deposition load and the mechanisms underlying the temporal change of soil solution DOC.

**Inorganic nitrogen**

Our results suggest that at sites with lower N deposition and lower soil $NO_3^-$, DOC
concentration in the soil solution is predominantly decreasing (Fig. 8A and 10) and in these
forests, we showed that decreasing trends in $SO_4^{2-}$ deposition coincided with increasing
trends in soil solution DOC (Fig. S3). The role of atmospheric N deposition in increasing
DOC leaching from soils has been well documented (Bragazza et al., 2006; Pregitzer et al.,
2004; Rosemond et al., 2015). The mechanisms behind this relationship are either physico-
chemical or biological. Chemical changes in soil solution through the increase of $NO_3^-$ ions
can trigger desorption of DOC (Pregitzer et al., 2004), and biotic forest responses to N
deposition, namely, enhanced photosynthesis, altered carbon allocation, and reduced soil
microbial activity (Bragazza et al., 2006; de Vries et al., 2009; Janssens et al., 2010), can
affect the final amount of DOC in the soil. One proposed mechanism is incomplete lignin
degradation and greater production of DOC in response to increased soil $NH_4^+$ (Pregitzer et
al., 2004; Zech et al., 1994). Alternatively, N-induced reductions of forest heterotrophic
respiration (Janssens et al., 2010) may lead to greater accumulation of DOC.

**Sulphate**

Similar to our observation for soil solution DOC, decreasing $SO_4^{2-}$ deposition has been linked
to increasing surface water DOC (Evans et al., 2006; Monteith et al., 2007; Oulehle and
Hruska, 2009). Sulphate deposition triggers soil acidification and a subsequent release of $Al^{3+}$
in acid soils. The amount of $Al^{3+}$ is negatively related to soil solution DOC due to two
plausible mechanisms: 1) The released $Al^{3+}$ can build complexes with organic molecules,
enhancing DOC precipitation and, in turn, suppress DOC solubility, therefore decreasing
DOC concentrations in soil solution (de Wit et al., 2001; Tipping and Woof, 1991;



Vanguelova et al., 2010), and 2) at higher levels of soil solution $Al^{3+}$ in combination with low
pH, DOC production through SOM decomposition decreases due to toxicity of $Al^{3+}$ to soil
organisms (Mulder et al., 2001). Consequently, when $SO_4^{2-}$ deposition is lower, increases of
soil solution DOC concentration could be expected (Fig. 11A, B). Finally, an indirect effect
of plant response to nutrient-limited acidified soil could also contribute to the trend in soil
solution DOC by changes to plant belowground C allocation (Vicca et al., 2012) (see Sect.

572   4.2.1.).

The $SO_4^{2-}$ deposition effect on the trends of DOC in soil solution depended on the soil acidity
(Fig. 9). Moreover, the soil chemical characteristics, more specifically the soil solution
conductivity (which is an indirect measure of ionic strength (Griffin and Jurinak, 1973)), and
the soil solution $NO_3^-$ and $SO_4^{2-}$ concentrations, were the most important factors determining
whether DOC concentrations increased or decreased over time (Fig. 10).
Ultimately, internal soil processes control the final concentration of DOC in the soil solution.
The solubility and biological production and consumption of DOC are regulated by pH, ionic
strength of the soil solution and the presence of $Al^{3+}$ and Fe (Bolan et al., 2011; De Wit et al.,
2007; Schwesig et al., 2003). These conditions are modulated by changes in atmospheric
deposition but not uniformly across sites: soils differ in acid-buffering capacity (Tian and
Niu, 2015), and the response of DOC concentrations to changes in $SO_4^{2-}$ deposition will thus
be a function of the initial soil acidification and buffer range (Fig. 9 and 11). Finally,
modifications of soil properties induced by changes in atmospheric deposition are probably
an order of magnitude lower than the spatial variation of these soil properties across sites,
making it difficult to isolate controlling factors on the final observed response of soil solution
DOC at continental scale (Clark et al., 2010).
In conclusion, the response of DOC to changes in atmospheric deposition seems to be
controlled by the past and present N deposition loads and acidification of soils (Clark et al.,
2010; Evans et al., 2012; Tian and Niu, 2015). It suggests that the mechanisms of recovery
from $SO_4^{2-}$ deposition and acidification take place only in non-N-saturated forests, as it has
been observed for N deposition effects (de Vries et al., 2009). In high N deposition areas, it is
likely that impacts of N-induced acidification on forest health and soil condition lead to more
DOC leaching, even though $SO_4^{2-}$ deposition has been decreasing. Therefore, soil solution
DOC concentrations responded as expected to changes in acid deposition, particularly in non
N-saturated sites but the hypothesis of recovery from acidity cannot fully explain overall
trends in Europe, as was also previously suggested in local or national studies of long-term





trends in soil solution DOC (Löfgren et al., 2010; Stutter et al., 2011; Ukonmaanaho et al.,
2014; Verstraeten et al., 2014).
Finally, our results confirm the long-term monotonic trends of DOC in soil solution as a
consequence of the interactions between local (soil properties, forest growth), and regional
(atmospheric deposition) controls acting at different temporal scales. However, further work
is needed to quantify the role of each mechanism underlying the final response of soil
solution DOC to environmental controls. We recommend that particular attention should be
paid to the biological controls (e.g., net primary production, stem growth, root exudates or
litterfall and canopy infestations) on long-term trends in soil solution DOC, which remains
poorly understood.

### 609    4.3    Link between DOC trends in soil and streams

An underlying question is how DOC trends in soil solution relate to DOC trends in stream
waters. Several studies have pointed out recovery from acidification as a cause for  increasing
trends in DOC concentrations in surface waters (Dawson et al., 2009; Evans et al., 2012;
Monteith et al., 2007; Skjelkvåle et al., 2003). Overall, our results point to a noticeable
increasing trend in DOC in the organic layer of forest soils, which is qualitatively consistent
with the increasing trends found in stream waters and in line with positive DOC trends
reported for the soil organic layer or at maximum 10 cm depth of the mineral soil in Europe
(Borken et al., 2011; Hruška et al., 2009; Vanguelova et al., 2010). On the other hand, while
there was also evidence of increasing trends in the deep mineral horizon (> 80 cm), trends at
different soil horizons along the mineral soil were more variable and responded to other soil
internal processes.
Hence, the results from the trend analysis for the overall European dataset points out to a link
between the long-term dynamics in surface and deep soil and surface water DOC. However,
the individual trend analysis reflects a high heterogeneity in the long-term response of soil
DOC to environmental controls. In fact, it is currently difficult to link long-term dynamics in
soil and surface water DOC. Large scale processes become more important than local factors
when looking at DOC trends in surface waters (Lepistö et al., 2014), while the opposite
seems to apply for soil solution DOC trends. Furthermore, stream water DOC mainly reflects
the processes occurring in areas with a high hydraulic connectivity in the catchment, such as
peat soils or floodplains, which normally yield most of the DOC (Löfgren and Zetterberg,
2011). Further monitoring studies in forest soils with high hydraulic connectivity to streams



are needed to be able to link dynamics of DOC in forest soil with dynamics of DOC in stream
waters.

## 5    Conclusions

Different monotonic long-term trends of soil solution DOC have been found across European
forests at plot scale, with the majority of the trends for specific plots and depths not being
statistically significant (40%), followed by significantly positive (35%) and significantly
negative trends (25%). The distribution of the trends did not follow a specific regional
pattern. There was evidence that an overall increasing trend occurred in the organic layers
and, to a lesser extent, in the deep mineral soil, however, there is less agreement on the trends
found in different soil horizons along the mineral soils.
A multivariate analysis revealed a negative relation between long-term trends in soil solution
DOC and mean $SO_4^{2-}$ deposition and a positive relation to mean $NO_3^-$ deposition. While the
hypothesis of increasing trends of DOC due to reductions of $SO_4^{2-}$ deposition could be
confirmed in more N-limited forests, there was no significant relationship with $SO_4^{2-}$
deposition in more N-enriched forests. We found evidence that soil pH determines the
response of trends of DOC in soil solution to $SO_4^{2-}$ deposition, indicating that internal soil
processes control the final response of DOC in soil solution. Although correlative, our results
suggest that there is no single mechanism responsible for soil solution DOC trends operating
at large scale across Europe but that interactions between controls operating at local (soil
properties, site and stand characteristics) and regional (atmospheric deposition changes)
scales are taking place at the same time.

### Acknowledgements

We want to thank the numerous scientists and technicians who were involved in the data
collection, analysis, transmission, and validation of the ICP Forests Monitoring Program
across all the European countries from which data have been used in this work. The
evaluation was mainly based on data that are part of the UNECE ICP Forests PCC
Collaborative Database (see www.icp-forests.net) or national Databases (e.g., The service
and research facility of the Brandenburg forestry state agency, Eberswalde
*(Landeskompetenzzentrum Forst Eberswalde, LFE)* for parts of the data for Germany). For
soil, we used and acknowledge the aggregated forest soil condition database
(AFSCDB.LII.2.1) compiled by the ICP Forests Forest Soil Coordinating Centre. The long-
term collection of forest monitoring data was to a large extent funded by national research



institutions and ministries, with support from further bodies, services and land owners. It was
partially funded by the European Union under the Regulation (EC) No. 2152/2003
concerning monitoring of forests and environmental interactions in the Community (Forest
Focus) and the project LIFE 07 ENV/D/000218. SV is a postdoctoral research associate of
the Fund for Scientific Research – Flanders. IAJ, JP and PC acknowledge support from the
European Research Council Synergy grant ERC-2013-SyG-610028 IMBALANCE-P.
Finally, we acknowledge the E-OBS dataset from the EU-FP6 project ENSEMBLES
(http://ensembles-eu.metoffice.com) and the data providers in the ECA&D project
(http://www.ecad.eu).

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





Table 1. Temporal trends of DOC concentrations obtained with the linear mixed models (LMM) built for different forest types, soil depth intervals and collector types with the entire dataset (with breakpoints) and with the dataset without time series showing breakpoints (without breakpoints) and the Seasonal Mann Kendal tests (SMK). The table shows the relative slope (rslope in % yr-1), the number of observations (n) and the p value. For the SMK tests, the number of time series showing significant negative (N), non-significant (NS) and significant positive (P) trends are shown. LMMs for which no statistically significant trend was detected (p>0.1) are represented in grey and the LMMs for which a significant trend (p<0.05) was detected are in bold. (O: organic layer, M02: mineral soil 0-20 cm, M24: mineral soil 20-40 cm, M48: mineral soil 40-80 cm, M8: mineral soil > 80 cm/ TL: tension lysimeter, ZTL: zero-tension lysimeter/ n.s.: no significant.)

In broadleaved and coniferous forests:

| Collector type | Layer | LMM (with breakpoints) | | | LMM (without breakpoints) | | | SMK (without breakpoints) | | | |
|---|---|---|---|---|---|---|---|---|---|---|---|
| | | n | rslope | p value | n | rslope | p value | rslope | N | NS | P |
| TL | O | 3133 | 6.75 | *0.0782* | 1168 | -0.30 | n.s. | -1.03 (±1.65) | 1 | 3 | 1 |
| | M02 | 19311 | 0.10 | n.s. | 8917 | -1.06 | n.s. | 0.16 (±4.78) | 17 | 29 | 21 |
| | M24 | 7700 | 2.69 | n.s. | 3404 | 3.66 | n.s. | 0.6 (±9.03) | 11 | 12 | 11 |
| | M48 | 24614 | 0.95 | n.s. | 11065 | 0.80 | n.s. | 0.67 (±4.76) | 22 | 30 | 32 |
| | M8 | 9378 | **6.78** | **0.0036** | 3394 | 3.41 | n.s. | 1.007 (±8.79) | 8 | 9 | 16 |
| ZTL | O | 8136 | **3.75** | **<0.001** | 4659 | 1.63 | *0.0939* | **1.7 (±4.28)** | 3 | 16 | 8 |
| | M02 | 3389 | -0.54 | n.s. | 445 | 0.17 | n.s. | -0.7 (±1.85) | | 3 | 1 |
| | M24 | 739 | 0.36 | n.s. | | | | | | | |





| | | | | | | | | | | |
|---|---|---|---|---|---|---|---|---|---|---|
| M48 | 654 | -3.37 | n.s. | 336 | 1.05 | n.s. | 1.07 (±3.08) | 1 | 2 | 1 |
| M8 | 118 | 1.39 | n.s. | | | | | | | |

In broadleaved forests:

| Collector type | Layer | LMM (with breakpoints) | | | LMM (without breakpoints) | | | SMK (without breakpoints) | | | |
|---|---|---|---|---|---|---|---|---|---|---|---|
| | | n | rslope | p value | n | rslope | p value | rslope | N | NS | P |
| TL | O | 637 | -5.96 | n.s. | 475 | -0.17 | n.s. | -0.3 (±0.9) | 0 | 2 | 0 |
| | M02 | 8397 | 3.07 | *0.0764* | 3104 | 0.51 | n.s. | 0.89 (±5.94) | 4 | 7 | 10 |
| | M24 | 2584 | -0.05 | n.s. | 928 | 6.01 | n.s. | 1.03 (±11.31) | 3 | 5 | 4 |
| | M48 | 10635 | -0.93 | n.s. | 4634 | 2.46 | n.s. | 1.51 (±5.31) | 11 | 8 | 16 |
| | M8 | 4354 | -6.85 | *0.0672* | 1797 | -0.10 | n.s. | 0.3 (±6.28) | 4 | 5 | 6 |
| ZTL | O | 4057 | 0.37 | n.s. | 1956 | -0.90 | n.s. | 0.96 (±5.47) | 2 | 7 | 3 |
| | M02 | 608 | 0.26 | n.s. | 192 | 1.88 | n.s. | 2.72 | | | 1 |
| | M24 | 94 | **11.80** | **0.026** | | | | | | | |
| | M48 | 427 | -2.84 | n.s. | | | | 0 | | 1 | |
| | M8 | 34 | **-36.18** | **<0.001** | | | | | | | |

In coniferous forests:

| Collector type | Layer | LMM (with breakpoints) | | | LMM (without breakpoints) | | | SMK (without breakpoints) | | | |
|---|---|---|---|---|---|---|---|---|---|---|---|
| | | n | rslope | p value | n | rslope | p value | rslope | N | NS | P |
| TL | O | 2496 | 8.15 | *0.0633* | 693 | 1.33 | n.s. | -1.06 | 1 | 1 | 1 |





|     |      |       |        |        |      |       |        |                   |    |    |    |
|-----|------|-------|--------|--------|------|-------|--------|-------------------|----|----|----|
|     |      |       |        |        |      |       |        | (±2.25)           |    |    |    |
|     | M02  | 10914 | -0.97  | n.s.   | 5813 | -1.60 | n.s.   | -0.04 (±3.98)     | 13 | 22 | 11 |
|     | M24  | 5116  | 2.71   | n.s.   | 2476 | 3.66  | n.s.   | -0.3 (±7.82)      | 7  | 7  | 8  |
|     | M48  | 13979 | 1.24   | n.s.   | 6431 | 0.05  | n.s.   | 0.3 (±4.32)       | 16 | 22 | 11 |
|     | M8   | 5024  | **9.93** | **<0.001** | 1597 | 7.58  | n.s.   | 2.89 (±10.28)     | 4  | 4  | 10 |
| ZTL | O    | 4079  | **3.59** | **0.0018** | 2703 | **3.09** | **0.0045** | **1.85 (±2.88)** | 1  | 9  | 5  |
|     | M02  | 2781  | -0.60  | n.s.   | 253  | -1.44 | n.s.   | -0.83 (±0.4)      | 0  | 3  | 0  |
|     | M24  | 645   | 0.23   | n.s.   |      |       |        |                   |    |    |    |
|     | M48  | 227   | -0.39  | n.s.   | 251  | -0.55 | n.s.   | 2.14 (±3.66)      | 1  | 1  | 1  |
|     | M8   | 84    | 13.87  | *0.0995* |      |       |        |                   |    |    |    |





Table 2. Median relative trend (rslope in % yr$^{-1}$) of DOC concentrations and interquartile range of rslope and number of time series with statistically significant ($p < 0.05$) positive (P) and negative (N) trends and with non-significant (NS) trends of DOC using the seasonal Mann-Kendall test (SMK), the partial Mann-Kendall test (PMK) and the Breaks For Additive Seasonal and Trend test (BFAST). (O: organic layer, M02: mineral soil 0-20 cm, M24: mineral soil 20-40 cm, M48: mineral soil 40-80 cm, M8: mineral soil > 80 cm.)

| Soil depth | SMK | | | | PMK | | | | BFAST | | | |
|---|---|---|---|---|---|---|---|---|---|---|---|---|
| | rslope | N | NS | P | rslope | N | NS | P | rslope | N | NS | P |
| O | 1.18 (±3.37) | 4 | 19 | 9 | 1.0 (±3.44) | 4 | 18 | 9 | 1.15 (±3.47) | 5 | 18 | 9 |
| M02 | 0.04 (±3.41) | 17 | 32 | 22 | 0.10 (±3.29) | 16 | 33 | 21 | -0.40 (±3.56) | 19 | 34 | 18 |
| M24 | 0.61 (±8.62) | 11 | 12 | 11 | -0.03 (±8.97) | 10 | 11 | 11 | 0.83 (±9.31) | 10 | 11 | 13 |
| M48 | 1.01 (±4.79) | 23 | 32 | 33 | 0.77 (±4.75) | 22 | 31 | 33 | 0.59 (±6.32) | 23 | 33 | 32 |
| M8 | 1.18 (±9.39) | 8 | 9 | 16 | 1.01 (±8.48) | 8 | 11 | 14 | 1.75 (±9.59) | 7 | 9 | 17 |



Table 3. Site properties for the 13 plots showing consistent negative trends (N) of DOC concentrations and for the 12 plots showing consistent positive trends (P) of DOC concentrations. Soil properties (clay percentage, C/N ratio, pH(CaCl$_2$), cation exchange capacity (CEC)) are for the soil depth interval 0-20 cm. Mean atmospheric deposition (inorganic N and SO$_4^{2-}$) is throughfall deposition. When throughfall deposition was not available, bulk deposition is presented with an asterisk. Trends in soil solution pH, Ca$^{2+}$ and Mg$^{2+}$ concentrations were calculated using the seasonal Mann-Kendall test.

| Code Plot | Trend | Soil Type (WRB) | Clay (%) | C/N | pH | CEC (cmol$_+$kg$^{-1}$) | MAP (mm) | MAT (°C) | N depos. (kg N ha$^{-1}$ yr$^{-1}$) | SO$_4^{2-}$ deposition (kg S ha$^{-1}$ yr$^{-1}$) | slope pH (%yr$^{-1}$) | slope Ca$^{2+}$ (% yr$^{-1}$) | slope Mg$^{2+}$ (% yr$^{-1}$) |
|---|---|---|---|---|---|---|---|---|---|---|---|---|---|
| France (code = 1) | | | | | | | | | | | | | |
| 30 | N | Cambic Podzol | 3.79 | 16.8 | 3.96 | 1.55 | 567 | 11.9 | 7.28 | 4.25 | 0.10 | -0.90 | -1.00 |
| 41 | N | Mollic Andosol | 23.9 | 16.6 | 4.23 | 7.47 | 842 | 10.6 | 4.43 | 4.15 | 0.00 | -1.10 | -1.30 |
| 84 | N | Cambic Podzol | 4.09 | 22.8 | 3.39 | 4.07 | 774 | 10.5 | 7.66 | 3.77* | 0.50 | 2.00 | 1.00 |
| Belgium (code =2) | | | | | | | | | | | | | |
| 11 | P | Dystric Cambisol | 3.54 | 17.7 | 2.81 | 6.22 | 805 | 11.0 | 18.7 | 13.2 | 0.40 | -11.0 | -8.00 |
| 21 | P | Dystric Podzo-luvisol | 11.2 | 15.4 | 3.59 | 2.41 | 804 | 10.3 | 16.8 | 13.2 | 0.00 | -9.00 | -5.00 |
| Germany (code:= 4) | | | | | | | | | | | | | |
| 303 | N | Haplic Podzol | 17.3 | 16.5 | 3.05 | 8.77 | 1180 | 9.10 | 17.5 | | 0.40 | -5.00 | -2.00 |
| 304 | N | Dystric Cambisol | 21.3 | 17.7 | 3.63 | 6.14 | 1110 | 6.20 | 16.4 | | 0.00 | -3.00 | -0.40 |
| 308 | N | Albic | 3.80 | 16.5 | 3.41 | 1.63 | 816 | 9.20 | 14.2* | | 0.00 | -5.00 | -2.00 |





| Code Plot | Trend | Soil Type (WRB) | Clay (%) | C/N | pH | CEC (cmol$_+$ kg$^{-1}$) | MAP (mm) | MAT (°C) | N depos. (kg N ha$^{-1}$ yr$^{-1}$) | SO$_4^{2-}$ deposition (kg S ha$^{-1}$ yr$^{-1}$) | slope pH (% yr$^{-1}$) | slope Ca$^{2+}$ (% yr$^{-1}$) | slope Mg$^{2+}$ (% yr$^{-1}$) |
|---|---|---|---|---|---|---|---|---|---|---|---|---|---|
| 802 | N | Arenosol Cambic Podzol | 6.00 | 25.7 | 3.35 | 4.33 | 836 | 11.9 | 25.2 | 13.2 | 0.50 | -2.40 | -1.50 |
| 1502 | N | Haplic Arenosol | 4.40 | 23.8 | 3.78 | 2.35 | 593 | 9.40 | 9.79 | 5.66 | | -16.0 | -14.0 |
| 306 | P | Haplic Calcisol | | | | | 782 | 10.2 | 13.9 | | 0.50 | 2.00 | 2.00 |
| 707 | P | Dystric Cambisol | | | | | 704 | 10.7 | 18.3 | 8.49 | 0.00 | -10.0 | -2.00 |
| 806 | P | Dystric Cambisol | | | | | 1349 | 8.30 | 23.0 | 6.81 | 0.30 | -7.00 | -6.00 |
| 903 | P | Dystric Cambisol | | | | | 905 | 9.60 | | | 0.20 | -5.00 | -3.00 |
| 920 | P | Dystric Cambisol | | | | | 908 | 8.90 | | | -1.00 | -6.00 | -0.50 |
| 1402 | P | Haplic Podzol | 8.65 | 26.2 | 3.24 | 9.04 | 805 | 6.90 | 13.5 | 24.3 | 1.20 | -6.00 | 9.00 |
| 1406 | P | Eutric Gleysol | 15.9 | 23.1 | 3.59 | 6.67 | 670 | 8.80 | 15.3 | 6.23 | 1.11 | -4.00 | -3.00 |
| Italy (code = 5) | | | | | | | | | | | | | |
| 1 | N | Humic Acrisol | 3.14 | 12.2 | 5.32 | 31.6 | 670 | 23.3 | | | -0.30 | -10.0 | -10.0 |
| United Kingdom (code = 6) | | | | | | | | | | | | | |
| 922 | P | Umbric Gleysol | 34.8 | 15.6 | 3.31 | 10.8 | 1355 | 9.50 | | | 0.40 | -9.00 | 2.00 |
| Austria (code = 14) | | | | | | | | | | | | | |





| Code Plot | Trend | Soil Type (WRB) | Clay (%) | C/N | pH | CEC ($cmol_+ kg^{-1}$) | MAP (mm) | MAT (°C) | N depos. ($kg\ N\ ha^{-1}\ yr^{-1}$) | $SO_4^{2-}$ deposition ($kg\ S\ ha^{-1}\ yr^{-1}$) | slope pH (%$yr^{-1}$) | slope $Ca^{2+}$ (% $yr^{-1}$) | slope $Mg^{2+}$ (% $yr^{-1}$) |
|---|---|---|---|---|---|---|---|---|---|---|---|---|---|
| 9 | N | Eutric Cambisol | 20.1 | 12.8 | 5.26 | 25.9 | 679 | 10.8 | | 3.80* | 0.40 | -1.50 | -0.60 |
| Switzerland (code = 50) | | | | | | | | | | | | | |
| 15 | N | Dystric Planosol | 17.6 | 14.7 | 3.73 | 7.76 | 1201 | 8.90 | 15.1 | 4.67 | -0.10 | -13.0 | -4.00 |
| 2 | P | Haplic Podzol | 14.7 | 18.3 | 3.17 | 3.59 | 1473 | 4.40 | | | -0.80 | -5.00 | -3.00 |
| Norway (code =55) | | | | | | | | | | | | | |
| 14 | N | Cambic Arenosol | 9.83 | 25.4 | 3.46 | | | | 14.7 | 21.9 | 0.10 | -1.70 | -3.30 |
| 19 | N | | 10.5 | 18.7 | 3.79 | | 836 | 4.60 | 1.54 | 2.61 | 0.50 | -7.00 | -4.00 |
| 18 | P | | 3.05 | 29.5 | 3.69 | | 1175 | 0.35 | | 2.40 | -0.90 | 0.00 | 0.00 |





Table 4. Statistics (Wilks' Lambda and p value) of the General Discriminant Analysis among groups of plot-soil depth combinations with different trend in DOC during the last years conducted with 10 different soil solution and deposition variables as independent continuous variables and soil depth as categorical independent variable. Bold type indicates a significant effect of the variable in the model ($p < 0.05$)

| Independent variables | Wilks' Lambda | *p* value |
|---|---|---|
| **pH** | 0.913 | 0.158 |
| **log(NH$_4^+$_TF)** | 0.973 | 0.575 |
| **log(NO$_3^-$_BD)** | 0.944 | 0.308 |
| **log(SO$_4^{2-}$_BD)** | 0.920 | 0.182 |
| **log(SO$_4^{2-}$_SS)** | 0.857 | **0.042** |
| **log(NO$_3^-$_SS)** | 0.814 | **0.015** |
| **log(NH$_4^+$_SS)** | 0.947 | 0.331 |
| **log(AL_SS)** | 0.961 | 0.434 |
| **log(FE_SS)** | 0.930 | 0.224 |
| **log(CONDUCTIVITY_SS)** | 0.807 | **0.012** |





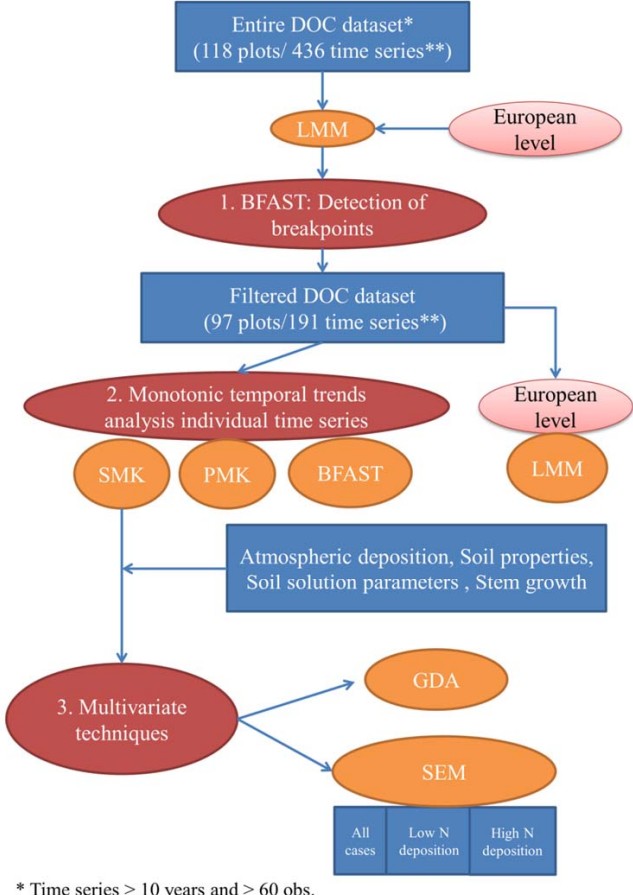

Figure 1. Flow-diagram of the sequence of methods applied for analysis of temporal trends of soil solution DOC and their drivers.



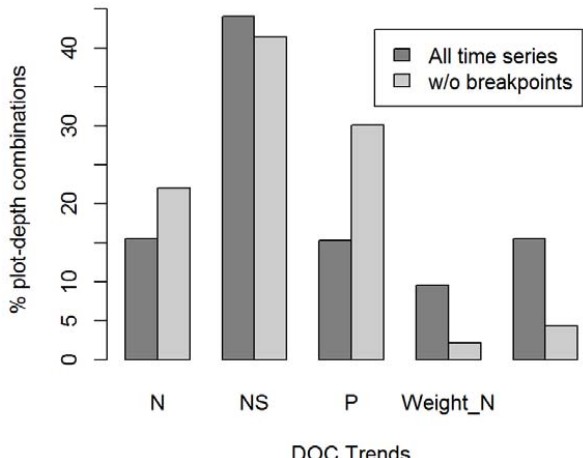

Figure 2. Percentage of plot-soil depth combinations for which negative (N), non-significant (NS), positive (P), negative and non-significant (Weight_N) and positive and non-significant (Weight_P) trends of DOC concentrations were found using SMK (seasonal Mann-Kendall) tests when 1) all the 436 time series were used, 2) only 191 time series without breakpoints (detected using the BFAST (*Breaks For Additive Seasonal and Trend*) analysis) were used.





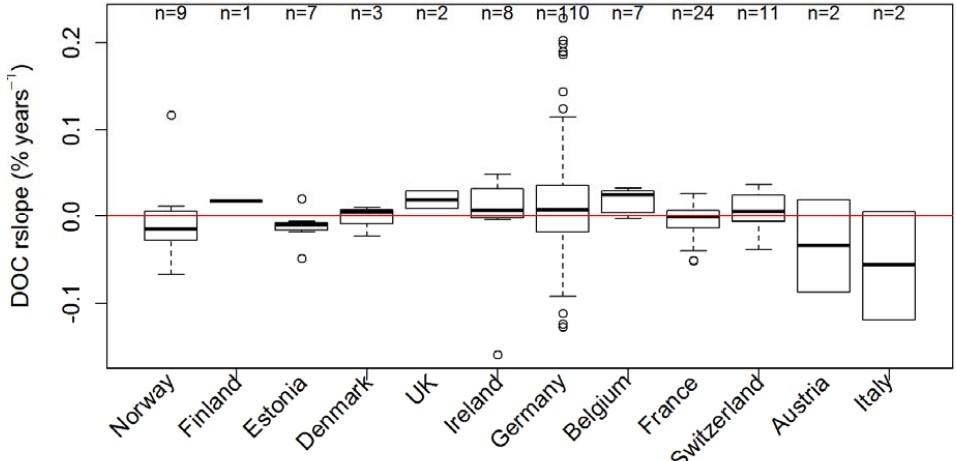

Figure 3. Relative trend slope of DOC trends calculated using the seasonal Mann-Kendall test (SMK) for time series with more than 10 years of measurements and no breakpoints in 12 European countries, ranked from north to south.





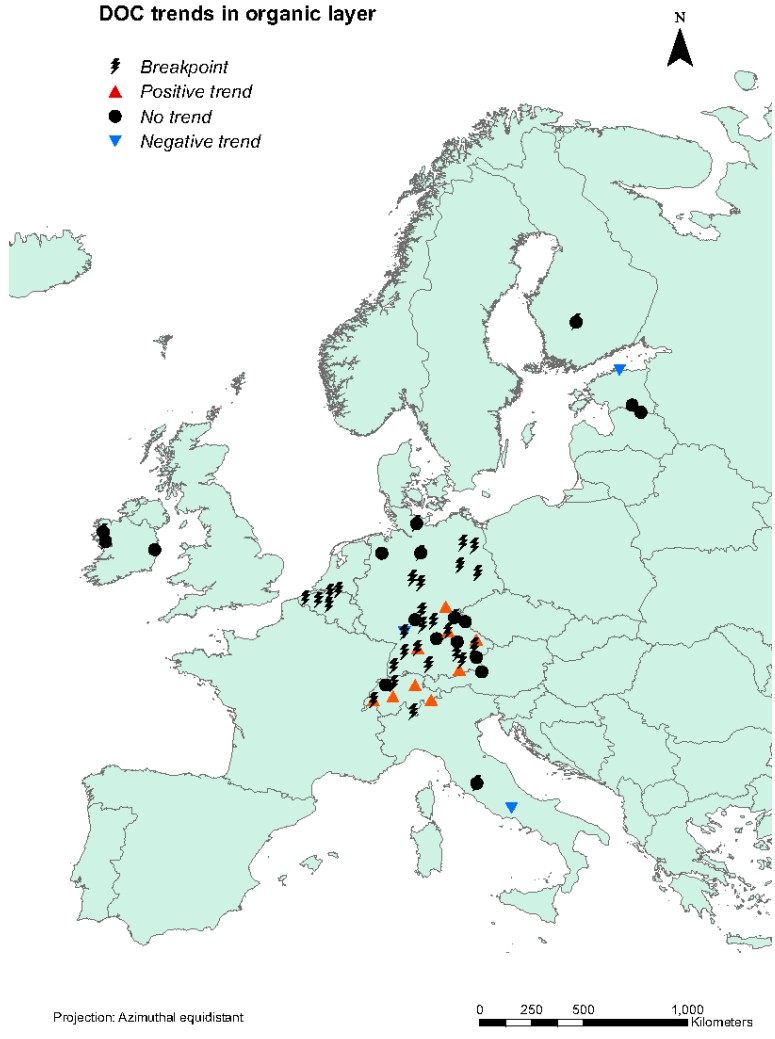

Figure 4. Directions of the temporal trends in soil solution DOC concentration in the organic layer at plot level. Trends were evaluated using the seasonal Mann-Kendall test. Data span from 1991 to 2011.





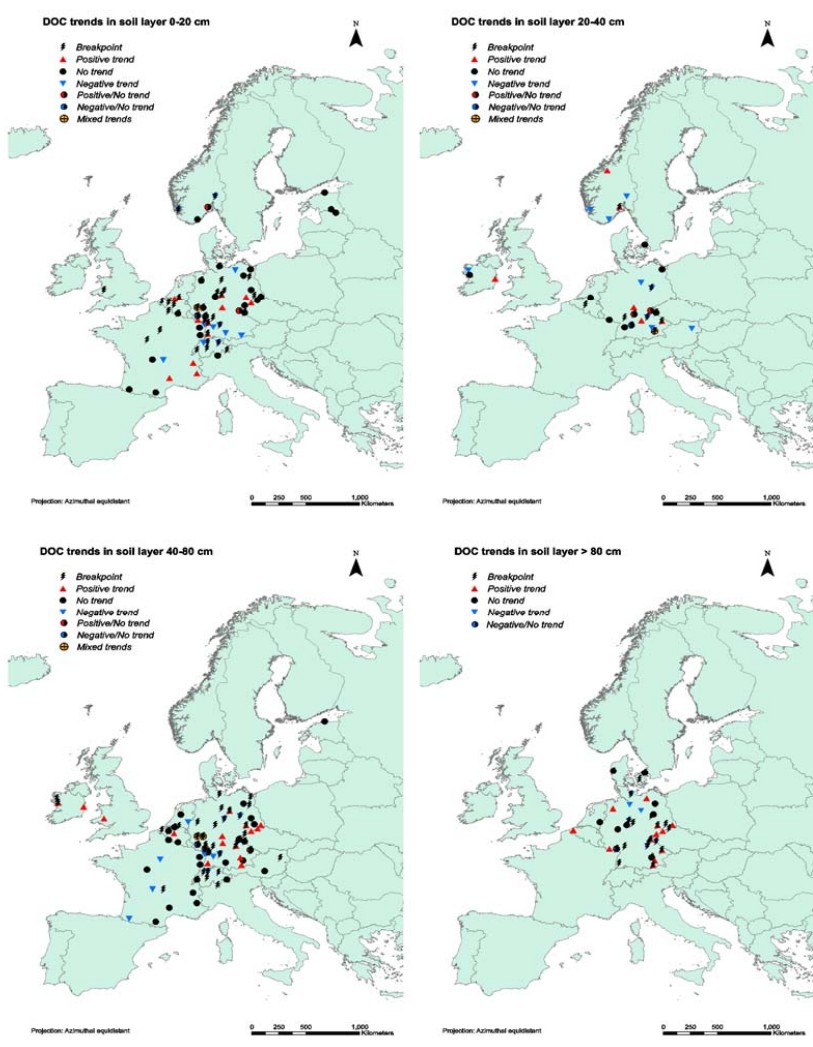

Figure 5. Directions of temporal trends in soil solution DOC concentration at plot level in the mineral soil for soil layers: a) topsoil (0–20 cm), b) intermediate (20–40 cm), c) subsoil (40–80 cm) and d) deep subsoil (> 80 cm). Trends were evaluated using the seasonal Mann-Kendall test. Data span from 1991 to 2011.



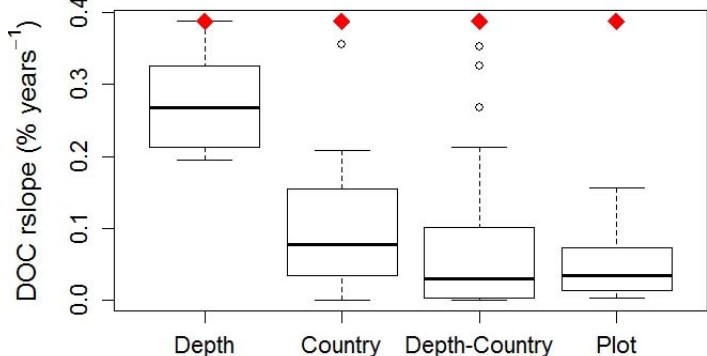

Figure 6. Range of relative trend slopes (max-min) for trends of DOC concentration in soil solution within each 1) depth interval, 2) country, 3) depth interval per country, and 4) plot. The boxplots show the median, 25% and 75% quantiles (box), minimum and 1.5 times the interquartile range (whiskers) and higher values (circles). The red diamond marks the maximum range of slopes in soil solution trends in the entire dataset.



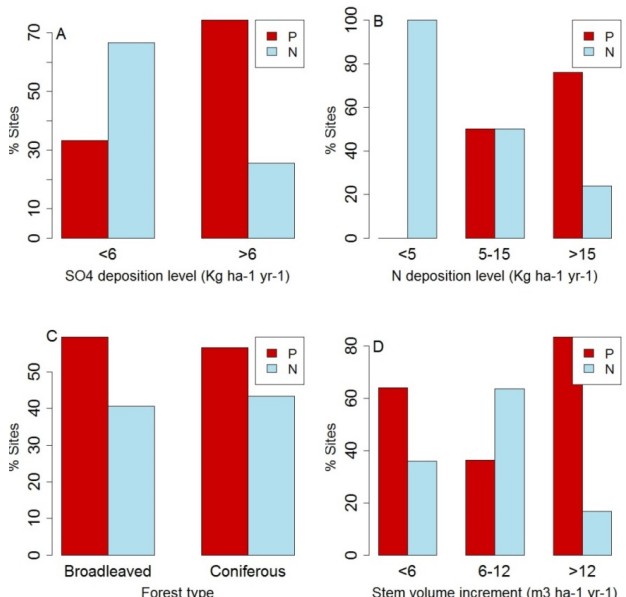

Figure 7. Percentage of occurrence of positive and negative trends in soil solution separated by A) throughfall $SO_4^{2-}$ deposition level (kg S ha$^{-1}$ yr$^{-1}$), B) throughfall inorganic N deposition level (kg N ha$^{-1}$ yr$^{-1}$), C) forest type and D) stem volume increment (m$^3$ ha$^{-1}$ yr$^{-1}$).



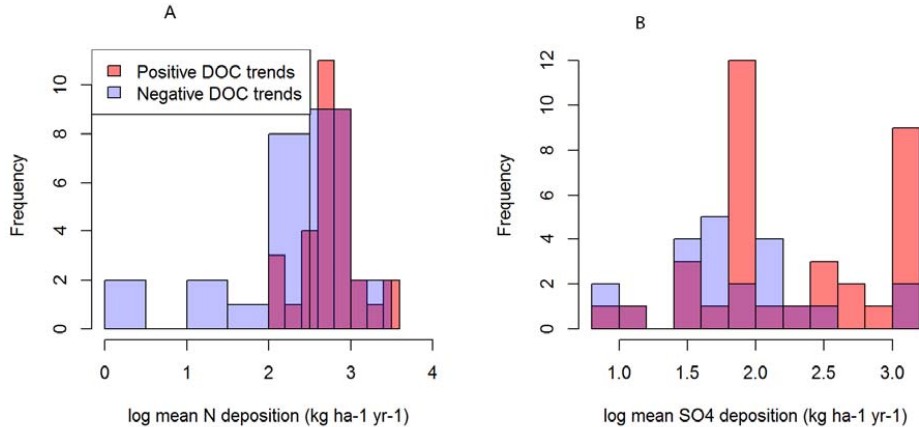

Figure 8. Histograms for natural log-transformed mean throughfall $SO_4^{2-}$ deposition (A) and for log-transformed mean throughfall inorganic N deposition (B) for positive and negative trends of DOC.



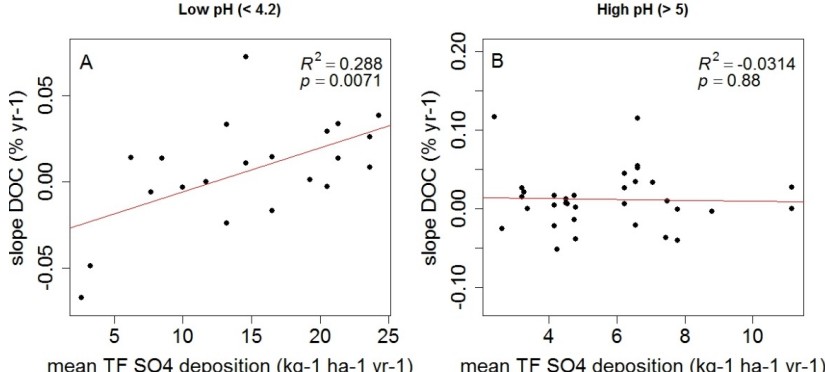

Figure 9. Relationship between mean throughfall $SO_4^{2-}$ deposition and relative slopes of DOC for very acid soils (pH in soil solution < 4.2) (left) and non-acid soils (pH in soil solution > 5) (right).





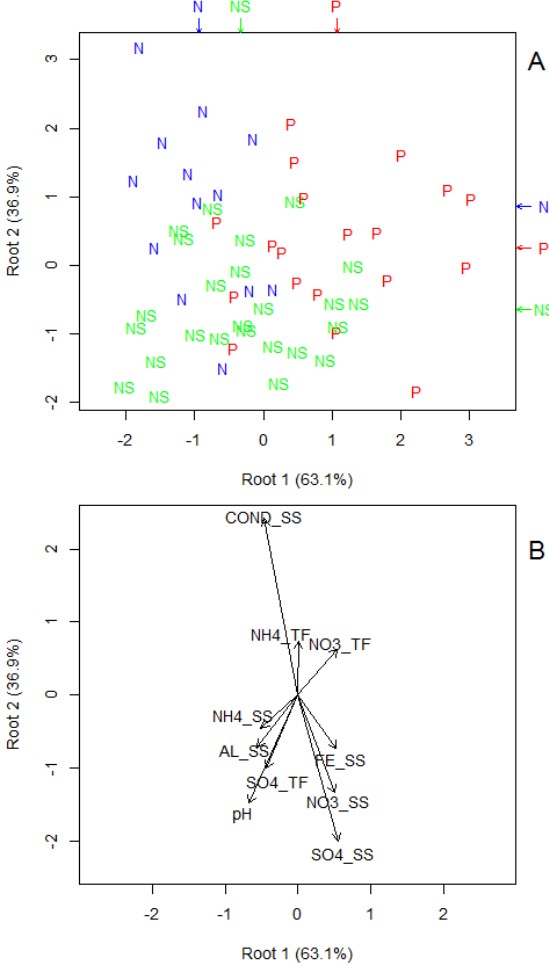

Figure 10. Biplot representing the scores for the single plot-soil depth combinations for the two roots of the General Discriminant Analysis (GDA). (B) Biplot representing the standardized canonical discriminate function coefficients for the two roots of this GDA. The GDA is generated to explain the variance among groups of plot-soil depth combinations with different trend in soil solution DOC (N for negative trends, P for positive trends and NS for non-significant trends) during the last years conducted with 7 soil solution variables (pH, NH4_SS, NO3_SS, FE_SS, SO4_SS, COND_SS, AL_SS) and three throughfall deposition variables (NH4_TF, NO3_TF, SO4_TF) as independent continuous variables and different soil layers as categorical independent variable.





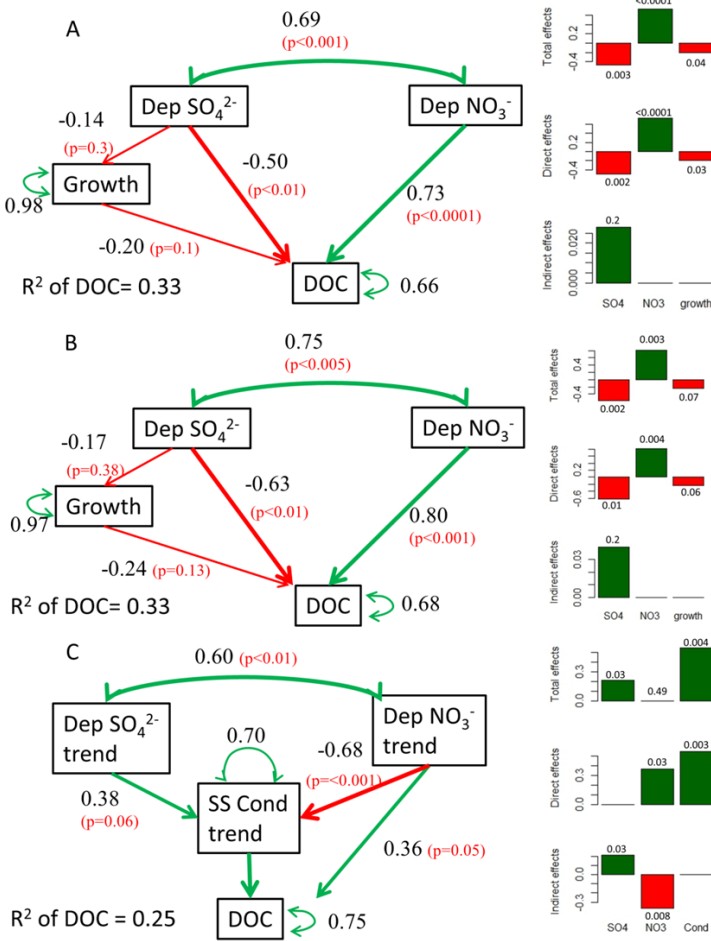

Figure 11. Diagrams of the structural equation models (SEM) that best explain the maximum variance of the resulting trends of DOC concentrations in soil solution for: A) all the cases , B) cases with low or medium throughfall inorganic N deposition (> 15 kg N ha$^{-1}$ yr$^{-1}$), and C) cases with high throughfall inorganic N deposition with mean or trends in annual $SO_4^{2-}$ and $NO_3^-$ deposition (kg N ha$^{-1}$ yr$^{-1}$) with direct and indirect effects through effects on soil solution parameters (trends of conductivity in µS/cm) and mean annual stem volume increment (growth) in m$^3$ ha$^{-1}$ yr$^{-1}$). p-values of the significance of the corresponding effect are between brackets. Green arrows indicate positive effects and red arrows indicate negative effects. Side bar graphs indicate the magnitude of the total, direct and indirect effects and their p-value.