# Peer review of "Trends in soil solution dissolved organic carbon (DOC) concentrations across European forests"

_Biogeosciences, 2015_

## Referee Comment (RC1) · Anonymous Referee #1 · 19 Feb 2016

Dear editor,

This manuscript deals with DOC trends in soil water at different soil horizons in European forests. The statistical analyzes were mainly performed on data from a subset of sites from the ICP Forests level II plots. Based on 436 soil water time series (number uncertain, see below) from 118 plots (Figure 1), DOC concentration trends at the European level were evaluated with linear mixed-effects models (LMM). A second analysis was performed after removing time series with breakpoints determined with the Breaks For Additive Seasonal and Trend (BFAST) algorithm. The remaining data set included 191 time series (number uncertain, see below) from 97 plots. Different statistical non-parametric methods including Seasonal Mann Kendal (SMK) and Partial

Mann Kendall (PMK) were used on each time series without breakpoints to evaluate monotonic trends and to test influence of precipitation as a co-variable, respectively. The same types of analyses were performed on other soil solution chemical parameters and on precipitation and temperature. Two multivariate statistical methods were used (General Discriminant Analysis [GDA] and Structural Equation Models [SEM]) to identify the main factors co-varying with the differences in DOC trends among the selected plots.

The presented work is of great scientific interest and well suitable for publishing in BG. The manuscript is well written, technically sound (see below for some comments) and well organized. However, the data presented are not allowing for comparisons with other soil water DOC investigations. There is a lack of quantitative data on real DOC concentrations and trends. Hence, the manuscript needs a major revision before it can be published in BG.

General comments

1. In the manuscript and supplement, there is no information on the soil solution chemistry at the studied plots. A quantitative description of the concentrations of DOC and other relevant water chemical parameters is missing. This information could be given in tables and figures in manuscript and supplementary materials, describing e.g. median values, 25- and 75-percentiles and number of observations or in boxplots. The information should be available separated on collector type and soil layer (cf. Table 1) for all classes used in the assessment (forest type, soil type, soil pH, N and S depositions).

2. The above information should be used for assessing how the standardized trends (rslope) are affected by the median concentrations (see comment 3) and for defining whether the statistically significant rslope trends in the filtered data (LMM, SMK and PMK) are found over the entire DOC concentration range or within certain intervals. Additionally, what does the statistically significant rslope trends correspond to in DOC concentration trends? Are they quantitatively large or not?

3. The trends are reported in standardized terms (rslope), which means that the slope (Sen slope) of each time series was divided by the median concentration over the studied period. This implies that the rslope-value can be identical regardless of the DOC concentration level. Hence, rslope will be 0.1 if you have a trend of 0.2 mg DOC yr-1 and a median DOC concentration of 2 mg l-1 at one plot-soil depth as well as if the trend is 5 mg DOC yr-1 at 50 mg DOC l-1 at another plot-soil depth. The significance of the latter example is of course much larger than in the former. Are e.g. the statistical trends in deep mineral soils (Table 1, layer M8) a result of this phenomenon?

4. Evaluating hundreds of time series may introduce random effects affecting the number of statistically significant trends. The theoretical and if possible quantitative implication of this (false positive and false negative trends) should be discussed.

5. Using relative data in multivariate statistical models like GDA, may cause biased results strongly exaggerating the effects of trends in low DOC-concentration soil horizons. A discussion on the latter is missing.

6. Throughout the manuscript, the information gained from comments 1-3 should be commented where relevant. The information is especially important for the results and discussions dealing with the directions and controls on soil solution temporal trends (Chapters 3.1 and 4.2) partly based on the GDA and SEM results. Are the indicated effects quantitatively important, do they occur both at low and high DOC concentrations in soil solution and has the DOC concentration level any influence on the trend strength and direction?

7. The title of chapter 4.1.1 as well as some of the text are obscure. The number of non-significant trends is determined by the data and the statistical methods used. The authors themselves have selected data after quality check and chosen the statistical methods including probabilities to accept or reject trends. By speculating on whether the non-significant trends are real or not, the authors seem to reject their own data and methods? Change title and remove these speculations, but keep the general discussions on factors affecting trend analysis including what you have found related to comment 4 (see above).

Detailed comments

1. Lines 71-81: Riparian zones and peat lands, the most important DOC sources for surface waters are not referred to. Add some text and references.

2. Line 205: "...more than 60 observations of soil solution DOC of individual or groups of collectors". What do the 60 observations refer to? Individual or groups of collectors? If the latter, was it pooled composite samples?

3. Line 209: In Figure 1, the number is 436 time series instead of 529. Which figure is the correct one?

4. Line 218: Did you use the same pH ranges for all soil horizons? If so, you may have a bias towards organic and upper mineral soil horizons in the Low pH class. Additionally it is not clear whether it is soil pH as stated in text or in soil solution as stated in Figure 9? Clarify!

5. Lines 219-222: From which time period do the S and N depositions originate? Is it median values or...?

6. Line 276: Add p-value to "...overall positive trend...". $p < 0.05$ or $p < 0.10$??

7. Lines 296-301 and Table 2. In the last sentence, it is stated that trends computed with SMK and PMK agreed well. However, at soil depth M24 the two methods results in very different rslopes (Table 2) both as regards directions and values. Comment on this and present a possible explanation.

8. Lines 309-311: nNS-trends=104, nP-trends=91 and nN-trends=63 makes up a total of 258 time series, which corresponds to the value in Supplementary materials. However, the number of monotonic trends is 191 according to Figure 1. Correct where appropriate.

9. Lines 324-332 and Table 2: There are increased rslope values towards deeper mineral soil horizons. Is this a result of lower soil solution DOC concentrations (cf. general comment 3) and thereby very small DOC trends in absolute numbers? The rslope values in the O-horizon, generally showing high DOC concentrations, are close to those found for M8, indicating large DOC trends if statistically significant (N or P). Comment on this.

10. Lines 360-361: "…we found evidence…". The rslope=f(mean TF SO4 deposition) relations in Figure 9 are no evidence, however, they show a relatively strong indication (r2=0.288) on that the SO4 deposition may tangibly affect the rslope values in acidic soils. Rephrase the sentence.

11. Lines 367-372: Complement the GDA analysis with the DOC concentrations as an independent continuous variable and comment on the results. Is DOC concentration an important variable for explaining the variation (cf. general comment 3)?

12. Line 414: A bracket in front of Fig. 11A is missing.

13. Line 536: Again the total number of observations (n=174) does not match the number (n=191) stated in Figure 1. In the methods chapter, it may be wise to further explain the different number of observations occurring in different analysis and why so.

14. Lines 645-647: "We found evidence that soil pH determines the response of trends of DOC in soil solution to SO42- deposition…". This statement is not correct. What you have found is a relation between relative slopes of DOC and S-deposition in very acidic soils with a pH<4.2 in soil solution, but not in non-acid soils with a pH>5 (Figure 9). The multivariate analyses do not show that as stated. The relation in soils with 4.2≤pH≤5 is not shown or discussed in the text. Additionally, the statement refers to the entire soil column, but I suppose that the low pH in soil solution (pH<4.2) is primarily found in O-horizons and upper mineral soils. Rephrase the statement.

15. In the conclusions, I would suggest that you stress the large local variation related to a multitude of factors and discuss the regional processes in a more humble way, supported by your results. I also suggest that you describe the differences found between DOC trends in organic and mineral soil layers and possible influences by different drivers/processes. Finally, if there is any relation between DOC concentration level and DOC trends (levels or directions), this should be stressed.

Comments on tables and figures

1. Table 1: In the legend, information on how $0.05 \leq p \leq 0.1$ is indicated is missing (italics?). In the table, there is a mess among grey, bold and italic figures. Related to the SMK results, the number zero is sometimes missing.

2. Table 2: The different statistical methods do not always show the same direction on rslope for all soil layers (BFAST M02 and PMK M24 are negative). This should be commented on in the text.

3. Table 3. Change name on slope to rslope in column headings and explain in legend. Which year(s) do the S and N depositions data refer to?

4. Table 4, Legend: What do you mean with ". . .during the last years. . ."? Explain.

5. Figure 2: Weight_P is missing on the X-axis

6. Figure 3, legend: Explain boxplots (c.f. Figure 6) and "n" in figure.

7. Figure 7: Defining that the trends refer to DOC is missing in the legend. The Y-axis is too short in Figure 7C and perhaps also in the others. Maximum values on the Y-axis seem to be very close to the observed maximum numbers.

8. Figure 8. Define whether it is natural logarithms or 10-logarithms on the X-axis. The X-axis is too short in Figure 8B.

9. Figure 9: In the legend, define which soil layers the data points refer to.

10. Figure 10: Use the same scales on the XY-axes in Figure A and B.

11. Figure 11: In the legend change from (>15 kg N ha-1 yr-1) to (<15 kg N ha-1 yr-1)

Comments on Supplementary material

1. S2: For the GDA analysis, it is unclear whether the "Weighed positive" and "Weighed negative" trends are included. Clarify.

2. S2: For the SEM analysis, it is unclear whether the analyses are performed on SEN slopes or rslopes. Clarify.

3. Figure S1: The legend box hides some bars.

---

## Referee Comment (RC2) · Anonymous Referee #2 · 9 Mar 2016

Review on work of Camino-Serrano et al. (March 2016) Analysing an european-scale dataset on dissolved organic carbon (DOC) in soil solution, the study seeks 1) to determine the temporal evolution of DOC under forests during the last twenty years (1991-2011) and 2) to correlate it with changes of environmental factors (e.g. atmospheric N & S depositions) in order to propose a mechanistic understanding of DOC evolution. The main hypothesis is that the current increase of DOC content in european surface waters arises from an increase in soil solution DOC content resulting from the recovery of past acidification. The study presents interesting results deserving publication in BG such as 1) an overall increase in DOC content in the organic and the deepest mineral layers and 2) the correlation of this increase with changes in atmospheric N & S depositions and tree growth. However, saying that this article is not very clear is litotes. The abstract and introduction are rather well written. However, the result and discussion sections are extremely hard to read, the number of figures and tables is not tenable (I counted 27 items, figures+tables) and the usefulness of numerous statistical analyses is not convincing since they provide similar results and conclusions. I think that an effort of synthesis is necessary to simplify messages and prevent a dilution of important results with accessory observations. For example, at the end of the reading of your manuscript, I was not able to say whether sulphate depositions increase or decrease soil solution DOC content. In your abstract, it is suggested that DOC concentrations and sulphate depositions are positively linked, a statement which is then contradicted in the manuscript (e.g. L412-413 but L348-349). Moreover, it would be useful to create a figure summarizing the main chemical reactions and biological processes controlling soil solution DOC content. More fundamentally, I am not sure that the measurement of soil solution DOC provides an accurate estimate of the amount of DOC flowing out of terrestrial ecosystems (and supply of DOC to surface water). The leaching of DOC happens at specific moments of the year depending on hydric balance (precipitation-evapotranspiration), soil type, plant activity etc. I am even sure that the DOC soil solution concentration can be inversely related to DOC leaching in some conditions. Just an example: soil solution DOC concentration is higher in summer than in winter, but DOC leaching only occurs in winter in France. This issue could explain why the present study fails to show clear overall trend in soil solution DOC at individual plot and soil depths. A warming-induced change of ecosystem water balance could also contribute to changes in DOC content in soil solution and surface water. Therefore, I suggest to present (in manuscript or in supplementary materials) the volume of water harvested in lysimeters or calculations of theoretical water balance (precipication-evapotranspiration). I am not really convinced by the relevance of removing the breakpoints. These breakpoints are not necessary the result of site disturbances (change of sensors etc) but could result from sudden change of atmospheric chemical composition or ecosystem functionning. After all, SO2 emissions by human activities have been reduced by 70% during 90s.

The terminology used in the manuscript is often not clear. The term "trend" is vague and does not specifically refer to change with time. The terms "trend slope", "trend direction" and "relative trend slope" are even more difficult to understand. The terms "depositions" and "troughfall" are interchangeably used, I suggest you to use only one of the two terms. The term "fertile soil" is weak and, as usual, does not refer to measurable variable. The fact that tree growth is high does not necessary mean that the soil is fertile. The tree growth is often linked to forest dynamics and age (tree growth of old forests is typically slow irrespective of soil characteristics; tree growth after forest disturbance (drought events, storm etc) is typically high because tree mortality allows the recruitment of seedling with fast growth rate).

―――――――――――――――――――

---

## Author Comment (AC1) · 20 May 2016

General comments

1. In the manuscript and supplement, there is no information on the soil solution chemistry at the studied plots. A quantitative description of the concentrations of DOC and other relevant water chemical parameters is missing. This information could be given in tables and figures in manuscript and supplementary materials, describing e.g. median values, 25- and 75-percentiles and number of observations or in boxplots. The information should be available separated on collector type and soil layer (cf. Table 1) for all classes used in the assessment (forest type, soil type, soil pH, N and S depositions).

We agree that a quantitative description of the concentrations of DOC is missing. In the revised version, we will add a table to the Supplementary Material with the median values, 25- and 75- percentiles and number of observations of DOC, and other water chemical parameters (pH, conductivity, $Ca_{2+}$, $Mg_{2+}$, $SO_{42-}$, $NO_{3-}$, $Al_{3+}$ and $Fe_{2+}$). The information will be separated on collector type, soil layer and forest type, as for Table 1 in submitted manuscript.

2. The above information should be used for assessing how the standardized trends (rslope) are affected by the median concentrations (see comment 3) and for defining whether the statistically significant rslope trends in the filtered data (LMM, SMK and PMK) are found over the entire DOC concentration range or within certain intervals. Additionally, what does the statistically significant rslope trends correspond to in DOC concentration trends? Are they quantitatively large or not?

To determine whether the absolute trend in DOC is quantitatively large or not from an ecological perspective, we used the median DOC as a reference. That is, we calculated the relative (standardized) trend slope dividing the absolute trend slope by the median DOC level.

Before deciding to use the standardized trend slopes, we tested whether there was a relation between the DOC Sen slopes and the median DOC concentrations, and we saw that a priori there was no such a relationship (Figure 1).

This Figure 1 will be included in Supplementary Material in order to support the discussion of the relation between the relative slopes and the DOC concentrations. Moreover, we will use the information gained from the new table with median DOC concentrations (Comment 1) to discuss the possible effect that DOC concentration levels may have, i.e., exaggerating the DOC trends at low DOC levels, such as may occur in the subsoil.

3. The trends are reported in standardized terms (rslope), which means that the slope (Sen slope) of each time series was divided by the median concentration over the studied period. This implies that the rslope-value can be identical regardless of the

DOC concentration level. Hence, rslope will be 0.1 if you have a trend of 0.2 mg DOC yr-1 and a median DOC concentration of 2 mg l-1 at one plot-soil depth as well as if the trend is 5 mg DOC yr-1 at 50 mg DOC l-1 at another plot-soil depth. The significance of the latter example is of course much larger than in the former. Are e.g. the statistical trends in deep mineral soils (Table 1, layer M8) a result of this phenomenon?

Indeed, the fact that we used the standardized terms (rslope) implies that the rslope may be identical for two sites with very different mean DOC concentrations. It is also true that DOC concentration decreases with depth and is therefore lower in the deep mineral soil than in the upper mineral soil. Although the actual change in DOC concentration with depth is normally more limited than that of the example taken in this comment. However, in our opinion, the detected trends in mineral soils are not less important because DOC concentrations are lower. In fact, we decided to use the standardized slopes to be able to compare trends amongst sites, independently of the absolute DOC concentrations. Furthermore, standardizing the slopes allows for comparisons in trends among soil layers, which have very different DOC values. Otherwise, using the absolute trends will introduce a bias when we try to explain the DOC trends with other parameters, because the trend slope will be highly dependent on the initial DOC concentrations of the site.

Actually, both relative and absolute slopes will give different information: as mentioned, with the absolute slope we can see the real magnitude and significance of the trend, but it does not allow for comparison among sites or horizons. Consequently, as we were interested in whether there is a general DOC trend and in the intercomparison among sites at European scale, we decided that using the relative slope was more consistent. It is however possible that the large statistical trends in deep mineral soils are a result of this phenomenon (see Figure 1 below) and in the revised manuscript we will test for it.

Finally, we would like to highlight that the statistical analyses (LMM, SMK and PMK) were done on the absolute value and were then transformed to facilitate the interpretation. Thus, the fact that we express the resulting trend in relative terms can influence our interpretation but has no influence on the statistical test itself (carried out on the absolute values of DOC). In order to provide all the information to the reader, we will add the median DOC concentration in the table reporting the trends. A more detailed discussion on this issue (relative/absolute slopes) will be added in the revised manuscript.

4. Evaluating hundreds of time series may introduce random effects affecting the number of statistically significant trends. The theoretical and if possible quantitative implication of this (false positive and false negative trends) should be discussed.

To check how robust the significant tests are we did an extra test (not included in the manuscript). We considered the trend tests to be significant at the 0.01 level, instead of 0.05, and compared the number of significant and non-significant trends (Table 1):

We can see that, at the 0.05 level, out of the 258 tests, we would expect to see a significance just by chance (type I error) in 13 cases, so about 6-7 in positive and about 6-7 in negative direction. As we detected many more trends in both direction, most of the significant results are not a type I error but are genuine effects. Moreover, the fact that we found many trends in both positive and negative direction implies that patterns vary across Europe, an argument that also stands if we test at the 0.01 level. Therefore, we believe that we can trust that the conclusions based on trends at 0.05 level are correct. We could add this conclusion in the discussion section (chapter 4.1.1.) in the revised manuscript.

Finally, the results from the Linear Mixed Models (LMM), also presented in the manuscript (Table 1 in submitted manuscript), are not subject to this issue.

5. Using relative data in multivariate statistical models like GDA, may cause biased results strongly exaggerating the effects of trends in low DOC-concentration soil horizons. A discussion on the latter is missing.

We decided to use relative data in the multivariate statistical models to be able to

compare trends among sites and soil layers (see answer to comment 3). However, it is true that a discussion on the effects of this choice is missing. We will mention the potential exaggeration of the effects of trends in low DOC-concentration soil horizons in the discussion of the revised manuscript.

6. Throughout the manuscript, the information gained from comments 1-3 should be commented where relevant. The information is especially important for the results and discussions dealing with the directions and controls on soil solution temporal trends (Chapters 3.1 and 4.2) partly based on the GDA and SEM results. Are the indicated effects quantitatively important, do they occur both at low and high DOC concentrations in soil solution and has the DOC concentration level any influence on the trend strength and direction?

We will rework the discussion in the revised manuscript. The controls on soil solution DOC trends have been discussed from a relative point of view, as we focused on explaining the high heterogeneity of DOC trends found across Europe, instead of the quantification of the trends at local scale. We can add an explanation of this. We will also describe if the trends occur at low and high DOC concentrations based on the new table (See Comment 1). Finally, the initial DOC concentration seems not to influence the trend strength and direction and this will be added in the text.

7. The title of chapter 4.1.1 as well as some of the text are obscure. The number of non-significant trends is determined by the data and the statistical methods used. The authors themselves have selected data after quality check and chosen the statistical methods including probabilities to accept or reject trends. By speculating on whether the non-significant trends are real or not, the authors seem to reject their own data and methods? Change title and remove these speculations, but keep the general discussions on factors affecting trend analysis including what you have found related to comment 4 (see above).

We agree with this comment and the title of chapter 4.1.1. will be changed to "Evaluation of non-significant trends" in the revised manuscript. Some speculative sentences will be deleted and the discussion mentioned in reply to comment 4 will be added.

Detailed comments

1. Lines 71-81: Riparian zones and peat lands, the most important DOC sources for surface waters are not referred to. Add some text and references. In the introduction, we focused on forest soils because our study deals only with forest soils, but we suggest to add some text and references on the importance of riparian zones and peat lands.

2. Line 205: ". . .more than 60 observations of soil solution DOC of individual or groups of collectors". What do the 60 observations refer to? Individual or groups of collectors? If the latter, was it pooled composite samples?

As mentioned previously in the manuscript (line 159-160), in some countries, samples from these replicates were pooled before analyses or averaged prior to data transmission. Therefore, we selected time series with more than 60 observations from individual or groups of collectors, when the samples were pooled before analysis or prior to data transmission.

3. Line 209: In Figure 1, the number is 436 time series instead of 529. Which figure is the correct one?

Table 2 below clearly explains where these numbers came from, as it seemed to be confusing throughout the manuscript. This table will be included in the revised manuscript (in Supplementary Material).

Thus, both 436 and 529 were correct, but they referred to different stages of the study: 529 in the number of time series for the entire dataset and 436 is the number of time series after aggregating per plot-soil depth combinations. In the revised manuscript, we will change the number in Figure 1 for consistency.

4. Line 218: Did you use the same pH ranges for all soil horizons? If so, you may have a

bias towards organic and upper mineral soil horizons in the Low pH class. Additionally it is not clear whether it is soil pH as stated in text or in soil solution as stated in Figure 9? Clarify!

Yes, the same pH ranges were used for all the soil horizons and as a consequence we have more organic soil horizons in the Low pH class (Table 3).

Since the manuscript contains too many figures (also suggested by referee 2), we suggest to move Figure 9 to Supplementary Material. We still find this result as potentially important, and by moving the graph to Supplementary Material we can refer to it through the text, but it will not be part of the manuscript. When discussing the graph, the issue of the bias towards organic and upper mineral soil horizons in the Low pH class will be discussed.

Sites were classified according to soil solution pH, and thus the revised text will be corrected.

5. Lines 219-222: From which time period do the S and N depositions originate? Is it median values or. . .?

S and N deposition data covers the period 1999-2010 (Waldner et al., 2014) and for our classification (Lines 219-222), we used mean values of deposition for this period. In the revised manuscript, this will be mentioned in the text.

 c Waldner, P., Marchetto, A., Thimonier, A., Schmitt, M., Rogora, M., Granke, O., Mues, V., Hansen, K., Karlsson, G. P., Zlindra, D., Clarke, N., Verstraeten, A., Lazdins, A., Schimming, C., Iacoban, C., Lindroos, A. J., Vanguelova, E., Benham, S., Meesenburg, H., Nicolas, M., Kowalska, A., Apuhtin, V., Napa, U., Lachmanova, Z., Kristoefel, F., Bleeker, A., Ingerslev, M., Vesterdal, L., Molina, J., Fischer, U., Seidling, W., Jonard, M., O'Dea, P., Johnson, J., Fischer, R., and Lorenz, M.: Detection of temporal trends in atmospheric deposition of inorganic nitrogen and sulphate to forests in Europe, Atmos Environ, 95, 363-374, 2014.

6. Line 276: Add p-value to ". . .overall positive trend. . .". p<0.05 or p<0.10??

The p-value (p<0.05) will be added in the text referring to this overall positive trend in the organic layer.

7. Lines 296-301 and Table 2. In the last sentence, it is stated that trends computed with SMK and PMK agreed well. However, at soil depth M24 the two methods results in very different rslopes (Table 2) both as regards directions and values. Comment on this and present a possible explanation.

We have checked for these results and the difference in the rslopes between SMK and PMK originates from the difference in sites available for computing the SMK and the PMK. There are two extra sites for which SMK tests were performed, but not the PMK. These two extra sites show a positive trend (1.1 and 2 % yr -1), creating the difference in the median value at M24 between the two methods (Table 2 in submitted manuscript). However, when using exactly the same set of sites, the trend did not differ between the two methods.

8. Lines 309-311: nNS-trends=104, nP-trends=91 and nN-trends=63 makes up a total of 258 time series, which corresponds to the value in Supplementary materials. However, the number of monotonic trends is 191 according to Figure 1. Correct where appropriate.

The numbers 258 and 191 correspond to different observations (see response to detailed comment 3): 258 is the number of time series with less than 60 observations and more than 10 years, while 191 refers to the same time series, but after aggregation per plot-depth combination. This will be clarified in the text of the revised manuscript.

9. Lines 324-332 and Table 2: There are increased rslope values towards deeper mineral soil horizons. Is this a result of lower soil solution DOC concentrations (cf. general comment 3) and thereby very small DOC trends in absolute numbers? The rslope values in the O-horizon, generally showing high DOC concentrations, are close

to those found for M8, indicating large DOC trends if statistically significant (N or P). Comment on this.

We have checked the median values of the absolute DOC slopes and absolute trends in the organic horizons are indeed higher than in mineral soils (0.33 mg L-1 yr-1 for the organic layer versus 0.03 below 80 cm). This is a natural consequence of lower DOC concentrations in deep soils. This issue will be commented in the paragraph in lines 324-332 in the revised manuscript.

10. Lines 360-361: ". . .we found evidence. . .". The rslope=f(mean TF SO4 deposition) relations in Figure 9 are no evidence, however, they show a relatively strong indication (r2=0.288) on that the SO4 deposition may tangibly affect the rslope values in acidic soils. Rephrase the sentence.

In the revised manuscript, the sentence will be rephrased to "However, our results point out to a control of soil acidity on the SO42- deposition effect on the trends of DOC in soil solution (Fig. 9)." Figure 9 will be moved to Supplementary Material.

11. Lines 367-372: Complement the GDA analysis with the DOC concentrations as an independent continuous variable and comment on the results. Is DOC concentration an important variable for explaining the variation (cf. general comment 3)?

As we did not find a relationship between median DOC concentrations and absolute or relative slopes (see Figure 1 below), we do not see necessary to include DOC as a variable in the GDA. Introducing an additional variable would lower the robustness of the analysis. Moreover, we suggest to remove the GDA analysis from the manuscript (Fig. 10 and Table 4 in submitted mansucript) to shorten the paper (see answers to comments from referee 2).

12. Line 414: A bracket in front of Fig. 11A is missing.

This will be corrected in the revised manuscript.

13. Line 536: Again the total number of observations (n=174) does not match the

number (n=191) stated in Figure 1. In the methods chapter, it may be wise to further explain the different number of observations occurring in different analysis and why so.

191 is the total number of time series aggregated per plot and soil layer, 174 of this 191 time series show positive, negative or non-significant trends. The rest of plot/soil depth combinations (17) correspond to the plots that showed different trends (P, N or NS) in DOC within the same depth interval, which was the case for 17 plot-depth combinations (16 in Germany and one in Norway) (lines 321-322 in submitted manuscript). In the revised manuscript, this will be added in the text in line 536 for clarification.

14. Lines 645-647: "We found evidence that soil pH determines the response of trends of DOC in soil solution to SO42- deposition. . .". This statement is not correct. What you have found is a relation between relative slopes of DOC and S-deposition in very acidic soils with a pH<4.2 in soil solution, but not in non-acid soils with a pH>5 (Figure 9). The multivariate analyses do not show that as stated. The relation in soils with 4.2_pH_5 is not shown or discussed in the text. Additionally, the statement refers to the entire soil column, but I suppose that the low pH in soil solution (pH<4.2) is primarily found in O-horizons and upper mineral soils. Rephrase the statement.

We did not show the relation in soils with intermediate pH (between 4.2 and 5) to avoid introducing more information, as the manuscript is already dense in statistical analysis, number of figures and tables.

Regarding the fact that the statement refers to the entire soil column, we could not do the statistical analysis separately for the different soil layers due to a lack of data. Therefore, to check the influence of mixing soil layers, we re-did the SEMs models (Figure 11 in submitted manuscript) with horizon type (organic versus mineral) as an explanatory variable. For the model obtained for 1) all the cases, 2) for low and medium nitrogen deposition and 3) for high nitrogen deposition (Figure 11A, 11B and 11C, respectively), the variable "depth" (organic vs mineral) was not significantly correlated with DOC slopes (p =0.85, p=0.34 and p=0.56). Based on this test, horizon type does

not appear to play a role in explaining the differences between the trend slopes of DOC and, thus we trusted the findings from the SEMs even when mixing soil layers.

In the revised manuscript, this statement will be rephrased as "We found that the response of trends of DOC in soil solution to $SO_4^{2-}$ deposition depends on soil pH" and the issue of the bias towards organic horizons at Low pH class will be discussed (see answer to detailed comment 4).

15. In the conclusions, I would suggest that you stress the large local variation related to a multitude of factors and discuss the regional processes in a more humble way, supported by your results. I also suggest that you describe the differences found between DOC trends in organic and mineral soil layers and possible influences by different drivers/processes. Finally, if there is any relation between DOC concentration level and DOC trends (levels or directions), this should be stressed.

We agree that we should focus the conclusions more on the large local variation instead of the regional processes. This will be changed in the conclusions of the revised manuscript. We will also stress the differences between the DOC trends found in the organic vs mineral layers and the information gained from the influence of DOC concentration levels in the DOC trends will be added. However, we think that we cannot describe the influences by different drivers/processes based on our results without being too speculative for a conclusion.

Comments on tables and figures

1. Table 1: In the legend, information on how 0.05_p_0.1 is indicated is missing (italics?). In the table, there is a mess among grey, bold and italic figures. Related to the SMK results, the number zero is sometimes missing. 2. Table 2:The different statistical methods do not always show the same direction on rslope for all soil layers (BFAST M02 and PMK M24 are negative). This should be commented on in the text. 3. Table 3. Change name on slope to rslope in column headings and explain in legend. Which year(s) do the S and N depositions data refer to? 4. Table 4, Legend: What do you

mean with ". . .during the last years. . ."? Explain. 5. Figure 2: Weight_P is missing on the X-axis 6. Figure 3, legend: Explain boxplots (c.f. Figure 6) and "n" in figure. 7. Figure 7: Defining that the trends refer to DOC is missing in the legend. The Y-axis is too short in Figure 7C and perhaps also in the others. Maximum values on the Y-axis seem to be very close to the observed maximum numbers. 8. Figure 8. Define whether it is natural logarithms or 10-logarithms on the X-axis. The X-axis is too short in Figure 8B. 9. Figure 9: In the legend, define which soil layers the data points refer to. 10. Figure 10: Use the same scales on the XY-axes in Figure A and B. 11. Figure 11: In the legend change from (>15 kg N ha-1 yr-1) to (<15 kg N ha-1 yr-1)

All these Tables and Figures will be corrected accordingly to these comments.

Comments on Supplementary material

1. S2: For the GDA analysis, it is unclear whether the "Weighed positive" and "Weighed negative" trends are included. Clarify.

For the GDA analysis, the classes Weighted positive and Weighted negative are not included. This will be clarified in the revised version.

2. S2: For the SEM analysis, it is unclear whether the analyses are performed on SEN slopes or rslopes. Clarify.

The SEM analyses were performed on the relative slopes (rslopes): this will be clarified in the revised version.

3. Figure S1: The legend box hides some bars.

This figure will be re-done.
* * *
[Figure]

**Fig. 1.** Figure 1. Left: Standardize trends (relative slope DOC) versus median DOC concentrations. Right: Absolute trends (absolute slope DOC) versus median DOC concentrations. The different colors represent

|          | Significant positive | Significant negative | Non-significant |
|----------|----------------------|----------------------|-----------------|
| P<0.05   | 91                   | 63                   | 104             |
| P<0.01   | 70                   | 50                   | 138             |

**Fig. 2.** Table 1. Comparison significant positive, negative and non-significant at p<0.05 and p<0.01.

|  | Entire dataset | Without breakpoints |
|---|---|---|
| All time series | 1480 (173 plots) | 1480 |
| Time series <60 observations and > 10 years | 529 | 258 |
| Aggregated plot-depth combinations | 436 | 191 |
| Plots | 118 | 97 |

**Fig. 3.** Table 2. Summary of number of time series used in the study

|      | High pH | Low pH |
|------|---------|--------|
| O    | 7       | 14     |
| M02  | 15      | 13     |
| M24  | 9       | 3      |
| M48  | 21      | 6      |
| M8   | 9       | 6      |

**Fig. 4.** Table 3. Number of cases at high and low pH classified by organic layer.

---

## Author Comment (AC2) · 20 May 2016

Referee 2: However, saying that this article is not very clear is litotes. The abstract and introduction are rather well written. However, the result and discussion sections are extremely hard to read, the number of figures and tables is not tenable (I counted 27 items, figures+tables) and the usefulness of numerous statistical analyses is not convincing since they provide similar results and conclusions. I think that an effort of synthesis is necessary to simplify messages and prevent a dilution of important results with accessory observations. For example, at the end of the reading of your manuscript, I was not able to say whether sulphate depositions increase or decrease soil solution DOC content. In your abstract, it is suggested that DOC concentrations

and sulphate depositions are positively linked, a statement which is then contradicted in the manuscript (e.g. L412-413 but L348-349).

We agree that the manuscript includes many statistical approaches and we understand that this might confuse the reader. This approach diversity comes from the large community of scientists involved in the study by providing data and scientific input. Concerning the temporal analyses of DOC concentrations, we decided to show results from the different statistical methods (LMM, seasonal and partial Mann Kendall) because the approaches are complementary. Each method has pros and cons. This allowed us to show that DOC concentrations have increased during the observation period overall in coniferous forests in the organic layers. However, at individual plots and depths, DOC concentrations did not show any consistent temporal trend (increase, decrease or no change). We could also show that there was no geographical pattern either.

Concerning the multivariate analyses, we propose to shorten the manuscript by removing three figures (Fig 8, Fig. 9, Fig. 10), parts of Figure 7 (7A, 7B), and one table (Table 4). Fig. 9 will be moved to Supplementary Material. This would not change the conclusions of the manuscript. In Figure 7, we would focus on the relationships between the temporal trends of DOC concentrations, forest types and classes of stem volume increment (proxy for forest productivity). The relationships between trends in DOC concentrations and SO4 and NO3 throughfall deposition would be explored in Figure 11.

Referee 2: Moreover, it would be useful to create a figure summarizing the main chemical reactions and biological processes controlling soil solution DOC content

We could include such a figure in the introduction based on the numerous mechanistic models proposed in the literature. Our results come from an exploratory statistical approach (and not deterministic) of a large European dataset and it would be preposterous at this stage to build a model based on such a variety of local (e.g. soil

properties) and regional (e.g. atmospheric deposition) factors.

Referee 2: More fundamentally, I am not sure that the measurement of soil solution DOC provides an accurate estimate of the amount of DOC flowing out of terrestrial ecosystems (and supply of DOC to surface water). The leaching of DOC happens at specific moments of the year depending on hydric balance (precipitationevapotranspiration), soil type, plant activity etc. I am even sure that the DOC soil solution concentration can be inversely related to DOC leaching in some conditions. Just an example: soil solution DOC concentration is higher in summer than in winter, but DOC leaching only occurs in winter in France. This issue could explain why the present study fails to show clear overall trend in soil solution DOC at individual plot and soil depths. A warming-induced change of ecosystem water balance could also contribute to changes in DOC content in soil solution and surface water. Therefore, I suggest to present (in manuscript or in supplementary materials) the volume of water harvested in lysimeters or calculations of theoretical water balance (precipication-evapotranspiration).

We agree with the referee that we did not assess DOC fluxes flowing out from forest soils. Our time series analysis aimed to detect long-term changes of DOC concentrations that are not due to seasonal effects or dilution-concentration effects caused by fluctuations in soil water content. Therefore, we decided to apply both Seasonal Mann Kendall and Partial Mann Kendall using precipitation as a co-variable to remove the seasonality and dilution-concentration effects. This method allowed us to detect significant monotonic changes (increase or decrease) of DOC concentrations over a period of 10 years at least. Studies having shown temporal changes of DOC in surface waters also reported concentrations rather than fluxes (e.g. Worral & Burt 2004, Evans et al. 2005, Monteith et al. 2007, Dawson et al. 2009, all cited in the manuscript).

Using water volume collected by lysimeters to assess water fluxes was not possible, since these data were available since 2011 only. In addition, the volume collected by tension lysimetry depends on the suction applied to the system. To assess water fluxes at different soil depths, we would need to model the water balance at 118 forest sites, which is very challenging, since many input parameters (meteorology, soil, vegetation) would be required. This was beyond the scope of this study. A simple estimate based on the difference between precipitation and evapotranspiration, would add a substantial uncertainty to the calculation of DOC fluxes and therefore detecting long-term changes of DOC fluxes would be even more difficult. Because of large variations in soil water fluxes (e.g. Borken et al. 2011, Verstraeten et al. 2014, Meesenburg et al. 2016), it is more difficult to detect long-term trends of fluxes than long-term trends of concentrations in soil solution. Since most times series of DOC concentrations in soil solution do not indicate any long-term trend in our dataset, the chance of finding long-term changes in DOC fluxes are even lower.

References

- Borken, W., Ahrens, B., Schulz, C., & Zimmermann, L. (2011). Site‐to‐site variability and temporal trends of DOC concentrations and fluxes in temperate forest soils. Global change biology, 17(7), 2428-2443.

- Meesenburg, H., Ahrends, B., Fleck, S., Wagner, M., Fortmann, H., Scheler, B., ... & Meiwes, K. J. (2016). Long-term changes of ecosystem services at Solling, Germany: Recovery from acidification, but increasing nitrogen saturation?. Ecological Indicators, 65, 103–112

- Verstraeten, A., De Vos, B., Neirynck, J., Roskams, P., & Hens, M. (2014). Impact of air-borne or canopy-derived dissolved organic carbon (DOC) on forest soil solution DOC in Flanders, Belgium. Atmospheric Environment, 83, 155-165

Referee 2: I am not really convinced by the relevance of removing the breakpoints. These breakpoints are not necessary the result of site disturbances (change of sensors etc) but could result from sudden change of atmospheric chemical composition or ecosystem functionning.

Monotonicity of time series is generally assumed when analyzing DOC data for tem-

poral trends (Filella and Rodriguez-Murillo, 2014). However, it is rarely statistically tested and, thus, potential abrupt changes in the time series may be overlooked. This issue becomes important in temporal trend analysis since a breakpoint may cause changes in the direction of the trend and could lead us, for example, to classify a time series as constant, when in reality we may have averaged out separate periods with significant changes (de Jong et al., 2013). On the other hand, breakpoints may erroneously induce the detection of a significant trend in long-term time series due to artifacts (see Supplementary Material).The aim of our study is to analyze monotonic trends related to factors that have been measured within the ICP Forests database. Therefore, DOC time series were first analyzed using the Breaks For Additive Seasonal and Trend (BFAST) algorithm to detect the presence of breakpoints.

We agree that removing breakpoints using the BFAST technique may remove time series that show abrupt changes not only due to artifacts (collector replacement, etc.,), but also due to natural causes (meteorological conditions, extreme events), forest management (changes in soil condition, thinning, etc.), sudden change of atmospheric chemical composition or ecosystem functionning. Nevertheless, many breakpoints are the consequence of technical issues or even inconsistencies in the database. The ICP Forests soil solution dataset has a great potential for analysis of large scale trends, but at the same time it may also contains some inconsistencies. The BFAST analysis proved to be effective at removing breakpoints caused by some dataset errors and thus the most defective time series were left out.

Although the investigation of the potential causes of the abrupt changes (breakpoints) in the individual time series can indeed provide a very valuable information, we do not count with the site-level information necessary for that purpose. To attribute the different causes of the breakpoints at site scale, we would need information of the management and climate history at each particular site, which is not available at the time of writing the present manuscript. Consequently, we cannot be sure of the origin of each breakpoint, and thus we decided to leave out all the time series showing abrupt

changes to avoid erroneous detections of significant trends. Moreover, in this way, we are confident that the trends found in the time series that we analyzed are not a consequence of local factors. The alternative is to study each time series individually to identify the local (or regional) factors causing abrupt changes at plot scale, and this task is beyond the scope of this study, but is a very interesting topic to be addressed in a follow-up paper.

References

- Filella, M. and Rodriguez-Murillo, J. C.: Long-term Trends of Organic Carbon Concentrations in Freshwaters: Strengths and Weaknesses of Existing Evidence, Water-Sui, 6, 1360-1418, 2014.

- de Jong, R., Verbesselt, J., Zeileis, A., and Schaepman, M. E.: Shifts in Global Vegetation Activity Trends, Remote Sens-Basel, 5, 1117-1133, 2013.

Referee 2: The terminology used in the manuscript is often not clear. The term "trend" is vague and does not specifically refer to change with time. The terms "trend slope", "trend direction" and "relative trend slope" are even more difficult to understand.

To avoid any confusion about the term "trend", we could add a small paragraph in the Method section explaining the different temporal components of time series analyses. Time series can be typically decomposed into random noise, seasonal and trend components (e.g De Livera et al., 2011).

We also suggest to explain the meaning of "trend slope", "trend direction" and "relative trend slope" in the Method section. However we would prefer to keep the same terms in the whole manuscript. They frequently appear and we would need much more words if we had to explain them in each occurrence.

Reference

- De Livera, A. M., Hyndman, R. J., & Snyder, R. D. (2011). Forecasting time series with complex seasonal patterns using exponential smoothing. Journal of the American

Statistical Association, 106(496), 1513-1527.

Referee 2: The terms "depositions" and "troughfall" are interchangeably used, I suggest you to use only one of the two terms.

We used throughfall deposition of sulfate and organic nitrogen, except in Table 3 where bulk deposition was provided for three sites because throughfall deposition was not available. This was marked by an asterisk in the table. To avoid any confusion, we will use only throughfall deposition in the text.

Referee 2: The term "fertile soil" is weak and, as usual, does not refer to measurable variable. The fact that tree growth is high does not necessary mean that the soil is fertile. The tree growth is often linked to forest dynamics and age (tree growth of old forests is typically slow irrespective of soil characteristics; tree growth after forest disturbance (drought events, storm etc) is typically high because tree mortality allows the recruitment of seedling with fast growth rate).

We agree with the referee that tree growth is not necessarily related to soil fertility. We suggest to reformulate chapter 4.2.1. This chapter aims to discuss the relationship between forest productivity (in our case only stem growth is available) and DOC in soil solution. A number of factors such as climate, soil water availability, soil fertility, tree age and competition between neighboring trees can influence tree growth.

---

## Author Response (AR1)

**Answers to Referee 1**

**General comments**

*1. In the manuscript and supplement, there is no information on the soil solution chemistry at the studied plots. A quantitative description of the concentrations of DOC and other relevant water chemical parameters is missing. This information could be given in tables and figures in manuscript and supplementary materials, describing e.g. median values, 25- and 75-percentiles and number of observations or in boxplots. The information should be available separated on collector type and soil layer (cf. Table 1) for all classes used in the assessment (forest type, soil type, soil pH, N and S depositions).*

We agree that a quantitative description of the concentrations of DOC is missing. We have added tables to the Supplementary Material with the median values, 25- and 75- percentiles and number of observations of DOC, and other water chemical parameters: pH, conductivity, Ca, Mg, $SO_4^{2-}$, $NO_3^-$, $Al^{3+}$ (Tables S4 to S11, pages 21-27 in the Supplementary Material of the revised manuscript). The information is separated on collector type, soil layer and forest type, as for Table 1 in submitted manuscript.
* * *
*2. The above information should be used for assessing how the standardized trends (rslope) are affected by the median concentrations (see comment 3) and for defining whether the statistically significant rslope trends in the filtered data (LMM, SMK and PMK) are found over the entire DOC concentration range or within certain intervals. Additionally, what does the statistically significant rslope trends correspond to in DOC concentration trends? Are they quantitatively large or not?*

To determine whether the absolute trend in DOC is quantitatively large or not from an ecological perspective, we used the median DOC as a reference. That is, we calculated the relative (standardized) trend slope dividing the absolute trend slope by the median DOC level. Nevertheless, since the aim of this paper is to understand the soil solution DOC trends at European level, being able to compare them among sites and soil depths, we discussed the results from a relative point of view. The quantification of the DOC trends at individual plots are, however, beyond the scope of this study.

In Supplementary Material of the revised manuscript we have added a new figure (Figure S5, page 20), showing the effect of the median DOC levels on the relative and absolute trend slopes. We saw that, a priori, there was no relationship between median DOC concentrations and DOC trend slopes. This Figure was also useful to support the discussion about the potential effect that our decision of using the standardized slopes may have, i.e., exaggerating the DOC trends at low DOC levels, such as may occur in the subsoil (see Comment 3).
* * *
*3. The trends are reported in standardized terms (rslope), which means that the slope (Sen slope) of each time series was divided by the median concentration over the studied period. This implies that the rslope-value can be identical regardless of the DOC concentration level. Hence, rslope will be 0.1 if you have a trend of 0.2 mg DOC yr-1 and a median DOC concentration of 2 mg l-1 at one plot-soil depth as well as if the trend is 5 mg DOC yr-1 at 50 mg DOC l-1 at another plot-soil depth. The significance of the latter example is of course much larger than in the former. Are e.g. the statistical trends in deep mineral soils (Table 1, layer M8) a result of this phenomenon?*

We used the standardized slopes instead of the absolute trend slopes for our analysis in order to remove the effect that the DOC concentration levels have on the absolute DOC trends. Standardizing the slopes allows us for comparisons in trends among soil layers, which have very different DOC

values. Otherwise, using the absolute trends will introduce a bias when we try to explain the DOC trends with other parameters, because the trend slope will be highly dependent on the median DOC concentrations of the site.

It is indeed true that by standardizing the slopes (rslopes), we may have identical DOC trend slopes for two sites with very different mean DOC concentrations. It is also true that DOC concentration decreases with depth and is therefore lower in the deep mineral soil than in the upper mineral soil. However, in our opinion, the detected trends in mineral soils are not less important because DOC concentrations are lower. In fact, as previously commented, we decided to use the standardized slopes to be able to compare trends amongst sites, independently of the absolute DOC concentrations.

Nevertheless, expressing the trend slopes in relative terms influence the magnitude and thus can affect our interpretation of the results but has no influence on the significance and direction of the trends, as the statistical analyses (LMM, SMK and PMK) were done on the absolute value and were then transformed to facilitate the interpretation. Thus, the standardization of the slopes did not affect the statistical tests itself (carried out on the absolute values of DOC).

To address this issue of the absolute versus relative DOC trend slopes, we have added a new section in Supplementary Material with a more detailed discussion on it (pages 18-20 in Supplementary Material of the revised version).

Moreover, in order to provide all the information to the reader, we have added the median DOC concentration in the table reporting the trends in the revised manuscript (Table 1).
* * *
*4. Evaluating hundreds of time series may introduce random effects affecting the number of statistically significant trends. The theoretical and if possible quantitative implication of this (false positive and false negative trends) should be discussed.*

To check how robust the significant tests are we did an extra test (not included in the manuscript). We considered the trend tests to be significant at the 0.01 level, instead of 0.05, and compared the number of significant and non-significant trends (Table 1):

Table 1. Comparison significant positive, negative and non-significant at $p<0.05$ and $p<0.01$.

|          | Significant positive | Significant negative | Non-significant |
|----------|----------------------|----------------------|-----------------|
| p<0.05   | 91                   | 63                   | 104             |
| p<0.01   | 70                   | 50                   | 138             |

We can see that, at the 0.05 level, out of the 258 tests, we would expect to see a significance just by chance (type I error) in 13 cases, so about 6-7 in positive and about 6-7 in negative direction. As we detected many more trends in both direction, most of the significant results are not a type I error but are genuine effects. Moreover, the fact that we found many trends in both positive and negative direction implies that patterns vary across Europe, an argument that also stands if we test at the 0.01 level. Therefore, we believe that we can trust that the conclusions based on trends at 0.05 level are correct.

Finally, the results from the Linear Mixed Models (LMM), also presented in the manuscript (Table 1 in submitted manuscript), are not subject to this issue.

We have added this discussion in section 4.1.1.of the revised manuscript (lines 421-428).
* * *
*5. Using relative data in multivariate statistical models like GDA, may cause biased results strongly exaggerating the effects of trends in low DOC-concentration soil horizons. A discussion on the latter is missing.*

We decided to use relative data in the multivariate statistical models to be able to compare trends among sites and soil layers (see answer to comment 3). Although it is not ideal, soil layers had to be mixed to compute the multivariate statistical analysis due to lack of data. We acknowledged this limitation of our study in a section of the Supplementary Material (pages 13-14). However, it is true that a discussion on the effects of this choice is missing. This discussion has been added in Supplementary Material in the revised manuscript, with mention to the potential exaggeration of the effects of trends in low DOC-concentration soil horizons (page 18-20 in Supplementary Material of the revised version).
* * *
*6. Throughout the manuscript, the information gained from comments 1-3 should be commented where relevant. The information is especially important for the results and discussions dealing with the directions and controls on soil solution temporal trends (Chapters 3.1 and 4.2) partly based on the GDA and SEM results. Are the indicated effects quantitatively important, do they occur both at low and high DOC concentrations in soil solution and has the DOC concentration level any influence on the trend strength and direction?*

The controls on soil solution DOC trends have been discussed from a relative point of view, as we focused on explaining the high heterogeneity of DOC trends found across Europe, instead of the quantification of the trends at local scale. We have added an explanatory sentence at the end of Section 4.1.3 (lines 487-493 in revised manuscript), but, to avoid making the manuscript even longer, a more detailed explanation on the use of relative versus absolute slopes and the effect of DOC concentration levels has been added in Supplementary Material (pages 18-20). We decided to add this information in Supplementary Material because the main issue raised by referee number 2 was to synthesize the paper to clarify the message, and we also believe that adding more information to the main manuscript will make it more difficult to understand.
* * *
*7. The title of chapter 4.1.1 as well as some of the text are obscure. The number of non-significant trends is determined by the data and the statistical methods used. The authors themselves have selected data after quality check and chosen the statistical methods including probabilities to accept or reject trends. By speculating on whether the non-significant trends are real or not, the authors seem to reject their own data and methods? Change title and remove these speculations, but keep the general discussions on factors affecting trend analysis including what you have found related to comment 4 (see above).*

In this section, we do not aim to reject our own data and methods, but to discuss potential (and unavoidable) limitations of the methods selected, such as the length of the time series, or the strength of the trends in the time series of our dataset. However, we agree that this section needed to be reformulated and the title of chapter 4.1.1. has been changed to "Evaluation of the trend analysis techniques", the last paragraph has been deleted and the discussion mentioned in reply to comment 4 has been added in the revised manuscript (lines 421-428).
* * *
**Detailed comments**

*1. Lines 71-81: Riparian zones and peat lands, the most important DOC sources for surface waters are not referred to. Add some text and references.*

In the introduction, we focused on forest soils because our study deals only with forest soils. We believe that adding text and references on the importance of riparian zones and peat lands will just extend the introduction and probably interrupt the flow of the text.

*2. Line 205: ". . .more than 60 observations of soil solution DOC of individual or groups of collectors". What do the 60 observations refer to? Individual or groups of collectors? If the latter, was it pooled composite samples?*

As mentioned previously in the manuscript (line 164-165), in some countries, samples from these replicates were pooled before analyses or averaged prior to data transmission. Therefore, we selected time series with more than 60 observations from individual or groups of collectors, when the samples were pooled before analysis or prior to data transmission.

*3. Line 209: In Figure 1, the number is 436 time series instead of 529. Which figure is the correct one?*

Both 436 and 529 are correct numbers, but they refer to different stages of the study: 529 in the number of time series for the entire dataset and 436 is the number of time series after aggregating per plot-soil depth combinations. In the revised manuscript, we have changed the number in Figure 1 for consistency (from 436 to 529). Moreover, a table clearly explaining where the different numbers of time series come from has been added in Supplementary Material (Table S2, pages 11-12 in Supplementary Material), as it seemed to be confusing throughout the manuscript.

*4. Line 218: Did you use the same pH ranges for all soil horizons? If so, you may have a bias towards organic and upper mineral soil horizons in the Low pH class. Additionally it is not clear whether it is soil pH as stated in text or in soil solution as stated in Figure 9? Clarify!*

Yes, we used the same pH ranges for all the soil horizons and as a consequence we have more organic soil horizons in the Low pH class (Table 1).

Table 1. Number of cases at high and low pH classified by organic layer.

|     | High pH | Low pH |
| --- | --- | --- |
| O | 7 | 14 |
| M02 | 15 | 13 |
| M24 | 9 | 3 |
| M48 | 21 | 6 |
| M8 | 9 | 6 |

Since the manuscript contains too many figures (also suggested by referee number 2), we have moved Figure 9 from the manuscript. Even though we still find this result as potentially important, we think that the main message of the manuscript will be clearer by focusing only in the results from the SEM. Figure 9 was a supplementary figure to confirm that the response of DOC to environmental factors was a function of site deposition and acidification status.

Finally, sites were classified according to soil solution pH, and thus the revised text has been corrected (lines 224-225).

*5. Lines 219-222: From which time period do the S and N depositions originate? Is it median values or. . .?*

S and N deposition data covers the period 1999-2010 (Waldner et al., 2014). This have been mentioned in the text of the revised manuscript (lines 183-184). For our classification, we used mean values of deposition for this period (Lines 225-229).

- Waldner, P., Marchetto, A., Thimonier, A., Schmitt, M., Rogora, M., Granke, O., Mues, V., Hansen, K., Karlsson, G. P., Zlindra, D., Clarke, N., Verstraeten, A., Lazdins, A., Schimming, C., Iacoban, C., Lindroos, A. J., Vanguelova, E., Benham, S., Meesenburg, H., Nicolas, M., Kowalska, A., Apuhtin, V., Napa, U., Lachmanova, Z., Kristoefel, F., Bleeker, A., Ingerslev, M., Vesterdal, L., Molina, J., Fischer, U., Seidling, W., Jonard, M., O'Dea, P., Johnson, J., Fischer, R., and Lorenz, M.: Detection of temporal trends in atmospheric deposition of inorganic nitrogen and sulphate to forests in Europe, Atmos Environ, 95, 363-374, 2014.

*6. Line 276: Add p-value to ". . .overall positive trend. . .". p<0.05 or p<0.10??*

The p-value (p<0.05) has been added in the text referring to this overall positive trend in the organic layer.
* * *
*7. Lines 296-301 and Table 2. In the last sentence, it is stated that trends computed with SMK and PMK agreed well. However, at soil depth M24 the two methods results in very different rslopes (Table 2) both as regards directions and values. Comment on this and present a possible explanation.*

We have checked for these results and the difference in the rslopes between SMK and PMK originates from the difference in sites available for computing the SMK and the PMK. There are two extra sites for which SMK tests were performed, but not the PMK. These two extra sites show a positive trend (1.1 and 2 % yr -1), creating the difference in the median value at M24 between the two methods (Table 2 in submitted manuscript). However, when using exactly the same set of sites, the trend did not differ between the two methods.

Nevertheless, Table 2 was removed from the manuscript and the text dealing with the comparison between methods was moved to Supplementary Material, as it is a rather technical information. The above explanation has been commented in this new section of the Supplementary Material (page 17).

*8. Lines 309-311: nNS-trends=104, nP-trends=91 and nN-trends=63 makes up a total of 258 time series, which corresponds to the value in Supplementary materials. However, the number of monotonic trends is 191 according to Figure 1. Correct where appropriate.*

The numbers 258 and 191 correspond to different observations: 258 is the number of time series with less than 60 observations and more than 10 years, while 191 refers to the same time series, but after aggregation per plot-depth combination. This has been clarified in the revised Supplementary Material (Table S2).

*9. Lines 324-332 and Table 2: There are increased rslope values towards deeper mineral soil horizons. Is this a result of lower soil solution DOC concentrations (cf. general comment 3) and thereby very small DOC trends in absolute numbers? The rslope values in the O-horizon, generally showing high DOC concentrations, are close to those found for M8, indicating large DOC trends if statistically significant (N or P). Comment on this.*

We have checked the median values of the absolute DOC slopes and absolute trends in the organic horizons are indeed higher than in mineral soils (0.33 mg L$^{-1}$ yr$^{-1}$ for the organic layer versus 0.03 below 80 cm). This is a natural consequence of lower DOC concentrations in deep soils. This issue has been commented in the new section added to the Supplementary Material (pages 18-20), where also a table comparing relative versus absolute DOC trend slopes has been added (Table S3).

*10. Lines 360-361: ". . .we found evidence. . .". The rslope=f(mean TF SO4 deposition) relations in Figure 9 are no evidence, however, they show a relatively strong indication (r2=0.288) on that the SO4 deposition may tangibly affect the rslope values in acidic soils. Rephrase the sentence.*

In order to shorten the manuscript, Figure 9 has been removed from the manuscript and consequently the mentioned sentence has been deleted.
* * *
*11. Lines 367-372: Complement the GDA analysis with the DOC concentrations as an independent continuous variable and comment on the results. Is DOC concentration an important variable for explaining the variation (cf. general comment 3)?*

As we did not find a relationship between median DOC concentrations and absolute or relative slopes (see Figure S5 in the revised Supplementary Material), we did not see necessary to include DOC as a variable in the GDA. Introducing an additional variable would lower the robustness of the analysis. Moreover, we have removed the GDA analysis (Figure 10) from the manuscript to simplify the message, since results from the GDA are used just to support our findings from the SEM.
* * *
*12. Line 414: A bracket in front of Fig. 11A is missing.*

When re-writing the discussion, this paragraph has been deleted.
* * *
*13. Line 536: Again the total number of observations (n=174) does not match the number (n=191) stated in Figure 1. In the methods chapter, it may be wise to further explain the different number of observations occurring in different analysis and why so.*

is the total number of time series aggregated per plot and soil layer, 174 of this 191 time series show positive, negative or non-significant trends. The rest of plot/soil depth combinations (17) correspond to the plots that showed different trends (P, N or NS) in DOC within the same depth interval, which was the case for 17 plot-depth combinations (16 in Germany and one in Norway) (lines 312-315 in submitted manuscript).
* * *
*14. Lines 645-647: "We found evidence that soil pH determines the response of trends of DOC in soil solution to SO42- deposition. . .". This statement is not correct. What you have found is a relation between relative slopes of DOC and S-deposition in very acidic soils with a pH<4.2 in soil solution, but not in non-acid soils with a pH>5 (Figure 9). The multivariate analyses do not show that as stated. The relation in soils with 4.2_pH_5 is not shown or discussed in the text. Additionally, the statement refers to the entire soil column, but I suppose that the low pH in soil solution (pH<4.2) is primarily found in O-horizons and upper mineral soils. Rephrase the statement.*

We did not show the relation in soils with intermediate pH (between 4.2 and 5) to avoid introducing more information, as the manuscript is already dense in statistical analysis, number of figures and tables.

Regarding the fact that the statement refers to the entire soil column, we could not do the statistical analysis separately for the different soil layers due to a lack of data (see explanation in Supplementary Material, pages 13-14). Therefore, to check the influence of mixing soil layers, we re-did the SEMs models (Figure 6 in the revised manuscript) with horizon type (organic versus mineral) as an explanatory variable. For the model obtained for 1) all the cases, 2) for low and medium nitrogen deposition and 3) for high nitrogen deposition (Figure 6A, 6B and 6C, respectively), the variable

"depth" (organic vs mineral) was not significantly correlated with DOC slopes (p =0.85, p=0.34 and p=0.56). Based on this test, horizon type does not appear to play a role in explaining the differences between the trend slopes of DOC and, thus we trusted the findings from the SEMs even when mixing soil layers.

The statement "We found evidence that soil pH determines the response of trends of DOC in soil solution to $SO_4^{2-}$ deposition. . ." has been removed from the conclusions in the revised manuscript.
* * *
*15. In the conclusions, I would suggest that you stress the large local variation related to a multitude of factors and discuss the regional processes in a more humble way, supported by your results. I also suggest that you describe the differences found between DOC trends in organic and mineral soil layers and possible influences by different drivers/processes. Finally, if there is any relation between DOC concentration level and DOC trends (levels or directions), this should be stressed.*

Part of the conclusion has been rephrased to stress the differences found between DOC trends in organic and mineral soil layers, however, we cannot describe the influences by different drivers/processes based on our results without being too speculative for a conclusion. We think that the importance of the local variation related to several factors is discussed in the last sentence. Finally, there was no finding regarding the relation between DOC concentration level and DOC trends that should be stressed in the conclusion.
* * *
**Comments on tables and figures**
*1. Table 1: In the legend, information on how 0.05_p_0.1 is indicated is missing (italics?). In the table, there is a mess among grey, bold and italic figures. Related to the SMK results, the number zero is sometimes missing.*
The legend of Figure 1 has been completed and the table corrected.

*2. Table 2:The different statistical methods do not always show the same direction on rslope for all soil layers (BFAST M02 and PMK M24 are negative). This should be commented on in the text.*
Table 2 was removed from the manuscript, but this issue has been commented on in the Supplementary Material (see detailed comment 7)

*3. Table 3. Change name on slope to rslope in column headings and explain in legend. Which year(s) do the S and N depositions data refer to?*
Table 3 has been corrected accordingly.

*4. Table 4, Legend: What do you mean with ". . .during the last years. . ."? Explain.*
Table 4 has been removed from the manuscript to shorten the paper.

*5. Figure 2: Weight_P is missing on the X-axis*
Figure 2 has been corrected and moved to Supplementary Material (now Figure S4).

*6. Figure 3, legend: Explain boxplots (c.f. Figure 6) and "n" in figure.*
Figure 3 has been removed from the manuscript to shorten the paper.

*7. Figure 7: Defining that the trends refer to DOC is missing in the legend. The Y-axis is too short in Figure 7C and perhaps also in the others. Maximum values on the Y-axis seem to be very close to the observed maximum numbers.*
Figure 7 has been modified in the revised version and corrected where appropriate.

*8. Figure 8. Define whether it is natural logarithms or 10-logarithms on the X-axis. The X-axis is too short in Figure 8B.*

Figure 8 has been removed from the manuscript to shorten the paper.

*9. Figure 9: In the legend, define which soil layers the data points refer to.*
Figure 9 has also been removed from the manuscript.

*10. Figure 10: Use the same scales on the XY-axes in Figure A and B.*
Figure 10 has been removed from the manuscript.

*11. Figure 11: In the legend change from (>15 kg N ha-1 yr-1) to (<15 kg N ha-1 yr-1)*
Figure 11 has been corrected.

**Comments on Supplementary material**
*1. S2: For the GDA analysis, it is unclear whether the "Weighed positive" and "Weighed negative" trends are included. Clarify.*
The GDA analysis has been omitted in the revised manuscript (see response to Detailed comment 11).
* * *
*2. S2: For the SEM analysis, it is unclear whether the analyses are performed on SEN slopes or rslopes. Clarify.*
The SEM analyses were performed on the relative slopes (rslopes): this has been clarified in the revised version.
* * *
*3. Figure S1: The legend box hides some bars.*

Figure S1 has been re-done.

**Answers to Referee 2**

*Referee 2: However, saying that this article is not very clear is litotes. The abstract and introduction are rather well written. However, the result and discussion sections are extremely hard to read, the number of figures and tables is not tenable (I counted 27 items, figures+tables) and the usefulness of numerous statistical analyses is not convincing since they provide similar results and conclusions. I think that an effort of synthesis is necessary to simplify messages and prevent a dilution of important results with accessory observations. For example, at the end of the reading of your manuscript, I was not able to say whether sulphate depositions increase or decrease soil solution DOC content. In your abstract, it is suggested that DOC concentrations and sulphate depositions are positively linked, a statement which is then contradicted in the manuscript (e.g. L412-413 but L348-349).*

We agree that the manuscript includes many statistical approaches and we understand that this might confuse the reader. This approach diversity comes from the large community of scientists involved in the study by providing data and scientific input. Concerning the temporal analyses of DOC concentrations, we decided to show results from the different statistical methods (LMM, seasonal and partial Mann Kendall) because the approaches are complementary. Each method has pros and cons. This allowed us to show that DOC concentrations have increased during the observation period overall in coniferous forests in the organic layers. However, at individual plots and depths, DOC concentrations did not show any consistent temporal trend (increase, decrease or no change). We could also show that there was no geographical pattern either.

Therefore, we have shorten the manuscript mainly by synthesizing the multivariate analyses. We have removed four figures (Fig. 3, Fig 8, Fig. 9, Fig. 10), parts of Figure 7 (7A, 7B), and one table (Table 4). Moreover, we have shorten the most technical part of the results by removing Table 2 and Fig. 2. This has not changed the conclusions of the manuscript, but made the discussion more concise. In the revised manuscript, in Figure 7 (now Figure 5), we focused only on the relationships between the temporal trends of DOC concentrations, forest types and classes of stem volume increment (proxy for forest productivity). The relationships between trends in DOC concentrations and SO4 and NO3 throughfall deposition are now explored only in Figure 6 to avoid confusion. The comparison among trend analysis techniques is now in the Supplementary Material of the revised manuscript. We believe that this reduction of the number of tables and figures will improve the readability of the manuscript.
* * *
*Referee 2: Moreover, it would be useful to create a figure summarizing the main chemical reactions and biological processes controlling soil solution DOC content*

Our results come from an exploratory statistical approach (and not deterministic) of a large European dataset and we are afraid that it would be preposterous at this stage to build a model based on such a variety of local (e.g. soil properties) and regional (e.g. atmospheric deposition) factors. The only alternative we can think of is to include a figure based on the numerous mechanistic models proposed in the literature, but it will increase the number of figures and will not provide with novel information.
* * *
*Referee 2: More fundamentally, I am not sure that the measurement of soil solution DOC provides an accurate estimate of the amount of DOC flowing out of terrestrial ecosystems (and supply of DOC to surface water). The leaching of DOC happens at specific moments of the year depending on hydric balance (precipitationevapotranspiration), soil type, plant activity etc. I am even sure that the DOC soil solution concentration can be inversely related to DOC leaching in some conditions. Just an example: soil solution DOC concentration is higher in summer than in winter, but DOC leaching only occurs in winter in France. This issue could explain why the present study fails to show clear overall trend in soil solution DOC at individual plot and soil depths. A warming-induced change of ecosystem water balance could also contribute to changes in DOC content in soil solution and surface water. Therefore, I suggest to present (in manuscript or in supplementary materials) the volume of water*

*harvested in lysimeters or calculations of theoretical water balance (precipication-evapotranspiration).*

We agree with the referee that we did not assess DOC fluxes flowing out from forest soils.
Our time series analysis aimed to detect long-term changes of DOC concentrations that are not due to seasonal effects or dilution-concentration effects caused by fluctuations in soil water content. Therefore, we decided to apply both Seasonal Mann Kendall and Partial Mann Kendall using precipitation as a co-variable to remove the seasonality and dilution-concentration effects. This method allowed us to detect significant monotonic changes (increase or decrease) of DOC concentrations over a period of 10 years at least. Studies having shown temporal changes of DOC in surface waters also reported concentrations rather than fluxes (e.g. Worral & Burt 2004, Evans et al. 2005, Monteith et al. 2007, Dawson et al. 2009, all cited in the manuscript).

Using water volume collected by lysimeters to assess water fluxes was not possible, because these data were available since 2011 only. In addition, the volume collected by tension lysimeters depends on the suction applied to the system.
To assess water fluxes at different soil depths, we would need to model the water balance at 118 forest sites, which is very challenging, since many input parameters (meteorology, soil, vegetation) would be required. This was beyond the scope of this study. A simple estimate based on the difference between precipitation and evapotranspiration, would add a substantial uncertainty to the calculation of DOC fluxes and therefore detecting long-term changes of DOC fluxes would be even more difficult. Because of large variations in soil water fluxes (e.g. Borken et al. 2011, Verstraeten et al. 2014, Meesenburg et al. 2016), it is more difficult to detect long-term trends of fluxes than long-term trends of concentrations in soil solution. Since most times series of DOC concentrations in soil solution do not indicate any long-term trend in our dataset, the chance of finding long-term changes in DOC fluxes are even lower.

- Borken, W., Ahrens, B., Schulz, C., & Zimmermann, L. (2011). Site-to-site variability and temporal trends of DOC concentrations and fluxes in temperate forest soils. *Global change biology*, *17*(7), 2428-2443.
- Meesenburg, H., Ahrends, B., Fleck, S., Wagner, M., Fortmann, H., Scheler, B., ... & Meiwes, K. J. (2016). Long-term changes of ecosystem services at Solling, Germany: Recovery from acidification, but increasing nitrogen saturation?. *Ecological Indicators, 65*, 103–112
- Verstraeten, A., De Vos, B., Neirynck, J., Roskams, P., & Hens, M. (2014). Impact of air-borne or canopy-derived dissolved organic carbon (DOC) on forest soil solution DOC in Flanders, Belgium. Atmospheric Environment, 83, 155-165
* * *
*Referee 2: I am not really convinced by the relevance of removing the breakpoints. These breakpoints are not necessary the result of site disturbances (change of sensors etc) but could result from sudden change of atmospheric chemical composition or ecosystem functionning.*

Monotonicity of time series is generally assumed when analyzing DOC data for temporal trends (Filella and Rodriguez-Murillo, 2014). However, it is rarely statistically tested and, thus, potential abrupt changes in the time series may be overlooked. This issue becomes important in temporal trend analysis since a breakpoint may cause changes in the direction of the trend and could lead us, for example, to classify a time series as constant, when in reality we may have averaged out separate periods with significant changes (de Jong et al., 2013). On the other hand, breakpoints may erroneously induce the detection of a significant trend in long-term time series due to artifacts (see Supplementary Material).The aim of our study is to analyze monotonic trends related to factors that have been measured within the ICP Forests database. Therefore, DOC time series were first analyzed using the Breaks For Additive Seasonal and Trend (BFAST) algorithm to detect the presence of breakpoints.

We agree that removing breakpoints using the BFAST technique may remove time series that show abrupt changes not only due to artifacts (collector replacement, etc.,), but also due to natural causes (meteorological conditions, extreme events), forest management (changes in soil condition, thinning, etc.), sudden change of atmospheric chemical composition or ecosystem functionning. Nevertheless, many breakpoints are the consequence of technical issues or even inconsistencies in the database. The ICP Forests soil solution dataset has a great potential for analysis of large scale trends, but at the same time it may also contains some inconsistencies. The BFAST analysis proved to be effective at removing breakpoints caused by some dataset errors and thus the most defective time series were left out.

Although the investigation of the potential causes of the abrupt changes (breakpoints) in the individual time series can indeed provide a very valuable information, we do not count with the site-level information necessary for that purpose. To attribute the different causes of the breakpoints at site scale, we would need information of the management and climate history at each particular site, which is not available at the time of writing the present manuscript. Consequently, we cannot be sure of the origin of each breakpoint, and thus we decided to leave out all the time series showing abrupt changes to avoid erroneous detections of significant trends. Moreover, in this way, we are confident that the trends found in the time series that we analyzed are not a consequence of local factors. The alternative is to study each time series individually to identify the local (or regional) factors causing abrupt changes at plot scale, and this task is beyond the scope of this study, but is a very interesting topic to be addressed in a follow-up paper.

- Filella, M. and Rodriguez-Murillo, J. C.: Long-term Trends of Organic Carbon Concentrations in Freshwaters: Strengths and Weaknesses of Existing Evidence, Water-Sui, 6, 1360-1418, 2014.
- de Jong, R., Verbesselt, J., Zeileis, A., and Schaepman, M. E.: Shifts in Global Vegetation Activity Trends, Remote Sens-Basel, 5, 1117-1133, 2013.
* * *
*Referee 2: The terminology used in the manuscript is often not clear. The term "trend" is vague and does not specifically refer to change with time. The terms "trend slope", "trend direction" and "relative trend slope" are even more difficult to understand.*

To avoid any confusion about the term "trend", we have added a sentence in the Method section explaining the different temporal components of time series analyses (lines 231-232 in the revised manuscript).

We have also added some explanations on the terms "trend slope" and "relative trend slope" in the Method section (lines 268-270 in the revised manuscript). The term "trend direction" now appear only once in the manuscript and it has been clarified (line 316 in the revised manuscript).
* * *
*Referee 2: The terms "depositions" and "troughfall" are interchangeably used, I suggest you to use only one of the two terms.*

We used throughfall deposition of sulfate and inorganic nitrogen, except in Table 3 where bulk deposition was provided for three sites because throughfall deposition was not available. This was marked by an asterisk in the table. To avoid any confusion, we have reviewed the whole manuscript and used only "throughfall deposition" or "deposition" in the text.
* * *
*Referee 2: The term "fertile soil" is weak and, as usual, does not refer to measurable variable. The fact that tree growth is high does not necessary mean that the soil is fertile. The tree growth is often linked to forest dynamics and age (tree growth of old forests is typically slow irrespective of soil*

*characteristics; tree growth after forest disturbance (drought events, storm etc) is typically high because tree mortality allows the recruitment of seedling with fast growth rate).*

We agree with the referee that tree growth is not necessarily related to soil fertility. Consequently, we have reformulated chapter 4.2.1. to show that this chapter aims to discuss the relationship between forest productivity (in our case only stem growth is available as a proxy for forest productivity) and DOC in soil solution. A number of other factors such as climate, soil water availability, soil fertility, tree age and competition between neighboring trees can influence tree growth too. We have shorten this section and clearly speculated about the relationship between soil fertility and DOC in soil solution. This is an interesting topic to be investigated in the future, but it is beyond the scope of this paper.

**Most relevant changes made in the manuscript**

1. The number of Tables and Figures has been reduced to improve clarity of the manuscript: Tables 2 and 4 and Fig. 2, Fig, 3, Fig. 7A, 7B, Fig 8, Fig. 9 and Fig. 10 has been deleted.

2. The comparison between methods (Section 3.2.1.) has been moved to Supplementary Material: Fig. 2 is now Fig. S4 (pages 16-17 of the Supplementary Material).

3. New tables have been added to the Supplementary Material with the median values, 25- and 75-percentiles and number of observations of DOC, and other water chemical parameters: pH, conductivity, Ca, Mg, $SO_4^{2-}$, $NO_3^-$, $Al_3^+$(Tables S4 to S11, pages 21-27 of the Supplementary Material).

3. A new section has been added to Supplementary Material titled: "Implications of using standardized DOC slopes versus absolute DOC slopes.". In this section, issues raised by reviewer number 1 concerning the use of the relative (standardized) slope and the effect of the median DOC on the DOC trends are discussed (pages 18-20 of the Supplementary Material). A new table (Table S3) and figure (Fig. S5) has been added to support this discussion.

4. Clarification of the number of time series used in the study: a table summarizing the number of time series used has been added to Supplementary material (Table S2) and two sentences has been added in the manuscript (lines 247-248 and 264-265).

5. The entire discussion has been rewritten to improve clarity, and more specifically:

5.1. Title and content of Section 4.1.1., discussing the trend detection methodology, has been changed. We have added a discussion on the potential multiple testing effect (lines 421-428).

5.2. Section 4.2.1, dealing with the discussion of the effect of stem growth on the DOC trends has been reformulated (lines 496-521).

6. The text and figures has been corrected according to the detailed comments from reviewer number 1.

7. The term "trend", "trend slope" and "relative trend slope" has been clarified in the text (lines 231-232, 268-270).

[revised manuscript text omitted]

There was a good agreement between results using the three methods: BFAST, SMK, and PMK. The direction and significance of the trend agreed for 84.5% of the time series analyzed. For the majority of the remaining time series for which the trends did not agree, BFAST did not detect a trend when SMK and PMK did, thus, the latter two methods seemed more sensitive for trend detection than BFAST. Trends computed with SMK and PMK agreed well. The direction of the trend for SMK and PMK only differed for the intermediate soil layer (20-40 cm), as a result of the two extra sites for which SMK tests were performed, but not the PMK, that showed a marked positive trend (1.1 and 2 % yr$^{-1}$). However, when using exactly the same set of sites, the trend did not differ between the two methods.

**Implications of using standardized DOC slopes versus absolute DOC slopes.**

The standardized (relative) slopes of DOC concentrations were used for the study of the factors affecting the soil solution DOC trends (Fig. 5 and 6). The main reason for this decision was that using the median DOC concentration as a reference (as we did with the standardization) allowed us to determine whether the absolute trend in DOC was quantitatively large or not from an ecological perspective, because the absolute trend slope will be highly dependent on the initial DOC concentrations of the site.

The absolute trend slopes show the real magnitude and significance of the trend, but do not allow for comparison among sites or horizons. Since the aim of this study is to test whether there is a general DOC trend and to compare sites across Europe, we decided that using the relative slope was more consistent.

Moreover, due to limitations of the statistical analysis, we worked with time series per "plot-soil depth combinations", which means that different soil layers were mixed in the statistical analysis. Again, the standardization of the slopes of DOC concentrations allowed us to compare trends among different soil horizons by removing the effect of the decreasing soil solution DOC concentrations with soil depth. Otherwise, using the absolute trends would introduce a bias when we try to explain the DOC trends in relation with other parameters, because the trend slope would be highly dependent on the actual DOC concentrations, which, in turn, are very variable, not only among sites, but also among soil depths.

The influence of the DOC concentration levels was checked before deciding to use the standardized slopes (Fig. S5). It seemed that there was no relationship between the DOC trend slopes (relative and absolute) and the median DOC concentrations, with positive and negative trends occurring at both low and high DOC concentrations and, thus, we decided that using the standardized slopes will not hide any effect of the median DOC concentrations on the direction of the DOC trends.

This decision, however, has a drawback: the strength of the trend is clearly influenced by the DOC concentration levels. The fact that we used the standardized slope of DOC implied that it may be identical for two sites with very different mean DOC concentrations. DOC concentration decreases with depth and is lower in the deep mineral soil than in the upper mineral soil (Table S3) and by standardizing the slope, the magnitude of the trend was exaggerated in lower soil layers where both the absolute slope of DOC and the median DOC

concentration are low (Table S3). This issue is well illustrated in Fig. S5, that shows how the highest standardized slopes are usually at low DOC concentrations (mostly in mineral soil layers), while the highest absolute slopes are at higher DOC concentrations (mostly in organic and upper soil layers).

In other words, in quantitative terms DOC trends are much higher in the organic layer than in the mineral soils but, in relative terms, DOC is increasing in the same proportion (Table S3). Because the aim of this study is to explain the high heterogeneity of DOC trends found across Europe, instead of the quantification of the trends at local scale, the relative trends were discussed throughout the manuscript. Consequently, our results should be interpreted with caution, keeping in mind that the relations between DOC trends and explaining factors are discussed only from a relative point of view.

Nevertheless, the statistical analyses (LMM, SMK, PMK and BFAST) were done on the absolute value and the resulting Sen's slopes were then standardized. Thus, the fact that trends are expressed in relative terms has consequences on the interpretation of the results, but has no influence on the statistical test itself (carried out on the absolute values of DOC), that is, on the significance and direction of the trends.

Table S3. Comparison of median relative trend slope (rslope in % yr-1) and absolute trend slope (abs slope in mg $L^{-1}$ $yr^{-1}$) of DOC concentrations in soil solution and their interquartile range using the Seasonal Mann-Kendall test (SMK). (O: organic layer, M02: mineral soil 0-20 cm, M24: mineral soil 20-40 cm, M48: mineral soil 40-80 cm, M8: mineral soil > 80 cm.)

| Soil depth | rslope (% $yr^{-1}$) | abs slope (mg $L^{-1}$ $yr^{-1}$) |
|---|---|---|
| O | 1.18 (±3.37) | 0.32 (±1.2) |
| M02 | 0.04 (±3.41) | 0.008 (±0.52) |
| M24 | 0.61 (±8.62) | 0.025 (±0.48) |
| M48 | 1.01 (±4.79) | 0.013 (±0.22) |
| M8 | 1.18 (±9.39) | 0.032 (±0.31) |

[Figure]

Figure S5. A) Standardized trends (relative DOC slope) versus median DOC concentrations. B) Absolute trends (absolute slope DOC) versus median DOC concentrations. The different colors represent the different soil layers.

**Supplementary material S4. Structural equation model with trends in $SO_4^{2-}$ and $NO_3^-$ deposition**

The same structural equation models (SEM) represented in Figure 11 were performed using the trends in $SO_4^{2-}$ and $NO_3^-$ deposition (% $yr^{-1}$) instead of the mean values of $SO_4^{2-}$ and $NO_3^-$ throughfall deposition (kg $ha^{-1}$ $yr^{-1}$) (Figure S3). The SEMs for all the cases and for cases with low and medium N deposition are shown in Figure S3.

[Figure]

Figure S3. Diagram of the structural equation model (SEM) that best explains the maximum variance of the resulting trends of DOC concentrations in soil solution for: A) all the cases and B) cases with low or medium N deposition, with trends in $SO_4^{2-}$ and $NO_3^-$ deposition (% $yr^{-1}$) with direct effects and indirect effects through effects on mean annual stem volume increment (growth) in $m^3$ $ha^{-1}$ $yr^{-1}$). P values of the significance of the corresponding effect between brackets. Green arrows indicate positive effects and red arrows indicate negative effects.**Information on the soil solution chemistry at the studied ICP Forests Level II plots**

Table S4. Median soil solution DOC concentrations (mg L$^{-1}$), 25% and 75% percentiles and number of observations (n) for the different forest types, soil depth intervals and collector types with the entire dataset (with breakpoints) and with the dataset without time series showing breakpoints (without breakpoints).

| | | WITH BREAKPOINTS | | | | WITHOUT BREAKPOINTS | | | |
|---|---|---|---|---|---|---|---|---|---|
| | | median [DOC] | 25% percentile | 75% percentile | n | median [DOC] | 25% percentile | 75% percentile | n |
| Broadleaved | | | | | | | | | |
| TL | O | 41.35 | 28.99 | 56.05 | 637 | 44.56 | 32.00 | 59.10 | 475 |
| | M02 | 8.80 | 4.30 | 21.20 | 8397 | 8.68 | 4.50 | 23.50 | 3104 |
| | M24 | 3.78 | 1.67 | 8.90 | 2584 | 3.19 | 1.85 | 4.76 | 928 |
| | M48 | 2.60 | 1.10 | 6.40 | 10635 | 2.70 | 1.08 | 5.80 | 4634 |
| | M8 | 2.60 | 1.17 | 6.53 | 4354 | 2.65 | 1.53 | 7.00 | 1797 |
| ZTL | O | 33.33 | 21.00 | 51.12 | 4057 | 30.88 | 18.01 | 51.10 | 1956 |
| | M02 | 4.26 | 3.51 | 6.28 | 608 | 4.30 | 2.80 | 9.30 | 192 |
| | M24 | 20.44 | 13.40 | 34.37 | 94 | | | | 0 |
| | M48 | 3.42 | 2.61 | 4.51 | 427 | 0.91 | 0.50 | 1.64 | 85 |
| | M8 | 2.42 | 2.11 | 3.62 | 34 | | | | 0 |
| Coniferous | | | | | | | | | |
| TL | O | 49.00 | 35.10 | 67.36 | 2496 | 50.90 | 38.20 | 65.40 | 693 |
| | M02 | 15.70 | 7.09 | 31.15 | 10914 | 12.80 | 5.90 | 25.50 | 5813 |
| | M24 | 5.72 | 2.40 | 16.50 | 5116 | 5.00 | 2.10 | 21.89 | 2476 |
| | M48 | 4.44 | 2.30 | 11.40 | 13979 | 4.30 | 2.29 | 10.90 | 6431 |
| | M8 | 3.70 | 1.60 | 7.91 | 5024 | 4.29 | 2.55 | 10.12 | 1597 |
| ZTL | O | 42.92 | 29.03 | 60.80 | 4079 | 44.60 | 30.18 | 60.80 | 2703 |
| | M02 | 36.90 | 22.20 | 56.40 | 2781 | 36.00 | 24.00 | 53.00 | 253 |
| | M24 | 16.34 | 8.76 | 31.59 | 645 | | | | 0 |
| | M48 | 44.00 | 17.40 | 62.35 | 227 | 13.70 | 10.30 | 36.25 | 251 |
| | M8 | 4.14 | 3.28 | 4.81 | 84 | | | | 0 |

Table S5. Median soil solution pH, 25% and 75% percentiles and number of observations (n) for the different forest types, soil depth intervals and collector types with the entire dataset (with breakpoints) and with the dataset without time series showing breakpoints (without breakpoints).

| | | WITH BREAKPOINTS | | | | WITHOUT BREAKPOINTS | | | |
|---|---|---|---|---|---|---|---|---|---|
| - | - | median pH | 25% percentile | 75% percentile | n | median pH | 25% percentile | 75% percentile | n |
| Broadleaved | - | - | - | - | - | - | - | - | - |
| TL | O | 3.9 | 3.8 | 4.1 | 636 | 3.90 | 3.80 | 4.10 | 518 |
| - | M02 | 4.5 | 4.2 | 5.2 | 8346 | 4.60 | 4.20 | 6.2 | 3322 |
| - | M24 | 6.3 | 4.9 | 7.1 | 2482 | 6.10 | 4.90 | 6.7 | 993 |
| - | M48 | 5.1 | 4.5 | 6.7 | 10496 | 5.10 | 4.40 | 6.5 | 5162 |
| - | M8 | 6.4 | 4.6 | 7.8 | 4228 | 4.50 | 4.30 | 6.46 | 2115 |
| ZTL | O | 5.30 | 4.40 | 6.30 | 4026 | 5.30 | 4.30 | 6.60 | 2025 |
| - | M02 | 6.15 | 5.00 | 7.6 | 608 | 5.00 | 4.80 | 5.75 | 227 |
| - | M24 | 4.70 | 4.50 | 5 | 93 | 0.00 | 0.00 | 0 | 0 |
| - | M48 | 8.30 | 8.20 | 8.4 | 426 | 5.20 | 5.10 | 5.3 | 108 |
| - | M8 | 8.20 | 8.00 | 8.3 | 34 | 0.00 | 0.00 | 0 | 0 |
| Coniferous | - | - | - | - | - | - | - | - | - |
| TL | O | 4.00 | 3.80 | 4.40 | 2496 | 3.80 | 3.60 | 4.00 | 726 |
| - | M02 | 4.30 | 4.00 | 4.7 | 10634 | 4.30 | 4.00 | 4.7 | 6930 |
| - | M24 | 4.60 | 4.30 | 5 | 4739 | 4.60 | 4.30 | 4.8 | 2849 |
| - | M48 | 4.50 | 4.30 | 4.9 | 13596 | 4.50 | 4.20 | 4.9 | 7462 |
| - | M8 | 4.57 | 4.30 | 6.4 | 4837 | 4.48 | 4.29 | 4.7 | 1660 |
| ZTL | O | 4.02 | 3.80 | 4.60 | 4038 | 4.00 | 3.80 | 4.80 | 2839 |
| - | M02 | 4.40 | 4.10 | 4.9 | 2412 | 4.80 | 4.53 | 5.3 | 254 |
| - | M24 | 4.90 | 4.50 | 5.4 | 551 | 0.00 | 0.00 | 0 | 0 |
| - | M48 | 4.80 | 4.10 | 5.1 | 225 | 4.40 | 4.27 | 4.9 | 319 |
| - | M8 | 4.70 | 4.60 | 4.8 | 84 | 0.00 | 0.00 | 0 | 0 |

Table S6. Median soil solution conductivity ($\mu S\ cm^{-1}$), 25% and 75% percentiles and number of observations (n) for the different forest types, soil depth intervals and collector types with the entire dataset (with breakpoints) and with the dataset without time series showing breakpoints (without breakpoints).

| | | WITH BREAKPOINTS | | | | WITHOUT BREAKPOINTS | | | |
|---|---|---|---|---|---|---|---|---|---|
| - | - | median COND | 25% percentile | 75% percentile | n | median COND | 25% percentile | 75% percentile | n |
| Broadleaved | - | - | - | - | - | - | - | - | - |
| TL | O | 128.00 | 93.50 | 189.50 | 631 | 140.00 | 103.00 | 212.50 | 507 |
| - | M02 | 60.00 | 42.25 | 99 | 7651 | 69.55 | 45.00 | 104 | 3066 |
| - | M24 | 86.00 | 47.00 | 180 | 1503 | 70.45 | 45.90 | 120 | 548 |
| - | M48 | 68.00 | 45.00 | 137 | 8538 | 70.00 | 48.58 | 145 | 4320 |

| | | median | 25% | 75% | n | median | 25% | 75% | n |
|---|---|---|---|---|---|---|---|---|---|
| - | M8 | 148.50 | 61.63 | 305.75 | 3006 | 133.00 | 59.00 | 210 | 1736 |
| ZTL | O | 71.00 | 48.00 | 110.00 | 2750 | 70.00 | 46.60 | 111.00 | 1489 |
| - | M02 | 63.35 | 34.00 | 86.775 | 608 | 28.20 | 19.10 | 51.05 | 227 |
| - | M24 | 44.00 | 28.00 | 56 | 93 | 0.00 | 0.00 | 0 | 0 |
| - | M48 | 282.00 | 254.00 | 318 | 425 | 19.30 | 16.38 | 25.325 | 108 |
| - | M8 | 485.50 | 446.50 | 539.75 | 34 | 0.00 | 0.00 | 0 | 0 |
| Coniferous | - | - | - | - | - | - | - | - | - |
| TL | O | 77.00 | 56.00 | 124.00 | 2425 | 85.00 | 65.00 | 155.00 | 725 |
| - | M02 | 58.00 | 31.00 | 92 | 9222 | 61.00 | 33.00 | 105.5 | 5699 |
| - | M24 | 50.00 | 30.00 | 97 | 2954 | 56.00 | 31.00 | 111 | 1715 |
| - | M48 | 56.00 | 37.00 | 94 | 10270 | 56.00 | 37.20 | 99 | 6658 |
| - | M8 | 104.00 | 55.00 | 207.75 | 2850 | 120.50 | 66.00 | 259 | 1118 |
| ZTL | O | 65.30 | 45.00 | 104.00 | 2296 | 64.00 | 42.30 | 106.00 | 1537 |
| - | M02 | 39.20 | 25.00 | 59 | 2627 | 27.00 | 20.08 | 41.1 | 228 |
| - | M24 | 32.00 | 21.00 | 57.95 | 615 | 0.00 | 0.00 | 0 | 0 |
| - | M48 | 39.05 | 28.00 | 150.5 | 214 | 95.85 | 46.48 | 155.5 | 290 |
| - | M8 | 50.00 | 31.75 | 69.25 | 84 | 0.00 | 0.00 | 0 | 0 |

Table S7. Median soil solution Ca (mg L$^{-1}$), 25% and 75% percentiles and number of observations (n) for the different forest types, soil depth intervals and collector types with the entire dataset (with breakpoints) and with the dataset without time series showing breakpoints (without breakpoints).

| | | WITH BREAKPOINTS | | | | WITHOUT BREAKPOINTS | | | |
|---|---|---|---|---|---|---|---|---|---|
| - | - | median [Ca] | 25% percentile | 75% percentile | n | median [Ca] | 25% percentile | 75% percentile | n |
| Broadleaved | - | - | - | - | - | - | - | - | - |
| TL | O | 4.18 | 1.83 | 7.85 | 633 | 5.369 | 3.193 | 9.204 | 515 |
| - | M02 | 2.12 | 0.80 | 5.3 | 8381 | 2.80 | 1.04 | 9.56525 | 3396 |
| - | M24 | 4.09 | 1.50 | 14.18 | 2555 | 3.69 | 0.92 | 9.005 | 999 |
| - | M48 | 2.31 | 0.70 | 9.385 | 10600 | 2.80 | 0.92 | 7.7 | 5204 |
| - | M8 | 5.68 | 1.50 | 41.7825 | 4322 | 2.80 | 0.51 | 13.75 | 2151 |
| ZTL | O | 4.10 | 2.05 | 7.06 | 4049 | 3.90 | 1.40 | 6.36 | 2030 |
| - | M02 | 8.33 | 1.67 | 13.59 | 608 | 1.23 | 0.75 | 2.425 | 227 |
| - | M24 | 2.35 | 1.25 | 3.296 | 94 | 0.00 | 0.00 | 0 | 0 |
| - | M48 | 58.86 | 51.26 | 67.485 | 419 | 0.72 | 0.58 | 1.06 | 108 |
| - | M8 | 73.75 | 60.78 | 92.8 | 34 | 0.00 | 0.00 | 0 | 0 |

| | | median | 25% | 75% | n | median | 25% | 75% | n |
|---|---|---|---|---|---|---|---|---|---|
| Coniferous | _ | _ | _ | _ | _ | _ | _ | _ | _ |
| TL | O | 3.36 | 1.47 | 6.39 | 2490 | 1.55 | 0.98 | 3.66 | 722 |
| _ | M02 | 0.66 | 0.25 | 1.72 | 10890 | 1.00 | 0.36 | 2.45 | 6985 |
| _ | M24 | 0.82 | 0.30 | 1.8665 | 5079 | 0.90 | 0.30 | 1.61 | 2901 |
| _ | M48 | 0.82 | 0.32 | 2.07 | 13901 | 0.92 | 0.32 | 2.285 | 7511 |
| _ | M8 | 2.10 | 0.49 | 10.6575 | 4986 | 1.97 | 0.53 | 8.285 | 1700 |
| ZTL | O | 1.50 | 0.72 | 2.80 | 4052 | 1.50 | 0.72 | 2.80 | 4052 |
| _ | M02 | 1.13 | 0.53 | 2.14 | 2777 | 1.13 | 0.53 | 2.14 | 2777 |
| _ | M24 | 1.20 | 0.62 | 2.31 | 644 | 1.20 | 0.62 | 2.31 | 644 |
| _ | M48 | 3.00 | 1.81 | 3.895 | 227 | 3.00 | 1.81 | 3.895 | 227 |
| _ | M8 | 0.76 | 0.47 | 1.1975 | 84 | 0.76 | 0.47 | 1.1975 | 84 |

Table S8. Median soil solution Mg (mg L$^{-1}$), 25% and 75% percentiles and number of observations (n) for the different forest types, soil depth intervals and collector types with the entire dataset (with breakpoints) and with the dataset without time series showing breakpoints (without breakpoints).

| _ | _ | WITH BREAKPOINTS | | | | WITHOUT BREAKPOINTS | | | |
|---|---|---|---|---|---|---|---|---|---|
| _ | _ | median [Mg] | 25% percentile | 75% percentile | n | median [Mg] | 25% percentile | 75% percentile | n |
| Broadleaved | _ | _ | _ | _ | _ | _ | _ | _ | _ |
| TL | O | 1.05 | 0.48 | 1.90 | 633 | 1.18 | 0.62 | 2.08 | 515 |
| _ | M02 | 0.80 | 0.42 | 1.5 | 8382 | 0.86 | 0.51 | 1.46 | 3395 |
| _ | M24 | 1.01 | 0.50 | 2.13 | 2563 | 1.18 | 0.62 | 2.295 | 999 |
| _ | M48 | 0.95 | 0.37 | 2.0745 | 10611 | 1.02 | 0.46 | 2.19 | 5205 |
| _ | M8 | 1.72 | 0.73 | 3.94 | 4323 | 1.29 | 0.51 | 2.88 | 2152 |
| ZTL | O | 1.06 | 0.61 | 1.80 | 4049 | 0.98 | 0.57 | 1.60 | 2029 |
| _ | M02 | 0.70 | 0.28 | 1.05 | 608 | 0.32 | 0.21 | 0.545 | 227 |
| _ | M24 | 0.63 | 0.30 | 0.808 | 94 | 0.00 | 0.00 | 0 | 0 |
| _ | M48 | 0.63 | 0.50 | 0.785 | 419 | 0.29 | 0.24 | 0.33 | 108 |
| _ | M8 | 3.76 | 3.18 | 4.01 | 34 | 0.00 | 0.00 | 0 | 0 |
| Coniferous | _ | _ | _ | _ | _ | _ | _ | _ | _ |
| TL | O | 0.72 | 0.33 | 1.24 | 2490 | 0.24 | 0.17 | 0.63 | 722 |
| _ | M02 | 0.36 | 0.20 | 0.68 | 10899 | 0.47 | 0.28 | 0.84 | 6990 |
| _ | M24 | 0.40 | 0.22 | 0.898 | 5081 | 0.40 | 0.22 | 0.83 | 2902 |
| _ | M48 | 0.44 | 0.21 | 0.9 | 13910 | 0.55 | 0.31 | 1.1 | 7518 |
| _ | M8 | 0.98 | 0.39 | 1.875 | 4990 | 0.93 | 0.50 | 2 | 1699 |
| ZTL | O | 0.40 | 0.20 | 0.76 | 4061 | 0.40 | 0.20 | 0.83 | 2789 |

| | M02 | 0.37 | 0.20 | 0.616 | 2773 | 0.49 | 0.38 | 0.6375 | 262 |
| | M24 | 0.44 | 0.25 | 0.927 | 644 | 0.00 | 0.00 | 0 | 0 |
| | M48 | 0.76 | 0.49 | 3.725 | 227 | 0.55 | 0.35 | 0.91 | 321 |
| | M8 | 0.85 | 0.37 | 1.3425 | 84 | 0.00 | 0.00 | 0 | 0 |

Table S9. Median soil solution S-$SO_4^{2-}$ (mg L$^{-1}$), 25% and 75% percentiles and number of observations (n) for the different forest types, soil depth intervals and collector types with the entire dataset (with breakpoints) and with the dataset without time series showing breakpoints (without breakpoints).

| | | WITH BREAKPOINTS | | | | WITHOUT BREAKPOINTS | | | |
|---|---|---|---|---|---|---|---|---|---|
| | | median $[SO_4^{2-}]$ | 25% percentile | 75% percentile | n | median $[SO_4^{2-}]$ | 25% percentile | 75% percentile | n |
| Broadleaved | | | | | | | | | |
| TL | O | 2.50 | 1.30 | 4.17 | 592 | 3.20 | 1.63 | 4.58 | 476 |
| | M02 | 2.00 | 1.33 | 3.3875 | 8383 | 1.93 | 1.19 | 3.3 | 3370 |
| | M24 | 2.63 | 1.60 | 3.8 | 2556 | 2.70 | 1.98 | 3.565 | 1007 |
| | M48 | 2.80 | 1.50 | 4.7 | 10571 | 3.10 | 1.90 | 5.5 | 5188 |
| | M8 | 4.04 | 2.83 | 6.371 | 4323 | 5.05 | 3.10 | 9.2 | 2116 |
| ZTL | O | 1.01 | 0.60 | 1.70 | 4041 | 0.86 | 0.53 | 1.40 | 2029 |
| | M02 | 0.75 | 0.52 | 1.21275 | 608 | 0.76 | 0.63 | 0.8785 | 227 |
| | M24 | 2.05 | 1.02 | 3.15975 | 94 | 0.00 | 0.00 | 0 | 0 |
| | M48 | 1.06 | 0.80 | 1.52 | 426 | 0.79 | 0.67 | 0.8625 | 108 |
| | M8 | 10.38 | 9.15 | 11.855 | 34 | 0.00 | 0.00 | 0 | 0 |
| Coniferous | | | | | | | | | |
| TL | O | 1.27 | 0.67 | 2.30 | 2483 | 0.80 | 0.46 | 1.37 | 722 |
| | M02 | 1.51 | 0.90 | 3 | 10885 | 1.94 | 1.08 | 3.608 | 7021 |
| | M24 | 2.39 | 1.40 | 3.862 | 5086 | 2.25 | 1.40 | 3.558 | 2933 |
| | M48 | 2.96 | 1.60 | 4.6 | 13941 | 2.90 | 1.70 | 4.63 | 7537 |
| | M8 | 4.34 | 2.42 | 7.2 | 4977 | 5.46 | 3.13 | 9.30125 | 1672 |
| ZTL | O | 0.71 | 0.34 | 1.48 | 4064 | 0.67 | 0.31 | 1.38 | 2800 |
| | M02 | 0.66 | 0.38 | 1.337 | 2776 | 0.57 | 0.42 | 0.77 | 261 |
| | M24 | 1.74 | 0.77 | 4.5975 | 644 | 0.00 | 0.00 | 0 | 0 |
| | M48 | 1.20 | 0.89 | 11.315 | 226 | 4.45 | 1.30 | 8.291 | 318 |
| | M8 | 1.33 | 1.09 | 1.60325 | 84 | 0.00 | 0.00 | 0 | 0 |

Table S10. Median soil solution N-NO$_3^-$ (mg L$^{-1}$), 25% and 75% percentiles and number of observations (n) for the different forest types, soil depth intervals and collector types with the entire dataset (with breakpoints) and with the dataset without time series showing breakpoints (without breakpoints).

| | | WITH BREAKPOINTS | | | | WITHOUT BREAKPOINTS | | | |
|---|---|---|---|---|---|---|---|---|---|
| | | median [NO$_3^-$] | 25% percentile | 75% percentile | n | median [NO$_3^-$] | 25% percentile | 75% percentile | n |
| Broadleaved | - | - | - | - | - | - | - | - | - |
| TL | O | 3.74 | 1.46 | 9.29 | 617 | 4.88 | 1.94 | 11.04 | 518 |
| - | M02 | 0.56 | 0.04 | 2.5285 | 8123 | 0.91 | 0.24 | 2.6825 | 3372 |
| - | M24 | 0.50 | 0.02 | 3.23 | 2535 | 0.62 | 0.02 | 2.8615 | 991 |
| - | M48 | 0.26 | 0.02 | 1.659 | 10358 | 0.33 | 0.03 | 2.3 | 5165 |
| - | M8 | 0.40 | 0.05 | 5.0275 | 4218 | 0.73 | 0.13 | 6.1595 | 2002 |
| ZTL | O | 1.60 | 0.56 | 3.79 | 3975 | 1.03 | 0.21 | 2.60 | 1994 |
| - | M02 | 0.86 | 0.40 | 1.8725 | 608 | 0.70 | 0.30 | 1.6 | 227 |
| - | M24 | 0.47 | 0.14 | 0.87975 | 94 | 0.00 | 0.00 | 0 | 0 |
| - | M48 | 0.35 | 0.06 | 0.8 | 423 | 0.52 | 0.23 | 0.8525 | 108 |
| - | M8 | 0.02 | 0.02 | 0.022 | 34 | 0.00 | 0.00 | 0 | 0 |
| Coniferous | - | - | - | - | - | - | - | - | - |
| TL | O | 1.14 | 0.16 | 4.19 | 2388 | 1.06 | 0.08 | 4.87 | 677 |
| - | M02 | 0.14 | 0.02 | 1.3 | 10431 | 0.27 | 0.02 | 1.87775 | 6940 |
| - | M24 | 0.17 | 0.02 | 1.267 | 4745 | 0.10 | 0.02 | 1.334 | 2844 |
| - | M48 | 0.10 | 0.02 | 1.2 | 13195 | 0.11 | 0.02 | 1.3 | 7194 |
| - | M8 | 0.27 | 0.02 | 1.0895 | 4971 | 0.37 | 0.06 | 1.2 | 1691 |
| ZTL | O | 0.56 | 0.13 | 1.74 | 4055 | 0.34 | 0.05 | 1.18 | 2777 |
| - | M02 | 0.02 | 0.02 | 0.06 | 2275 | 0.05 | 0.02 | 0.17 | 260 |
| - | M24 | 0.02 | 0.02 | 0.03 | 489 | 0.00 | 0.00 | 0 | 0 |
| - | M48 | 0.02 | 0.02 | 0.09875 | 226 | 0.65 | 0.03 | 7.988 | 321 |
| - | M8 | 2.54 | 0.50 | 4.6805 | 84 | 0.00 | 0.00 | 0 | 0 |

Table S11. Median soil solution Al (mg L$^{-1}$), 25% and 75% percentiles and number of observations (n) for the different forest types, soil depth intervals and collector types with the entire dataset (with breakpoints) and with the dataset without time series showing breakpoints (without breakpoints).

| | | WITH BREAKPOINTS | | | | WITHOUT BREAKPOINTS | | | |
|---|---|---|---|---|---|---|---|---|---|
| - | - | media | 25% | 75% | n | median | 25% | 75% | n |

| | | n [Al] | percentile | percentile | | [Al] | percentile | percentile | |
|---|---|---|---|---|---|---|---|---|---|
| Broadleaved | - | - | - | - | - | - | - | - | - |
| TL | O | 0.38 | 0.17 | 0.76 | 574 | 0.30 | 0.15 | 0.76 | 490 |
| - | M02 | 0.81 | 0.39 | 1.62 | 7767 | 0.78 | 0.30 | 1.7 | 3107 |
| - | M24 | 0.05 | 0.02 | 0.387 | 2406 | 0.05 | 0.02 | 0.333 | 979 |
| - | M48 | 0.30 | 0.02 | 1.02 | 9871 | 0.30 | 0.02 | 1 | 4918 |
| - | M8 | 0.05 | 0.02 | 0.87 | 4180 | 0.91 | 0.17 | 2.79 | 2101 |
| ZTL | O | 0.17 | 0.06 | 0.32 | 3278 | 0.12 | 0.03 | 0.22 | 1536 |
| - | M02 | 0.14 | 0.02 | 0.45 | 577 | 0.22 | 0.14 | 0.35 | 222 |
| - | M24 | 0.37 | 0.22 | 0.48 | 94 | 0.00 | 0.00 | 0 | 0 |
| - | M48 | 0.02 | 0.02 | 0.04 | 378 | 0.14 | 0.09 | 0.21 | 107 |
| - | M8 | 0.02 | 0.02 | 0.02 | 30 | 0.00 | 0.00 | 0 | 0 |
| Coniferous | - | - | - | - | - | - | - | - | - |
| TL | O | 1.14 | 0.74 | 1.79 | 2162 | 0.93 | 0.59 | 1.27 | 622 |
| - | M02 | 1.35 | 0.69 | 2.19 | 10398 | 1.44 | 0.72 | 2.44875 | 6514 |
| - | M24 | 0.92 | 0.36 | 2.2145 | 4871 | 0.90 | 0.38 | 2.391 | 2762 |
| - | M48 | 1.11 | 0.38 | 2.341 | 13454 | 0.96 | 0.32 | 2.2 | 7157 |
| - | M8 | 1.58 | 0.02 | 3.399 | 4857 | 2.63 | 1.01 | 5.475 | 1674 |
| ZTL | O | 0.24 | 0.12 | 0.49 | 3944 | 0.21 | 0.11 | 0.39 | 2704 |
| - | M02 | 0.87 | 0.44 | 1.48 | 2709 | 1.10 | 0.81 | 1.7 | 262 |
| - | M24 | 0.73 | 0.22 | 1.7235 | 611 | 0.00 | 0.00 | 0 | 0 |
| - | M48 | 2.01 | 1.20 | 7.015 | 210 | 2.95 | 1.90 | 5.568 | 303 |
| - | M8 | 1.62 | 1.01 | 2.3275 | 66 | 0.00 | 0.00 | 0 | 0 |

---

## Author Response (AR2)

**Answers to Referee 1**

The authors have provided a detailed response to my comments. However, they have not taken into account in their manuscript two important points I raised:

*1) a conceptual figure summarizing the main processes and factors that may control the DOC concentrations or fluxes. This article is aimed at a large audience (ecologists, biogeochemists, terrestrial and aquatic communities...) and thereby must accessible to this large audience and not to specialists only.*

A general conceptual figure showing the factors and processes affecting DOC has been added to the manuscript (Figure 1). We refer to it in the introduction, when we present the potential drivers of DOC changes.

*2) I would like to see a discussion on the possible bias linked to the fact you correlate DOC in surface water to DOC concentrations in soil and not to the amount of DOC flowing out of ecosystems.*

The final section *4.3. Link between DOC trends in soil and streams* has been modified and a new paragraph has been added at the end (lines 634-643 in revised manuscript). In this new paragraph, we discuss the potential implications of the fact that information on the hydrology of the site was not available for this study to calculate the fluxes. We also propose that future studies on DOC controls at large scales should take into account the hydrology.

**Answers to Referee 2**

*I agree with reviewer #2 that the authors have responded to most of the reviewers comments and that overall the manuscript has been improved. I also agree that the two unanswered points should be answered in the revised version. In addition, and unless I've missed this, however, there has been no analysis as to the extent of the spatial correlation between the decreasing trends in SO4 and the increasing trend in NOx at low to medium levels, which strikes me as important. I assume that the statistical methods should have taken care of the interdependency, but I think that the readers should be also informed about any potenential co-variation and likely effects on the outcome of the study.*

Indeed, the Structural Equation Models accounted for the co-variance between deposition of $SO_4^{2-}$ and $NO_3^-$ (it is represented by the arrow between Dep $SO_4^{2-}$ and Dep $NO_3^-$ in Figure 7 in the revised manuscript).
However, to inform the readers about the spatial correlation between $SO_4^{2-}$ and $NO_3^-$ deposition, and the DOC trends, a map showing the spatial variation of the $SO_4^{2-}$ and $NO_3^-$ deposition trends has been added (Figure 8). This map is now used in the results and discussion section to discuss the potential effect of co-variation between $SO_4^{2-}$ and $NO_3^-$ deposition trends on our results (lines 362-365 and 587-594 in the revised version of the manuscript).

*Minor comments:*
*remove line 534 and 559, or make new, numbered subheadings*

Lines 534 and 559 has been removed.

**Most relevant changes made in the manuscript**

1. A new conceptual figure (Figure 1) has been added to the manuscript.

2. A final paragraph has been added to the discussion "4.3. Link between DOC trends in soil and streams" (lines 634-643).

3. A map with the spatial variation in $SO_4^{2-}$ and $NO_3^-$ trends has been added (Figure 8).

4. The potential effect of covariation between $SO_4^{2-}$ and $NO_3^-$ is now discussed in lines 362-365 and 587-594.

[revised manuscript text omitted]

There was a good agreement between results using the three methods: BFAST, SMK, and PMK. The direction and significance of the trend agreed for 84.5% of the time series analyzed. For the majority of the remaining time series for which the trends did not agree, BFAST did not detect a trend when SMK and PMK did, thus, the latter two methods seemed more sensitive for trend detection than BFAST. Trends computed with SMK and PMK agreed well. The direction of the trend for SMK and PMK only differed for the intermediate soil layer (20-40 cm), as a result of the two extra sites for which SMK tests were performed, but not the PMK, that showed a marked positive trend (1.1 and 2 % yr $^{-1}$). However, when using exactly the same set of sites, the trend did not differ between the two methods.

**Implications of using standardized DOC slopes versus absolute DOC slopes.**

The standardized (relative) slopes of DOC concentrations were used for the study of the factors affecting the soil solution DOC trends (Fig. 5 and 6). The main reason for this decision was that using the median DOC concentration as a reference (as we did with the standardization) allowed us to determine whether the absolute trend in DOC was quantitatively large or not from an ecological perspective, because the absolute trend slope will be highly dependent on the initial DOC concentrations of the site.

The absolute trend slopes show the real magnitude and significance of the trend, but do not allow for comparison among sites or horizons. Since the aim of this study is to test whether there is a general DOC trend and to compare sites across Europe, we decided that using the relative slope was more consistent.

Moreover, due to limitations of the statistical analysis, we worked with time series per "plot-soil depth combinations", which means that different soil layers were mixed in the statistical analysis. Again, the standardization of the slopes of DOC concentrations allowed us to compare trends among different soil horizons by removing the effect of the decreasing soil solution DOC concentrations with soil depth. Otherwise, using the absolute trends would introduce a bias when we try to explain the DOC trends in relation with other parameters, because the trend slope would be highly dependent on the actual DOC concentrations, which, in turn, are very variable, not only among sites, but also among soil depths.

The influence of the DOC concentration levels was checked before deciding to use the standardized slopes (Fig. S5). It seemed that there was no relationship between the DOC trend slopes (relative and absolute) and the median DOC concentrations, with positive and negative trends occurring at both low and high DOC concentrations and, thus, we decided that using the standardized slopes will not hide any effect of the median DOC concentrations on the direction of the DOC trends.

This decision, however, has a drawback: the strength of the trend is clearly influenced by the DOC concentration levels. The fact that we used the standardized slope of DOC implied that it may be identical for two sites with very different mean DOC concentrations. DOC concentration decreases with depth and is lower in the deep mineral soil than in the upper mineral soil (Table S3) and by standardizing the slope, the magnitude of the trend was exaggerated in lower soil layers where both the absolute slope of DOC and the median DOC

concentration are low (Table S3). This issue is well illustrated in Fig. S5, that shows how the highest standardized slopes are usually at low DOC concentrations (mostly in mineral soil layers), while the highest absolute slopes are at higher DOC concentrations (mostly in organic and upper soil layers).

In other words, in quantitative terms DOC trends are much higher in the organic layer than in the mineral soils but, in relative terms, DOC is increasing in the same proportion (Table S3). Because the aim of this study is to explain the high heterogeneity of DOC trends found across Europe, instead of the quantification of the trends at local scale, the relative trends were discussed throughout the manuscript. Consequently, our results should be interpreted with caution, keeping in mind that the relations between DOC trends and explaining factors are discussed only from a relative point of view.

Nevertheless, the statistical analyses (LMM, SMK, PMK and BFAST) were done on the absolute value and the resulting Sen's slopes were then standardized. Thus, the fact that trends are expressed in relative terms has consequences on the interpretation of the results, but has no influence on the statistical test itself (carried out on the absolute values of DOC), that is, on the significance and direction of the trends.

Table S3. Comparison of median relative trend slope (rslope in % yr-1) and absolute trend slope (abs slope in mg $L^{-1}$ $yr^{-1}$) of DOC concentrations in soil solution and their interquartile range using the Seasonal Mann-Kendall test (SMK). (O: organic layer, M02: mineral soil 0-20 cm, M24: mineral soil 20-40 cm, M48: mineral soil 40-80 cm, M8: mineral soil > 80 cm.)

| Soil depth | rslope (% $yr^{-1}$) | abs slope (mg $L^{-1}$ $yr^{-1}$) |
| --- | --- | --- |
| O | 1.18 (±3.37) | 0.32 (±1.2) |
| M02 | 0.04 (±3.41) | 0.008 (±0.52) |
| M24 | 0.61 (±8.62) | 0.025 (±0.48) |
| M48 | 1.01 (±4.79) | 0.013 (±0.22) |
| M8 | 1.18 (±9.39) | 0.032 (±0.31) |

[Figure]

Figure S5. A) Standardized trends (relative DOC slope) versus median DOC concentrations. B) Absolute trends (absolute slope DOC) versus median DOC concentrations. The different colors represent the different soil layers.

**Information on the soil solution chemistry at the studied ICP Forests Level II plots**

Table S4. Median soil solution DOC concentrations (mg L$^{-1}$), 25% and 75% percentiles and number of observations (n) for the different forest types, soil depth intervals and collector types with the entire dataset (with breakpoints) and with the dataset without time series showing breakpoints (without breakpoints).

| | | WITH BREAKPOINTS | | | | WITHOUT BREAKPOINTS | | | |
|---|---|---|---|---|---|---|---|---|---|
| | | median [DOC] | 25% percentile | 75% percentile | n | median [DOC] | 25% percentile | 75% percentile | n |
| Broadleaved | | | | | | | | | |
| TL | O | 41.35 | 28.99 | 56.05 | 637 | 44.56 | 32.00 | 59.10 | 475 |
| | M02 | 8.80 | 4.30 | 21.20 | 8397 | 8.68 | 4.50 | 23.50 | 3104 |
| | M24 | 3.78 | 1.67 | 8.90 | 2584 | 3.19 | 1.85 | 4.76 | 928 |
| | M48 | 2.60 | 1.10 | 6.40 | 10635 | 2.70 | 1.08 | 5.80 | 4634 |
| | M8 | 2.60 | 1.17 | 6.53 | 4354 | 2.65 | 1.53 | 7.00 | 1797 |
| ZTL | O | 33.33 | 21.00 | 51.12 | 4057 | 30.88 | 18.01 | 51.10 | 1956 |
| | M02 | 4.26 | 3.51 | 6.28 | 608 | 4.30 | 2.80 | 9.30 | 192 |
| | M24 | 20.44 | 13.40 | 34.37 | 94 | | | | 0 |
| | M48 | 3.42 | 2.61 | 4.51 | 427 | 0.91 | 0.50 | 1.64 | 85 |
| | M8 | 2.42 | 2.11 | 3.62 | 34 | | | | 0 |
| Coniferous | | | | | | | | | |
| TL | O | 49.00 | 35.10 | 67.36 | 2496 | 50.90 | 38.20 | 65.40 | 693 |
| | M02 | 15.70 | 7.09 | 31.15 | 10914 | 12.80 | 5.90 | 25.50 | 5813 |
| | M24 | 5.72 | 2.40 | 16.50 | 5116 | 5.00 | 2.10 | 21.89 | 2476 |
| | M48 | 4.44 | 2.30 | 11.40 | 13979 | 4.30 | 2.29 | 10.90 | 6431 |
| | M8 | 3.70 | 1.60 | 7.91 | 5024 | 4.29 | 2.55 | 10.12 | 1597 |
| ZTL | O | 42.92 | 29.03 | 60.80 | 4079 | 44.60 | 30.18 | 60.80 | 2703 |
| | M02 | 36.90 | 22.20 | 56.40 | 2781 | 36.00 | 24.00 | 53.00 | 253 |
| | M24 | 16.34 | 8.76 | 31.59 | 645 | | | | 0 |
| | M48 | 44.00 | 17.40 | 62.35 | 227 | 13.70 | 10.30 | 36.25 | 251 |
| | M8 | 4.14 | 3.28 | 4.81 | 84 | | | | 0 |

Table S5. Median soil solution pH, 25% and 75% percentiles and number of observations (n) for the different forest types, soil depth intervals and collector types with the entire dataset (with breakpoints) and with the dataset without time series showing breakpoints (without breakpoints).

| | | WITH BREAKPOINTS | WITHOUT BREAKPOINTS |
|---|---|---|---|
| | | | |

|  |  | median pH | 25% percentile | 75% percentile | n | median pH | 25% percentile | 75% percentile | n |
|---|---|---|---|---|---|---|---|---|---|
| Broadleaved |  |  |  |  |  |  |  |  |  |
| TL | O | 3.9 | 3.8 | 4.1 | 636 | 3.90 | 3.80 | 4.10 | 518 |
|  | M02 | 4.5 | 4.2 | 5.2 | 8346 | 4.60 | 4.20 | 6.2 | 3322 |
|  | M24 | 6.3 | 4.9 | 7.1 | 2482 | 6.10 | 4.90 | 6.7 | 993 |
|  | M48 | 5.1 | 4.5 | 6.7 | 10496 | 5.10 | 4.40 | 6.5 | 5162 |
|  | M8 | 6.4 | 4.6 | 7.8 | 4228 | 4.50 | 4.30 | 6.46 | 2115 |
| ZTL | O | 5.30 | 4.40 | 6.30 | 4026 | 5.30 | 4.30 | 6.60 | 2025 |
|  | M02 | 6.15 | 5.00 | 7.6 | 608 | 5.00 | 4.80 | 5.75 | 227 |
|  | M24 | 4.70 | 4.50 | 5 | 93 | 0.00 | 0.00 | 0 | 0 |
|  | M48 | 8.30 | 8.20 | 8.4 | 426 | 5.20 | 5.10 | 5.3 | 108 |
|  | M8 | 8.20 | 8.00 | 8.3 | 34 | 0.00 | 0.00 | 0 | 0 |
| Coniferous |  |  |  |  |  |  |  |  |  |
| TL | O | 4.00 | 3.80 | 4.40 | 2496 | 3.80 | 3.60 | 4.00 | 726 |
|  | M02 | 4.30 | 4.00 | 4.7 | 10634 | 4.30 | 4.00 | 4.7 | 6930 |
|  | M24 | 4.60 | 4.30 | 5 | 4739 | 4.60 | 4.30 | 4.8 | 2849 |
|  | M48 | 4.50 | 4.30 | 4.9 | 13596 | 4.50 | 4.20 | 4.9 | 7462 |
|  | M8 | 4.57 | 4.30 | 6.4 | 4837 | 4.48 | 4.29 | 4.7 | 1660 |
| ZTL | O | 4.02 | 3.80 | 4.60 | 4038 | 4.00 | 3.80 | 4.80 | 2839 |
|  | M02 | 4.40 | 4.10 | 4.9 | 2412 | 4.80 | 4.53 | 5.3 | 254 |
|  | M24 | 4.90 | 4.50 | 5.4 | 551 | 0.00 | 0.00 | 0 | 0 |
|  | M48 | 4.80 | 4.10 | 5.1 | 225 | 4.40 | 4.27 | 4.9 | 319 |
|  | M8 | 4.70 | 4.60 | 4.8 | 84 | 0.00 | 0.00 | 0 | 0 |

Table S6. Median soil solution conductivity ($\mu S\ cm^{-1}$), 25% and 75% percentiles and number of observations (n) for the different forest types, soil depth intervals and collector types with the entire dataset (with breakpoints) and with the dataset without time series showing breakpoints (without breakpoints).

|  |  | WITH BREAKPOINTS | | | | WITHOUT BREAKPOINTS | | | |
|---|---|---|---|---|---|---|---|---|---|
|  |  | median COND | 25% percentile | 75% percentile | n | median COND | 25% percentile | 75% percentile | n |
| Broadleaved |  |  |  |  |  |  |  |  |  |
| TL | O | 128.00 | 93.50 | 189.50 | 631 | 140.00 | 103.00 | 212.50 | 507 |
|  | M02 | 60.00 | 42.25 | 99 | 7651 | 69.55 | 45.00 | 104 | 3066 |
|  | M24 | 86.00 | 47.00 | 180 | 1503 | 70.45 | 45.90 | 120 | 548 |
|  | M48 | 68.00 | 45.00 | 137 | 8538 | 70.00 | 48.58 | 145 | 4320 |
|  | M8 | 148.50 | 61.63 | 305.75 | 3006 | 133.00 | 59.00 | 210 | 1736 |

| | | | | | | | | |
|---|---|---|---|---|---|---|---|---|
| ZTL | O | 71.00 | 48.00 | 110.00 | 2750 | 70.00 | 46.60 | 111.00 | 1489 |
| | M02 | 63.35 | 34.00 | 86.775 | 608 | 28.20 | 19.10 | 51.05 | 227 |
| | M24 | 44.00 | 28.00 | 56 | 93 | 0.00 | 0.00 | 0 | 0 |
| | M48 | 282.00 | 254.00 | 318 | 425 | 19.30 | 16.38 | 25.325 | 108 |
| | M8 | 485.50 | 446.50 | 539.75 | 34 | 0.00 | 0.00 | 0 | 0 |
| Coniferous | | | | | | | | | |
| TL | O | 77.00 | 56.00 | 124.00 | 2425 | 85.00 | 65.00 | 155.00 | 725 |
| | M02 | 58.00 | 31.00 | 92 | 9222 | 61.00 | 33.00 | 105.5 | 5699 |
| | M24 | 50.00 | 30.00 | 97 | 2954 | 56.00 | 31.00 | 111 | 1715 |
| | M48 | 56.00 | 37.00 | 94 | 10270 | 56.00 | 37.20 | 99 | 6658 |
| | M8 | 104.00 | 55.00 | 207.75 | 2850 | 120.50 | 66.00 | 259 | 1118 |
| ZTL | O | 65.30 | 45.00 | 104.00 | 2296 | 64.00 | 42.30 | 106.00 | 1537 |
| | M02 | 39.20 | 25.00 | 59 | 2627 | 27.00 | 20.08 | 41.1 | 228 |
| | M24 | 32.00 | 21.00 | 57.95 | 615 | 0.00 | 0.00 | 0 | 0 |
| | M48 | 39.05 | 28.00 | 150.5 | 214 | 95.85 | 46.48 | 155.5 | 290 |
| | M8 | 50.00 | 31.75 | 69.25 | 84 | 0.00 | 0.00 | 0 | 0 |

Table S7. Median soil solution Ca (mg L$^{-1}$), 25% and 75% percentiles and number of observations (n) for the different forest types, soil depth intervals and collector types with the entire dataset (with breakpoints) and with the dataset without time series showing breakpoints (without breakpoints).

| | | WITH BREAKPOINTS | | | | WITHOUT BREAKPOINTS | | | |
|---|---|---|---|---|---|---|---|---|---|
| | | median [Ca] | 25% percentile | 75% percentile | n | median [Ca] | 25% percentile | 75% percentile | n |
| Broadleaved | | | | | | | | | |
| TL | O | 4.18 | 1.83 | 7.85 | 633 | 5.369 | 3.193 | 9.204 | 515 |
| | M02 | 2.12 | 0.80 | 5.3 | 8381 | 2.80 | 1.04 | 9.56525 | 3396 |
| | M24 | 4.09 | 1.50 | 14.18 | 2555 | 3.69 | 0.92 | 9.005 | 999 |
| | M48 | 2.31 | 0.70 | 9.385 | 10600 | 2.80 | 0.92 | 7.7 | 5204 |
| | M8 | 5.68 | 1.50 | 41.7825 | 4322 | 2.80 | 0.51 | 13.75 | 2151 |
| ZTL | O | 4.10 | 2.05 | 7.06 | 4049 | 3.90 | 1.40 | 6.36 | 2030 |
| | M02 | 8.33 | 1.67 | 13.59 | 608 | 1.23 | 0.75 | 2.425 | 227 |
| | M24 | 2.35 | 1.25 | 3.296 | 94 | 0.00 | 0.00 | 0 | 0 |
| | M48 | 58.86 | 51.26 | 67.485 | 419 | 0.72 | 0.58 | 1.06 | 108 |
| | M8 | 73.75 | 60.78 | 92.8 | 34 | 0.00 | 0.00 | 0 | 0 |
| Coniferous | | | | | | | | | |

| | | median | 25% | 75% | n | median | 25% | 75% | n |
|---|---|---|---|---|---|---|---|---|---|
| TL | O | 3.36 | 1.47 | 6.39 | 2490 | 1.55 | 0.98 | 3.66 | 722 |
| | M02 | 0.66 | 0.25 | 1.72 | 10890 | 1.00 | 0.36 | 2.45 | 6985 |
| | M24 | 0.82 | 0.30 | 1.8665 | 5079 | 0.90 | 0.30 | 1.61 | 2901 |
| | M48 | 0.82 | 0.32 | 2.07 | 13901 | 0.92 | 0.32 | 2.285 | 7511 |
| | M8 | 2.10 | 0.49 | 10.6575 | 4986 | 1.97 | 0.53 | 8.285 | 1700 |
| ZTL | O | 1.50 | 0.72 | 2.80 | 4052 | 1.50 | 0.72 | 2.80 | 4052 |
| | M02 | 1.13 | 0.53 | 2.14 | 2777 | 1.13 | 0.53 | 2.14 | 2777 |
| | M24 | 1.20 | 0.62 | 2.31 | 644 | 1.20 | 0.62 | 2.31 | 644 |
| | M48 | 3.00 | 1.81 | 3.895 | 227 | 3.00 | 1.81 | 3.895 | 227 |
| | M8 | 0.76 | 0.47 | 1.1975 | 84 | 0.76 | 0.47 | 1.1975 | 84 |

Table S8. Median soil solution Mg (mg L$^{-1}$), 25% and 75% percentiles and number of observations (n) for the different forest types, soil depth intervals and collector types with the entire dataset (with breakpoints) and with the dataset without time series showing breakpoints (without breakpoints).

| | | WITH BREAKPOINTS | | | | WITHOUT BREAKPOINTS | | | |
|---|---|---|---|---|---|---|---|---|---|
| | | median [Mg] | 25% percentile | 75% percentile | n | median [Mg] | 25% percentile | 75% percentile | n |
| Broadleaved | | | | | | | | | |
| TL | O | 1.05 | 0.48 | 1.90 | 633 | 1.18 | 0.62 | 2.08 | 515 |
| | M02 | 0.80 | 0.42 | 1.5 | 8382 | 0.86 | 0.51 | 1.46 | 3395 |
| | M24 | 1.01 | 0.50 | 2.13 | 2563 | 1.18 | 0.62 | 2.295 | 999 |
| | M48 | 0.95 | 0.37 | 2.0745 | 10611 | 1.02 | 0.46 | 2.19 | 5205 |
| | M8 | 1.72 | 0.73 | 3.94 | 4323 | 1.29 | 0.51 | 2.88 | 2152 |
| ZTL | O | 1.06 | 0.61 | 1.80 | 4049 | 0.98 | 0.57 | 1.60 | 2029 |
| | M02 | 0.70 | 0.28 | 1.05 | 608 | 0.32 | 0.21 | 0.545 | 227 |
| | M24 | 0.63 | 0.30 | 0.808 | 94 | 0.00 | 0.00 | 0 | 0 |
| | M48 | 0.63 | 0.50 | 0.785 | 419 | 0.29 | 0.24 | 0.33 | 108 |
| | M8 | 3.76 | 3.18 | 4.01 | 34 | 0.00 | 0.00 | 0 | 0 |
| Coniferous | | | | | | | | | |
| TL | O | 0.72 | 0.33 | 1.24 | 2490 | 0.24 | 0.17 | 0.63 | 722 |
| | M02 | 0.36 | 0.20 | 0.68 | 10899 | 0.47 | 0.28 | 0.84 | 6990 |
| | M24 | 0.40 | 0.22 | 0.898 | 5081 | 0.40 | 0.22 | 0.83 | 2902 |
| | M48 | 0.44 | 0.21 | 0.9 | 13910 | 0.55 | 0.31 | 1.1 | 7518 |
| | M8 | 0.98 | 0.39 | 1.875 | 4990 | 0.93 | 0.50 | 2 | 1699 |
| ZTL | O | 0.40 | 0.20 | 0.76 | 4061 | 0.40 | 0.20 | 0.83 | 2789 |
| | M02 | 0.37 | 0.20 | 0.616 | 2773 | 0.49 | 0.38 | 0.6375 | 262 |

| | 0.44 | 0.25 | 0.927 | 644 | 0.00 | 0.00 | 0 | 0 |
|---|---|---|---|---|---|---|---|---|
| M24 | | | | | | | | |
| M48 | 0.76 | 0.49 | 3.725 | 227 | 0.55 | 0.35 | 0.91 | 321 |
| M8 | 0.85 | 0.37 | 1.3425 | 84 | 0.00 | 0.00 | 0 | 0 |

Table S9. Median soil solution $S\text{-}SO_4^{2-}$ (mg $L^{-1}$), 25% and 75% percentiles and number of observations (n) for the different forest types, soil depth intervals and collector types with the entire dataset (with breakpoints) and with the dataset without time series showing breakpoints (without breakpoints).

| | | WITH BREAKPOINTS | | | | WITHOUT BREAKPOINTS | | | |
|---|---|---|---|---|---|---|---|---|---|
| | | median $[SO_4^{2-}]$ | 25% percentile | 75% percentile | n | median $[SO_4^{2-}]$ | 25% percentile | 75% percentile | n |
| Broadleaved | | | | | | | | | |
| TL | O | 2.50 | 1.30 | 4.17 | 592 | 3.20 | 1.63 | 4.58 | 476 |
| | M02 | 2.00 | 1.33 | 3.3875 | 8383 | 1.93 | 1.19 | 3.3 | 3370 |
| | M24 | 2.63 | 1.60 | 3.8 | 2556 | 2.70 | 1.98 | 3.565 | 1007 |
| | M48 | 2.80 | 1.50 | 4.7 | 10571 | 3.10 | 1.90 | 5.5 | 5188 |
| | M8 | 4.04 | 2.83 | 6.371 | 4323 | 5.05 | 3.10 | 9.2 | 2116 |
| ZTL | O | 1.01 | 0.60 | 1.70 | 4041 | 0.86 | 0.53 | 1.40 | 2029 |
| | M02 | 0.75 | 0.52 | 1.21275 | 608 | 0.76 | 0.63 | 0.8785 | 227 |
| | M24 | 2.05 | 1.02 | 3.15975 | 94 | 0.00 | 0.00 | 0 | 0 |
| | M48 | 1.06 | 0.80 | 1.52 | 426 | 0.79 | 0.67 | 0.8625 | 108 |
| | M8 | 10.38 | 9.15 | 11.855 | 34 | 0.00 | 0.00 | 0 | 0 |
| Coniferous | | | | | | | | | |
| TL | O | 1.27 | 0.67 | 2.30 | 2483 | 0.80 | 0.46 | 1.37 | 722 |
| | M02 | 1.51 | 0.90 | 3 | 10885 | 1.94 | 1.08 | 3.608 | 7021 |
| | M24 | 2.39 | 1.40 | 3.862 | 5086 | 2.25 | 1.40 | 3.558 | 2933 |
| | M48 | 2.96 | 1.60 | 4.6 | 13941 | 2.90 | 1.70 | 4.63 | 7537 |
| | M8 | 4.34 | 2.42 | 7.2 | 4977 | 5.46 | 3.13 | 9.30125 | 1672 |
| ZTL | O | 0.71 | 0.34 | 1.48 | 4064 | 0.67 | 0.31 | 1.38 | 2800 |
| | M02 | 0.66 | 0.38 | 1.337 | 2776 | 0.57 | 0.42 | 0.77 | 261 |
| | M24 | 1.74 | 0.77 | 4.5975 | 644 | 0.00 | 0.00 | 0 | 0 |
| | M48 | 1.20 | 0.89 | 11.315 | 226 | 4.45 | 1.30 | 8.291 | 318 |
| | M8 | 1.33 | 1.09 | 1.60325 | 84 | 0.00 | 0.00 | 0 | 0 |

Table S10. Median soil solution $N\text{-}NO_3^-$ (mg $L^{-1}$), 25% and 75% percentiles and number of observations (n) for the different forest types, soil depth intervals and collector types with the entire dataset (with breakpoints) and with the dataset without time series showing breakpoints (without breakpoints).

|  |  | WITH BREAKPOINTS | | | | WITHOUT BREAKPOINTS | | | |
| --- | --- | --- | --- | --- | --- | --- | --- | --- | --- |
|  |  | median [NO$_3^-$] | 25% percentile | 75% percentile | n | median [NO$_3^-$] | 25% percentile | 75% percentile | n |
| Broadleaved |  |  |  |  |  |  |  |  |  |
| TL | O | 3.74 | 1.46 | 9.29 | 617 | 4.88 | 1.94 | 11.04 | 518 |
|  | M02 | 0.56 | 0.04 | 2.5285 | 8123 | 0.91 | 0.24 | 2.6825 | 3372 |
|  | M24 | 0.50 | 0.02 | 3.23 | 2535 | 0.62 | 0.02 | 2.8615 | 991 |
|  | M48 | 0.26 | 0.02 | 1.659 | 10358 | 0.33 | 0.03 | 2.3 | 5165 |
|  | M8 | 0.40 | 0.05 | 5.0275 | 4218 | 0.73 | 0.13 | 6.1595 | 2002 |
| ZTL | O | 1.60 | 0.56 | 3.79 | 3975 | 1.03 | 0.21 | 2.60 | 1994 |
|  | M02 | 0.86 | 0.40 | 1.8725 | 608 | 0.70 | 0.30 | 1.6 | 227 |
|  | M24 | 0.47 | 0.14 | 0.87975 | 94 | 0.00 | 0.00 | 0 | 0 |
|  | M48 | 0.35 | 0.06 | 0.8 | 423 | 0.52 | 0.23 | 0.8525 | 108 |
|  | M8 | 0.02 | 0.02 | 0.022 | 34 | 0.00 | 0.00 | 0 | 0 |
| Coniferous |  |  |  |  |  |  |  |  |  |
| TL | O | 1.14 | 0.16 | 4.19 | 2388 | 1.06 | 0.08 | 4.87 | 677 |
|  | M02 | 0.14 | 0.02 | 1.3 | 10431 | 0.27 | 0.02 | 1.87775 | 6940 |
|  | M24 | 0.17 | 0.02 | 1.267 | 4745 | 0.10 | 0.02 | 1.334 | 2844 |
|  | M48 | 0.10 | 0.02 | 1.2 | 13195 | 0.11 | 0.02 | 1.3 | 7194 |
|  | M8 | 0.27 | 0.02 | 1.0895 | 4971 | 0.37 | 0.06 | 1.2 | 1691 |
| ZTL | O | 0.56 | 0.13 | 1.74 | 4055 | 0.34 | 0.05 | 1.18 | 2777 |
|  | M02 | 0.02 | 0.02 | 0.06 | 2275 | 0.05 | 0.02 | 0.17 | 260 |
|  | M24 | 0.02 | 0.02 | 0.03 | 489 | 0.00 | 0.00 | 0 | 0 |
|  | M48 | 0.02 | 0.02 | 0.09875 | 226 | 0.65 | 0.03 | 7.988 | 321 |
|  | M8 | 2.54 | 0.50 | 4.6805 | 84 | 0.00 | 0.00 | 0 | 0 |

Table S11. Median soil solution Al (mg L$^{-1}$), 25% and 75% percentiles and number of observations (n) for the different forest types, soil depth intervals and collector types with the entire dataset (with breakpoints) and with the dataset without time series showing breakpoints (without breakpoints).

|  |  | WITH BREAKPOINTS | | | | WITHOUT BREAKPOINTS | | | |
| --- | --- | --- | --- | --- | --- | --- | --- | --- | --- |
|  |  | median [Al] | 25% percentile | 75% percentile | n | median [Al] | 25% percentile | 75% percentile | n |
| Broadleaved |  |  |  |  |  |  |  |  |  |

| | | | | | | | | | |
|---|---|---|---|---|---|---|---|---|---|
| TL | O | 0.38 | 0.17 | 0.76 | 574 | 0.30 | 0.15 | 0.76 | 490 |
| | M02 | 0.81 | 0.39 | 1.62 | 7767 | 0.78 | 0.30 | 1.7 | 3107 |
| | M24 | 0.05 | 0.02 | 0.387 | 2406 | 0.05 | 0.02 | 0.333 | 979 |
| | M48 | 0.30 | 0.02 | 1.02 | 9871 | 0.30 | 0.02 | 1 | 4918 |
| | M8 | 0.05 | 0.02 | 0.87 | 4180 | 0.91 | 0.17 | 2.79 | 2101 |
| ZTL | O | 0.17 | 0.06 | 0.32 | 3278 | 0.12 | 0.03 | 0.22 | 1536 |
| | M02 | 0.14 | 0.02 | 0.45 | 577 | 0.22 | 0.14 | 0.35 | 222 |
| | M24 | 0.37 | 0.22 | 0.48 | 94 | 0.00 | 0.00 | 0 | 0 |
| | M48 | 0.02 | 0.02 | 0.04 | 378 | 0.14 | 0.09 | 0.21 | 107 |
| | M8 | 0.02 | 0.02 | 0.02 | 30 | 0.00 | 0.00 | 0 | 0 |
| Coniferous | | | | | | | | | |
| TL | O | 1.14 | 0.74 | 1.79 | 2162 | 0.93 | 0.59 | 1.27 | 622 |
| | M02 | 1.35 | 0.69 | 2.19 | 10398 | 1.44 | 0.72 | 2.44875 | 6514 |
| | M24 | 0.92 | 0.36 | 2.2145 | 4871 | 0.90 | 0.38 | 2.391 | 2762 |
| | M48 | 1.11 | 0.38 | 2.341 | 13454 | 0.96 | 0.32 | 2.2 | 7157 |
| | M8 | 1.58 | 0.02 | 3.399 | 4857 | 2.63 | 1.01 | 5.475 | 1674 |
| ZTL | O | 0.24 | 0.12 | 0.49 | 3944 | 0.21 | 0.11 | 0.39 | 2704 |
| | M02 | 0.87 | 0.44 | 1.48 | 2709 | 1.10 | 0.81 | 1.7 | 262 |
| | M24 | 0.73 | 0.22 | 1.7235 | 611 | 0.00 | 0.00 | 0 | 0 |
| | M48 | 2.01 | 1.20 | 7.015 | 210 | 2.95 | 1.90 | 5.568 | 303 |
| | M8 | 1.62 | 1.01 | 2.3275 | 66 | 0.00 | 0.00 | 0 | 0 |